# Supersulphides provide airway protection in viral and chronic lung diseases

Tetsuro Matsunaga [1,12], Hirohito Sano [2,12], Katsuya Takita[2,12], Masanobu Morita[1,12], Shun Yamanaka[2], Tomohiro Ichikawa[2], Tadahisa Numakura[2,13], Tomoaki Ida [1,13], Minkyung Jung[1], Seiryo Ogata[1], Sunghyeon Yoon[1], Naoya Fujino [2], Yorihiko Kyogoku[2], Yusaku Sasaki[2], Akira Koarai [2], Tsutomu Tamada[2], Atsuhiko Toyama[3], Takakazu Nakabayashi[4], Lisa Kageyama [4], Shigeru Kyuwa[5], Kenji Inaba [6], Satoshi Watanabe[6], Péter Nagy [7], Tomohiro Sawa[8], Hiroyuki Oshiumi [9], Masakazu Ichinose[2], Mitsuhiro Yamada [2,13], Hisatoshi Sugiura [2] ✉, Fan-Yan Wei [10], Hozumi Motohashi [11] ✉ & Takaaki Akaike [1] ✉

Supersulphides are inorganic and organic sulphides with sulphur catenation with diverse physiological functions. Their synthesis is mainly mediated by mitochondrial cysteinyl-tRNA synthetase (CARS2) that functions as a principal cysteine persulphide synthase (CPERS). Here, we identify protective functions of supersulphides in viral airway infections (influenza and COVID-19), in aged lungs and in chronic lung diseases, including chronic obstructive pulmonary disease (COPD), idiopathic pulmonary fibrosis (IPF). We develop a method for breath supersulphur-omics and demonstrate that levels of exhaled super-sulphides increase in people with COVID-19 infection and in a hamster model of SARS-CoV-2 infection. Lung damage and subsequent lethality that result from oxidative stress and inflammation in mouse models of COPD, IPF, and ageing were mitigated by endogenous supersulphides production by CARS2/CPERS or exogenous administration of the supersulphide donor glutathione trisulphide. We revealed a protective role of supersulphides in airways with various viral or chronic insults and demonstrated the potential of targeting supersulphides in lung disease.

Supersulphides, which we currently define as hydropersulphide (RSSH) and polymeric sulphur species with sulphur catenation ($RSS_nR$, $n > 1$, R = hydrogen or alkyl, or cyclized polysulphides), are recognized in recent years as endogenously produced metabolites that are abundant in mammalian (including human) cells and tissues[1–3]. They are also known as universal bioactive metabolites formed physiologically in all organisms[4,5]. The most typical super-sulphides that are widely distributed among different organisms include various reactive hydropersulphides such as cysteine per-sulphide (CysSSH), glutathione persulphide (GSSH), and oxidized glutathione trisulphide (GSSSG); they are more redox-active than are other simple thiols and disulphides (see Supplementary Fig. 1 for the physicochemical and biochemical properties). Hydrogen sulphide ($H_2S$) was recently suggested to be a small signaling molecule; for example, the reaction of $H_2S$ with nitrosylated and sulphenylated cysteine residues may be physiologically relevant in terms of $H_2S$-mediated redox signaling[6,7]. Nevertheless, it actually acts as a marker for functionally active supersulphides, because many of the reported biological activities associated with $H_2S$ are apparently those of supersulphides, and in fact $H_2S$ is a major degradation product of supersulphides or, on some occasions, is their artefactual product[2,8]. Reactive persulphides truly act as

strong antioxidant and redox signaling molecules[1–3,9–14], which are thereby likely to improve, for example, chronic heart failure by reducing oxidative or electrophilic stress-induced cellular senescence[15].

Oxidative stress associated with airway inflammation has been implicated in the pathogenesis of influenza, COVID-19, and chronic lung disorders including chronic obstructive pulmonary disease (COPD), emphysema, idiopathic pulmonary fibrosis (IPF), and even lung ageing[16–23]. Our previous studies showed that endogenous production of supersulphides seemed to be reduced in COPD and related inflammatory airway diseases[24,25]. These observations prompted us to hypothesize a beneficial role of supersulphides in protecting the airways from oxidative stress occurring during various viral and inflammatory diseases including influenza and COVID-19. In fact, we recently reported that cysteinyl-tRNA synthetases (CARS) acted as the principal cysteine persulphide synthases (CPERS) in mammals and contributed solely to endogenous supersulphide production[26], whereas CARS2, a mitochondrial isoform of CARS, mediated mitochondrial biogenesis and bioenergetics via CysSSH production[26]. The physiological roles of supersulphides and their producing enzymes CARS/CPERS in various airway diseases are not fully understood, however. We report here that the innate defense functions of supersulphides, which are highly conserved among organisms, efficiently protect the lung and airways as well as their associated tissues such as vasculature against viral infections including influenza and COVID-19 and even chronic inflammatory lung injuries, e.g., COPD, pulmonary fibrosis, and ageing. Our current work may also have a significant translational impact on breath sulphur-omics technology that may be applied to clinical diagnosis and therapeutics for viral and chronic lung diseases.

## Results

### Anti-influenza defense by supersulphides

Although supersulphides are thought to have diverse physiological functions because of their various chemical reactivities, we identified that supersulphides have antiviral effects on influenza virus (Fig. 1) and SARS-CoV-2 (Figs. 2–4). As Fig. 1 and Supplementary Fig. 2 show, Cars2-deficient (Cars2[+/−]) mice and their wild-type wild type (WT) littermates were used to generate a mouse model of viral pneumonia caused by influenza virus A/PR8/34 (H1N1). We demonstrated a beneficial and protective effect of supersulphides in the influenza model. Endogenous formation supersulphides (e.g., CysSSH, GSSH, HSH, and HSSH) in lungs was found to be lower in Cars2[+/−] mice than in WT mice especially before (day 0) influenza infection (Supplementary Fig. 3). The viral infection affected Cars2[+/−] mice more extensively than did WT mice in terms of lung tissue damage and lethality, as well as inflammatory changes in the infected lungs, as assessed by pathological examinations and survival rate analysis (Fig. 1a and Supplementary Fig. 2). The influenza viral growth in the lung significantly increased in Cars2[+/−] mice compared with WT mice (Fig. 1d), which implied the antiviral effect of supersulphides on influenza virus replication in vivo. The direct anti-influenza viral effect of supersulphides was confirmed by the finding shown in Fig. 1e, which verified that treatment of influenza virus (PR8) with inorganic supersulphide donors (hydropolysulphides, $H_2S_n$) such as $Na_2S_{2-4}$ remarkably attenuated the viral infectivity (Supplementary Fig. 3b), possibly through cleavage of disulphide bridges of the viral hemagglutinin (HA), as discussed later.

Also, supersulphides produced by CARS2 are likely to alleviate not only cytokine storm-like excessive inflammatory responses but also nitrative and oxidative stress (Fig. 1 and Supplementary Fig. 2), all of which are critically involved in influenza pathogenesis, as we reported earlier[21–25,27,28]. For example, we found that influenza infection increased various biomarkers of oxidative (nitrative) stress, which include 8-hydroxy-2′-deoxyguanosine (8-OHdG) (Fig. 1i and Supplementary Fig. 2i), GSSG/GSH ratio (GSSG/GSH) (Fig. 1j), 3-nitrotyrosine

(3-NT) formation in Cars2[+/−] mice after infection (Supplementary Fig. 2e).

As an interesting finding, influenza virus infection per se reduced endogenous supersulphide production (Supplementary Fig. 3a), which may aid viral propagation and thereby accelerate inflammatory responses in virus-infected lungs (Fig. 1 and Supplementary Fig. 2). In fact, intranasal administration of the supersulphide donor GSSSG that we developed markedly reduced all pathological and inflammatory consequences, in influenza-infected WT animals (Fig. 1k–p and Supplementary Fig. 2f–j). This result suggests that supersulphides supplied exogenously compensate for impaired protective functions of supersulphides that are depleted by oxidative stress and excessive inflammatory reactions, e.g., a cytokine storm, in hosts, so that the exogenous supersulphides protect airways from detrimental effects of influenza.

### Anti-coronavirus activities of supersulphides in cells

The anti-coronavirus effects of supersulphides were also observed with both in vitro and in vivo models of the SARS-CoV-2 infection (Figs. 2 and 3). Of note, the SARS-CoV-2 replication was significantly higher, when endogenous supersulphides formation was impaired by knocking down (KD) the CARS2/CPERS expression in the VeroE6 cells overexpressing the type II transmembrane serine protease TMPRSS2 (VeroE6/TMPRSS2) cells in culture (Fig. 2 and Supplementary Figs. 4 and 5), even though SARS virus can reportedly propagate much more efficiently in VeroE6/TMPRSS2 cells than in the original VeroE6 cells[29]. The concentration of all supersulphide metabolites that we can assess and quantify is estimated to be around a hundred micromolar or so in various cell lines used in our present and previous studies (Supplementary Fig. 5b, c)[2,26]. It was highly plausible that the endogenous supersulphides could have anti-SARS-CoV-2 activity in cells during the viral infection. Indeed, the increased viral yield observed in the CARS2/CPERS KD cells was reversed by the exogenously added GSSSG, but not by GSH and GSSG (Fig. 2 and Supplementary Fig. 4). When we carefully examined the profile of supersulphide metabolites with or without CARS2/CPERS KD using our sulphur metabolome, the amount of supersulphides was found to be decreased by approximately 30 μM (Supplementary Fig. 5c), by which the SARS-CoV-2 production was remarkably elevated by almost threefolds, as assessed by the plaque-reduction assay. This indicates that endogenous and baseline formation of supersulphides should contribute significantly to the innate or naturally occurring antiviral effect of the host's cells. More importantly, such an attenuated antiviral effect was significantly restored by addition of GSSSG at the same range of concentrations, supporting the above notion that even baseline supersulphides have physiologically relevant defense-oriented consequences against SARS-CoV-2 replication. This interpretation is also substantiated by significant increases in the levels of various supersulphide metabolites in VeroE6/TMPRSS2 cells after treatment of the cells with GSSSG (Supplementary Fig. 6). Notably, the direct anti-SARS-CoV-2 effect was also observed with inorganic hydropolysulphides (their donors polysulphides such as $Na_2S_{2-4}$), but not with GSH, GSSG, and the $H_2S$ donor $Na_2S$ (Fig. 2e, f). It is therefore conceivable that 10–100 μM concentrations are likely required to achieve appreciable antiviral effects for GSSSG administered exogenously to the SARS-CoV-2-infected cells. A similar anti-coronaviral effect of GSSSG was shown in Supplementary Fig. 4i with mouse hepatitis virus (MHV).

### Anti-SARS-CoV-2 defense by supersulphides in vivo

We carried out SARS-CoV-2 infection in Cars2 mutant mice (Fig. 3a and Supplementary Figs. 7 and 8a), i.e., AINK mutant mice that possess full tRNA synthetase activity but show impaired supersulphides production, crossed with ACE2-transgenic mice[30]. By using CRISPR-Cas9 genome editing technology, we introduced non-synonymous point mutations that replaced the pyridoxal-5′-phosphate-binding motif

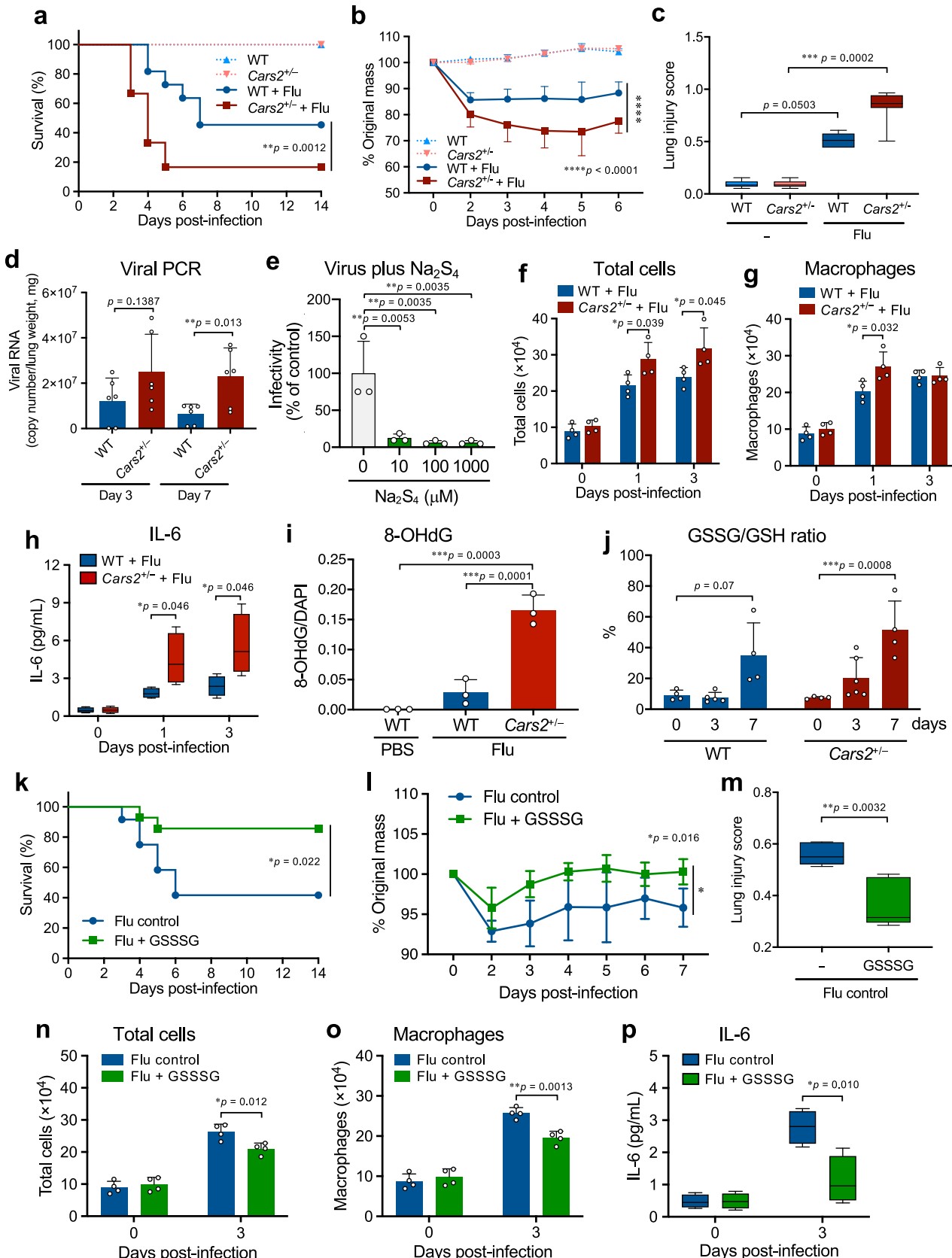

KIIK, which is critical for the CPERS activity of CARS2, with AINK to generate the *Cars2*<sup>AINK</sup> allele (Supplementary Fig. 7a). Homozygous *Cars2*<sup>AINK/AINK</sup> mice were embryonic lethal (Supplementary Fig. 7b), which indicated that CARS2-mediated mitochondrial supersulphide production is essential for embryonic development. The amounts of all supersulphide-related metabolites in lungs of *Cars2*<sup>AINK/+</sup> mice were reduced by about 50% (vs. WT littermates), as assessed by sulphur-omics analysis (Supplementary Fig. 7e), whereas mitochondrial

**Fig. 1 | Supersulphide-mediated anti-influenza host defense in mice.**
**a** Kaplan−Meier survival curves for WT ($n = 11$) and $Cars2^{+/-}$ ($n = 12$) mice infected with influenza A virus (Flu). **b** Mean body weight changes (95% confidence interval, CI) of Flu-infected WT ($n = 8$) and $Cars2^{+/-}$ ($n = 11$) mice. None-infected groups (**a**, **b**): WT and $Cars2^{+/-}$ ($n = 3$). **c** Quantification of lung injury scores for hematoxylin and eosin (HE) staining images of lungs: Flu (−) WT, $n = 7$, Flu (−) $Cars2^{+/-}$, Flu (+) WT, $n = 9$; Flu (+) $Cars2^{+/-}$, $n = 8$ (3 days post-infection). Supplementary Fig. 2a shows representative images. **d**, Numbers of viral RNA in lungs at days 3 and 7 after infection. **e** Viral infectivity of Flu attenuated with 10, 100, and 1000 μM $Na_2S_4$ (37 °C for 30 min). **f**, **g** Total cells (**f**) and macrophages (**g**) in BALF of Flu-infected mice. **h** IL-6 concentration in BALF from Flu-infected mice ($n = 4$). **i** 8-OHdG levels in Flu-infected WT and $Cars2^{+/-}$ mice ($n = 3$). **j** GSSG/GSH ratio in lungs in Flu-infected WT and $Cars2^{+/-}$ mice ($n = 4-6$), determined by supersulphide metabolome shown in Supplementary Fig. 3a. **k** Kaplan−Meier survival curves for Flu-infected WT mice treated with PBS as control ($n = 12$) or 32 μg GSSSG ($n = 14$). **l** Mean body weight changes (95% CI) in Flu-infected WT mice given PBS ($n = 4$) or 32 μg GSSSG ($n = 4$). **m** Lung injury scores of WT mice (day 8 after infection) treated with PBS ($n = 5$) or 32 μg GSSSG ($n = 5$). Supplementary Fig. 2f shows representative images. **n**, **o** Numbers of total cells (**n**), and macrophages (**o**) in BALF from Flu-infected WT mice given PBS ($n = 4$) or 32 μg GSSSG ($n = 4$). **p** Concentration of IL-6 in BALF from Flu-infected WT mice given PBS ($n = 4$) or 32 μg GSSSG ($n = 4$). Data are means ± s.d. $^*P < 0.05$, $^{**}P < 0.01$, $^{***}P < 0.001$, $^{****}P < 0.0001$. Source data are provided as a Source Data file.

translational (protein synthesis) activity remained unaffected as revealed by mitochondrial protein immunoblotting (Supplementary Fig. 7c, d).

These ACE2 and AINK (CPERS-deficient) double-mutant ($Cars2^{AINK/+}$::ACE2-Tg) mice were developed to produce SARS-CoV-2 infection in mice, because the wild-type mice that do not possess ACE2 originally had to be genetically engineered to express ACE2 so that they can be fully susceptible for SARS-CoV-2 in the airway. The results showed that the lethal effect of intranasal infection with SARS-CoV-2 was significantly higher in ACE2-AINK-mutant mice than in the ACE2-transgenic mice (Fig. 3b–d). The data thus suggest that persulphides or related supersulphides generated by CARS2/CPERS contribute to the anti-SARS-CoV-2 host defense. This interpretation is further supported by our supersulphur metabolome, which shows that the supersulphide production of ACE2-AINK mutant mice was decreased compared with that of WT mice during SARS-CoV-2 infection (Supplementary Figs. 7e and 8a); although the magnitude of reduced supersulphides is somewhat smaller in AINK mice (be it noted that only heterozygotes are available) than in $Cars2^{+/-}$ mice.

In addition, the COVID-19 model was produced in hamsters (Fig. 3e–g), which was applied to the pharmacological study to explore the antiviral effect of GSSSG and was also used to develop the breath analysis to experimentally identify biomarkers of COVID-19, as described below. We thus examined the effect of GSSSG pharmacologically administered intraperitoneally (i.p.) on the experimental viral pneumonia in hamsters, which was induced by the intratracheal inoculation of SARS-CoV-2. GSSSG treatment clearly resulted in the beneficial consequences, including reduced viral propagation and ameliorated pneumonia pathology as manifested by the body weight loss and pulmonary consolidation (Fig. 3e–g and Supplementary Fig. 8b).

## The antiviral mechanisms of supersulphides

Our present study shows the potent anti-coronavirus activities of supersulphides such as GSSSG and inorganic polysulphides (e.g., $Na_2S_2$, $Na_2S_3$, $Na_2S_4$), but not for the $H_2S$ donor ($Na_2S$). The suppression of the viral replication is most likely due to inhibition of the viral thiol proteases by supersulphides, as well as through conformational alterations of the coronavirus structural protein, i.e., spike glycoprotein (S protein), by supersulphide-induced cleavage of their disulphide formation (Fig. 4 and Supplementary Figs. 9–13). A similar antiviral mechanism via the disulphide cleavage at the identical site located in the receptor binding domain (RBD) of S protein was proposed recently[31].

The other target is the viral protease to be affected by supersulphides as well (Fig. 4). SARS-CoV-2 expresses two different thiol proteases−papain-like protease ($PL^{pro}$) and 3CL or main protease ($3CL^{pro}$)−that are essential for intracellular virus replication. Supersulphides reportedly react efficiently with protein thiols in a manner that depends on the redox status of the thiol moieties[32]. In our studies here, GSSSG strongly inhibited both $PL^{pro}$ and $3CL^{pro}$ in cell-free reaction systems of each recombinant protein (Fig. 4 and Supplementary

Figs. 9–12), with this result being supported by the docking model of GSSSG with $PL^{pro}$ and $3CL^{pro}$ (Fig. 4b). The crystal structures of proteases in SARS-CoV-2 were obtained from the Protein Data Bank. PDB ID for $PL^{pro}$ is 6W9C and $3CL^{pro}$ is 6Y2E. The $PL^{pro}$ active site contains a canonical cysteine protease catalytic triad (C111, H272, D286), while $3LC^{pro}$ has catalytic dyad (H41, C145). The docking model predicted that GSSSG is located close proximity to the cysteine residues at the triad of $PL^{pro}$ and dyad of $3CL^{pro}$. This result indicates that the possible GSSSG reaction with cysteine in the active site of these proteases, which may have resulted in inhibition of protease activities. The inhibitory effect of GSSSG on the coronaviral protease was also evident for MHV (Supplementary Fig. 11b), which may support the suppression of MHV replication by GSSSG in cells motioned above. As Supplementary Fig. 13b shows the chemical reaction (binding) mechanism, GSSSG is assumed to react with active center cysteine thiols to produce glutathionylation and perthioglutathionylation adducts that actually abolish the catalytic activities of these thiol proteases. Intriguingly, such an inhibitory effect was recovered only partially by the reductive treatment with tris(2-carboxyethyl)phosphine (TCEP) and dithiothreitol (DTT); whereas the same inhibitory effects on other thiol proteases like papain and cathepsin B were almost completely reversed. Not only GSSSG but also inorganic polysulphides were found to have strong viral protease-inhibitory activities, which were not fully reactivated by the reductive modification. It is therefore interpreted that the supersulphide-dependent antiviral protease effect may be caused by $CysS_nSH$ formation at their active sites, as schematized in Supplementary Fig. 13b, which was indeed verified by our proteome analyses shown in Fig. 4e, f and Supplementary Figs. 9 and 10. It may also deserve special emphasis that the effective concentrations of exogenous supersulphides (GSSSG and $H_2S_n$, $n > 1$) that affected SARS-CoV-2 and influenza virus infectivities (Figs. 1e and 2b, e, f and Supplementary Figs. 3b and 4a–c, f, i) are consistent with that of endogenous supersulphides formed in the host cells (Supplementary Figs. 5 and 6).

Earlier work described the anti-coronavirus activity of nitric oxide (NO)[33], with this finding likely supported by our present data showing a strong enhancing effect of NO on GSSSG-mediated inhibition of coronavirus $PL^{pro}$ (Supplementary Figs. 11c–e). These data indicate that the target thiol residues of $PL^{pro}$ may become more susceptible to supersulphides when the NO-induced oxidative effect is added to the effect of supersulphides on the $PL^{pro}$ thiol residues.

## The breath omics for viral infection and sulphur metabolome

The breath analysis was conducted with human subjects infected with SARS-CoV-2, in the context of the protective effect of supersulphides against COVID-19 (Fig. 5). We thus found that enhanced production of various supersulphide metabolites [e.g., persulphides and thiosulphate ($HS_2O_3^-$)] was evident in the exhaled breath condensate (EBC) of patients with moderate COVID-19, as revealed by results of breath sulphur-omics technology that we developed. Of great importance is a marked increase in levels of sulphur metabolites, particularly HSSH ($H_2S_2$), HSSSH ($H_2S_3$), sulphite ($HSO_3^-$), and $HS_2O_3^-$, in the EBC of a

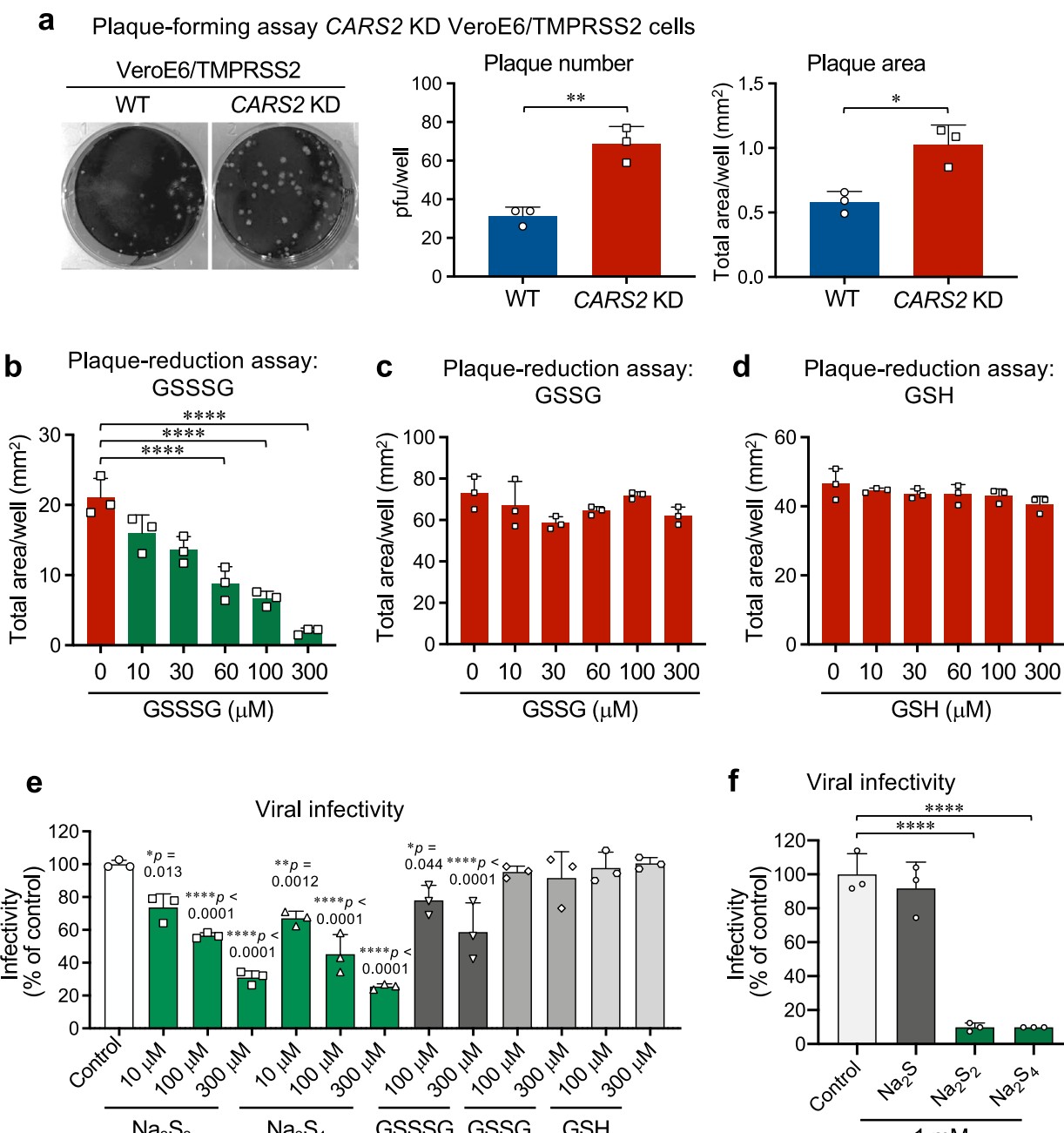

**Fig. 2 | Anti-SARS-CoV-2 effects of supersulphides in VeroE6/TMPRSS2 cells with or without *CARS2* KD. a** Plaque assay for SARS-CoV-2 with control (VeroE6/TMPRSS2 cells) and their *CARS2* KD cells. Left, middle, and right panels show representative plaque formation, plaque number, and plaque area, respectively. $P = 0.0031$ and 0.012. **b–d** Plaque-reduction assay with GSSSG (**b**), GSSG (**c**), and GSH (**d**) for SARS-CoV-2 using *CARS2* KD VeroE6/TMPRSS2 cells. The plaque-reduction efficacy was assessed in terms of the effect of various compounds on the plaque formation by measuring total areas of plaques ($mm^2$) observed in each well.

The related data for the plaque-reduction assay with control VeroE6/TMPRSS2 cells are shown in Supplementary Fig. 4f–h. All $P < 0.0001$. **e**, **f** Suppression of SARS-CoV-2 infectivity by supersulphides. SARS-CoV-2 was incubated with $Na_2S_2$, $Na_2S_4$, GSSSG, GSSG, or GSH at indicated concentrations (**e**), and with 1 mM $Na_2S$, $Na_2S_2$, or $Na_2S_4$ (**f**), at 37 °C for 30 min. The viral infectivity was determined by the plaque-forming assay. $P$ values are described in (**e**). $P < 0.0001$ in both in **f**. Data are means ± s.d. ($n = 3$). $^*P < 0.05$, $^{**}P < 0.01$, $^{***}P < 0.001$, $^{****}P < 0.0001$. Source data are provided as a Source Data file.

patient (as indicated by arrows), just before manifesting a progressive disease caused by severe viral pneumonia (Fig. 5a and Supplementary Fig. 14a). Of note, the increase of breath sulphur metabolites was replicated in a hamster model of COVID-19, as demonstrated below. Meanwhile, the COVID-19 patients, who were enrolled in this clinical investigation to provide EBC, had somehow distinct backgrounds or underlying diseases; for example, diabetes mellitus, and otherwise different ill conditions with or without requirement of hospitalization during the coronavirus infection. Most patients suffered in severity

from mild to moderate infections, without fatal lung damage or dysfunction, but with mild pneumonia and other moderate disorders or complications. We compared the abundance of various sulphur metabolites and supersulphides among the SARS-CoV-2-infected subjects stratified according to the severity of symptoms, underlying diseases, need for hospitalization, length of hospital stay, and so on. While each EBC value of supersulphides is significantly higher in the COVID-19 individuals than in healthy controls, there is no statistical difference in the striated comparison, as long as we examined the

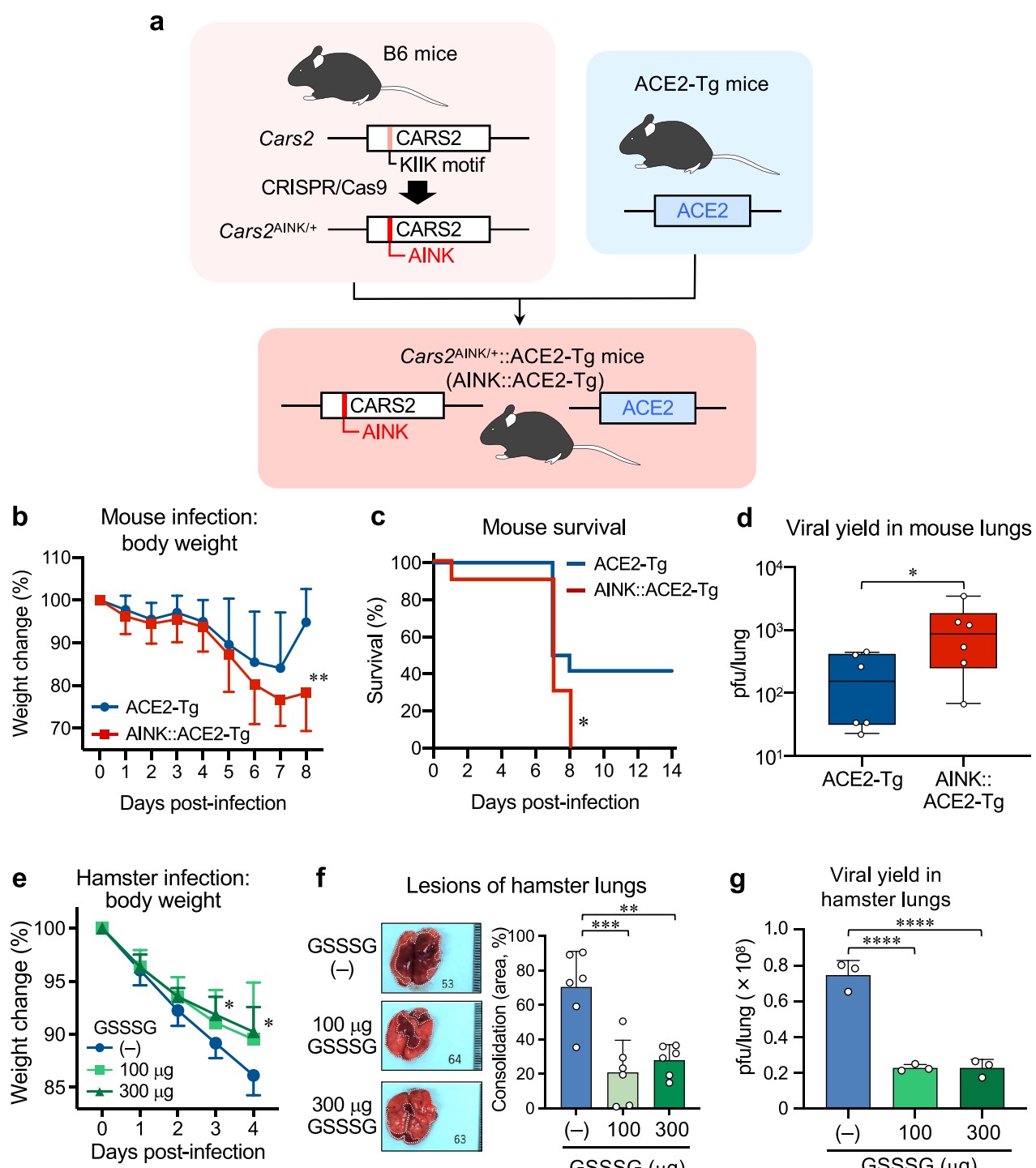

**Fig. 3 | Anti-COVID-19 effects of supersulphides in ACE2-Tg mice and hamsters.** **a** Schematic drawing of the method for generation of ACE2-Tg::CPERS-deficient (AINK::ACE2-Tg) mice. *Cars2*[AINK/+] mice were produced via CRISPR-Cas9 system and crossed with ACE2-Tg mice to generate AINK::ACE2-Tg mice. Supplementary Fig. 7 shows genomic modification at *Cars2* locus by the CRISPR-Cas9 system. **b–d** SARS-CoV-2 infection in mice. **b**, Mean body weight changes (95%, CI) of SARS-CoV-2-infected ACE2-Tg ($n = 12$) and AINK::ACE2-Tg mice ($n = 10$). All mice were infected intratracheally (i.t.) with 100 pfu (per 50 µl) of SARS-CoV-2; body weight was monitored until day 8 post-infection. $P = 0.0045$. **c** Kaplan–Meier survival curves for ACE2-Tg ($n = 12$) and AINK::ACE2-Tg mice ($n = 10$) infected with SARS-CoV-2; survival was monitored until day 14 post-infection. $P = 0.035$. **d** Viral yield in the infected lungs (homogenates) at 4 days post-infection. The viral titers were determined by plaque-forming assay and are expressed as a plaque-forming unit

(pfu)/lung. $P = 0.048$. **e–g** SARS-CoV-2 infection in hamsters. Hamsters were infected i.t. with $6 \times 10^6$ pfu (120 µl) of SARS-CoV-2; with simultaneous i.t. administration of 100 or 300 µg GSSSG. The hamsters were subsequently administered intraperitoneally (i.p.) 500 or 1000 µg GSSSG daily from day 1 to day 4. **e** Mean body weight changes (95% CI) in SARS-CoV-2-infected hamster that were given PBS ($n = 6$) or GSSSG ($n = 6$), as being monitored until day 4 post-infection. $P = 0.035$ (3 days) and 0.020 (4 days). **f** Semi-quantitative measurement of the area of consolidation (pathological lesion) of SARS-CoV-2-infected hamster lungs with or without GSSSG treatment ($n = 6$). $P = 0.0004$ and $0.0015$. **g** The amounts (pfu) of virus yielded in the lungs of hamsters treated or untreated with GSSSG after infection, as assessed by the plaque-forming assay at 4 days post-infection. $P < 0.0001$ in both. Data are means ± s.d. $^*P < 0.05$, $^{**}P < 0.01$, $^{***}P < 0.001$, $^{****}P < 0.0001$. Source data are provided as a Source Data file.

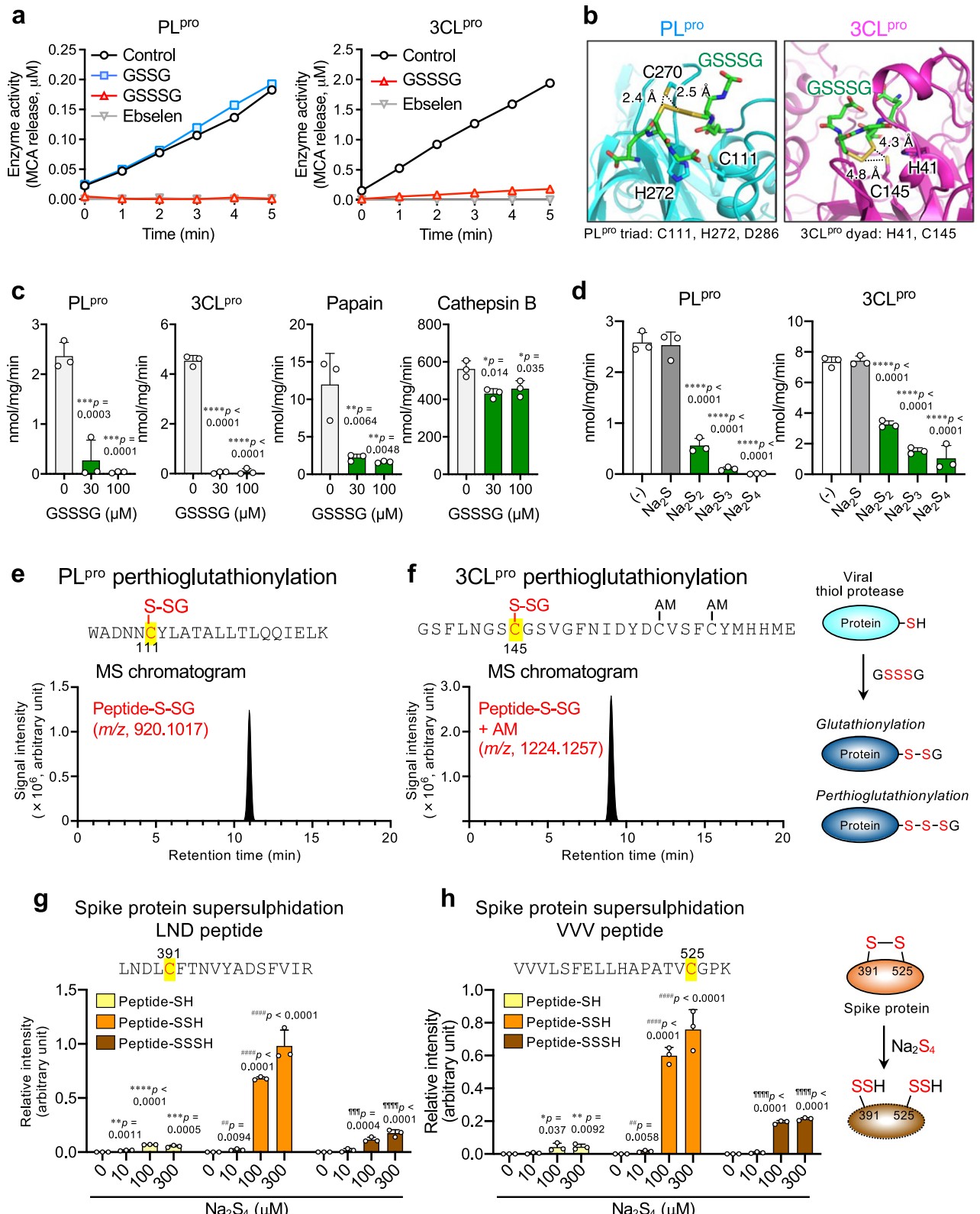

patients in view of varied degrees of severe illness or long-term hospitalization.

The hamster breath analysis that we developed herein (Fig. 5b and Supplementary Fig. 14b–d) thus revealed that various sulphur metabolites are remarkably elevated in the course of SARS-CoV-2 infection of the lungs. As the sulphur metabolites increase over time after the infection, body weight loss and pulmonary consolidation (Fig. 5c–f and Supplementary Fig. 14e). Also, it should be noted that the profile of the sulphur metabolites detected with the hamster EBC was almost consistent with that of human EBC from the COVID-19 patients. Moreover, considerable amounts of $HSO_3^-$, and $HS_2O_3^-$ in EBC were observed not only with human subjects but also the hamster model of COVID-19. Collectively, the sulphur-omics established herein from human and animal studies thus indicates the potential involvement of

**Fig. 4 | Anti-SARS-CoV-2 activity of supersulphides: impairment of viral thiol proteases and S protein integrity. a** Protease activity (SARS-CoV-2 PL$^{pro}$ and 3CL$^{pro}$, 1 μM each) was measured after treatment of proteases with 300 μM each GSSG, GSSSG, or ebselen. **b** Docking models of PL$^{pro}$ triad (left panel) and 3CL$^{pro}$ dyad (right panel) bound with GSSSG obtained by using SwissDock Yellow, blue, red, and green indicate sulphur, nitrogen, oxygen, and carbon atoms, respectively. **c** Inhibitory effects of GSSSG on PL$^{pro}$, 3CL$^{pro}$, papain, and cathepsin B (1 μM each). *$P < 0.05$, ***$P < 0.001$ vs. 0 μM GSSSG. $P$ values are described in (**c**). **d** Inhibitory effects of Na$_2$S, Na$_2$S$_2$, Na$_2$S$_3$, and Na$_2$S$_4$ (3 μM each) on PL$^{pro}$ and 3CL$^{pro}$ (1 μM each). ***$P < 0.001$ vs. GSSSG (−). $P$ values are described in (**d**). **e, f** Proteome analysis of GSSSG treated PL$^{pro}$ (**e**) and 3CL$^{pro}$ (**f**). LC-Q-TOF-MS chromatograms obtained from proteome analysis of PL$^{pro}$ and 3CL$^{pro}$: peptide fragments containing cysteine residues C111 and C145 are shown. Cys modification identified in the peptide fragments are shown in the panel headlines. Perthioglutathionylation (GSSH adducts) of cysteine residues in PL$^{pro}$ and 3CL$^{pro}$ were detected by monitoring at $m/z$ 920.1017 and 1224.1257, respectively: corresponding MS/MS spectra are illustrated in Supplementary Fig. 9. The GSSSG reaction with thiol proteases is schematized in the right panel. **g, h** Proteome of SARS-CoV-2 S protein, which includes each cysteine residue (C391, **g** and C525, **h**) identified via the LC-Q-TOF-MS. S proteins that were reacted with Na$_2$S$_4$ (0–300 μM), were subjected to the MS analysis (see Supplementary Fig. 10 for the spectra). The intensity of each peptide with Cys-modified was normalized to the 906-FNGIGVTQNVLYENQK-921 of S protein. Each signal intensity is shown as a relative value compared to the total intensity of corresponding fragments derived from DTT-treated S protein. The reaction of Na$_2$S$_4$ with S protein is schematized in the right panel. *$P < 0.05$, **$P < 0.01$, ***$P < 0.001$ vs. Peptide-SH/0 μM Na$_2$S$_4$; ##$P < 0.01$, ###$P < 0.001$ vs. Peptide-SSH/0 μM Na$_2$S$_4$; ¶¶¶$P < 0.001$ vs. Peptide-SSSH/0 μM Na$_2$S$_4$. $P$ values are described in (**g, h**). Data are means ± s.d. ($n = 3$). Source data are provided as a Source Data file.

supersulphides in anti-COVID-19 host defense as well as a possible application of supersulphide metabolites exhaled in the breath air as potential biomarkers for the coronavirus or other airborne infections emerging and re-emerging in the future.

## Supersulphide protection in COPD models

COPD is one of the high-risk medical conditions ever documented, for which disease control and precautions are now taken to prevent mortality[34]. Suppressed endogenous supersulphide formation in COPD lungs, which was reported earlier[24], is now known to be caused by markedly reduced protein expression of CARS2/CPERS in primary cultured cells obtained from COPD airways, as we demonstrated here (Fig. 6a and Supplementary Fig. 15a–c). The impaired CARS2 expression in primary airway cells correlated well with the degree of worsened airway resistance and with inflammatory alveolar destruction, as well as with reduced mitochondrial membrane potential, as observed in COPD (Fig. 6 and Supplementary Fig. 15). We observed a similar reduction in supersulphide production in the mouse model of COPD (see Fig. 7a and Supplementary Fig. 16). Therefore, supersulphides may have a potent protective function in COPD.

To clarify the beneficial roles of supersulphides in COPD, we investigated the effects of supersulphide deficiency on several mouse models of COPD, which were generated by using porcine pancreatic elastase (PPE) or cigarette smoke extract (CSExt) instilled intratracheally, as well as on physiological ageing of mouse lung. We used Cars2$^{+/−}$ and Cars2$^{AINK/+}$ mice and their WT littermates to analyze the effects of supersulphides on COPD pathogenesis. Cars2$^{+/−}$ mice demonstrated much more severe COPD pathology compared with WT mice; Fig. 6c–k shows markedly enhanced emphysematous changes, which were evidenced by airspace enlargement as measured in 3D morphometric images via micro-computed tomography (micro-CT) and histopathological examinations (e.g., mean linear intercept). Additional evidence included increased ratios of low-attenuation volume to total lung volume (LAV/TLV), worsened airway obstruction as assessed by forced expiratory volume in 0.1 second per forced vital capacity (FEV$_{0.1}$/FVC) estimated by using the flow-volume curve, and elevated lung elasticity (i.e., static compliance) as determined from the pressure–volume curve measured via the flexiVent (SCIREQ) system[35,36]. Enhanced inflammatory responses and lung damage were also observed in Cars2$^{+/−}$ mice, which were identified by using various parameters such as inflammatory cell infiltration, cytokine induction, oxidative stress, apoptosis, and matrix metalloproteinase (MMP) activation in the lung (Fig. 6l and Supplementary Fig. 15f–n). Also, supersulphide production in the lung showed a uniquely altered formation profile of different sulphide species (Fig. 7a and Supplementary Fig. 16). That is, the amounts of various reduced superpersulphides including CysSSH, GSSH, and hydrogen persulphide (HSS$^−$) were markedly decreased in both WT and Cars2$^{+/−}$ mice during the progression of disease after elastase administration. Levels of all supersulphides were much lower in Cars2$^{+/−}$ mice than in WT mice, however, and the levels of only one oxidized persulphide, i.e., GSSSG, were inversely related (increased) in a time-dependent fashion, although the simple disulphide GSSG was not appreciably affected in the COPD model (Fig. 7a and Supplementary Fig. 16).

Furthermore, we studied the COPD phenotype of Cars2$^{AINK/+}$ mice by analyzing emphysema progression and airway inflammation induced by elastase. COPD and emphysematous pathology and airway inflammation were significantly exacerbated in Cars2$^{AINK/+}$ mice compared with WT littermates (Fig. 7b–f). For example, markedly increased airspace enlargement and airflow obstruction were found in Cars2$^{AINK/+}$ mice, as identified by LAV/TLV and FEV$_{0.1}$/FVC, respectively. We thus unequivocally confirmed that supersulphides produced solely by CARS2/CPERS greatly contribute to regulation and protection of COPD pathogenesis through the strong anti-inflammatory and antioxidant effects of supersulphides.

## Supersulphides in lung premature ageing

All data described above together indicate that supersulphides should have a significant effect on protection of airways from COPD pathogenesis. In fact, we found the same exacerbation of COPD pathology in Cars2$^{+/−}$ mice treated with CSExt (Fig. 8a–c), in which both inflammatory responses and cellular senescence were significantly increased (Fig. 8d, e). In view of this beneficial effect of supersulphides, therefore, that the reduced intracellular supersulphide levels enhanced cellular senescence in vivo is quite intriguing. Also, when human fetal lung fibroblasts-1 (HFL-1) cells were exposed to CSExt, CARS2 expression and supersulphide production were significantly reduced (Supplementary Fig. 17a, b), which correlated well with increased p53 and p21 expression as well as cellular senescence as assessed by formation of senescence-associated β-galactosidase (SA-β-gal)-positive cells induced by CSExt (Supplementary Fig. 17c–e). Consistent with the induction of cellular senescence, the production of cytokines [e.g., interleukin (IL)-6 and IL-8] and the degree of gel contraction and fibronectin production in the floating medium were significantly lower after CSExt exposure (Supplementary Fig. 17f–h). Moreover, when we knocked down CARS2 with siRNA, CARS2 expression decreased by 90% (Supplementary Fig. 18e). Supersulphide amounts were also reduced by ~80% by CARS2 knockdown compared with control cells (Supplementary Fig. 18a). CARS2 knockdown in cells significantly enhanced cellular senescence as judged from the increase in numbers of SA-β-gal-positive cells and levels of p53 and p21 proteins (Supplementary Fig. 18b, e). The fibroblast-mediated tissue repair function was also impaired as determined by the collagen contraction assay (Supplementary Fig. 18c, d). These data suggest that CSExt-induced CARS2 deficiency promotes cellular senescence and impairs tissue repair.

To investigate how the Cars2 deficiency affects the physiological ageing of the lung, Cars2$^{+/−}$ and WT mice at 88 weeks of age were killed and whole lungs were obtained. Expression of p53 and

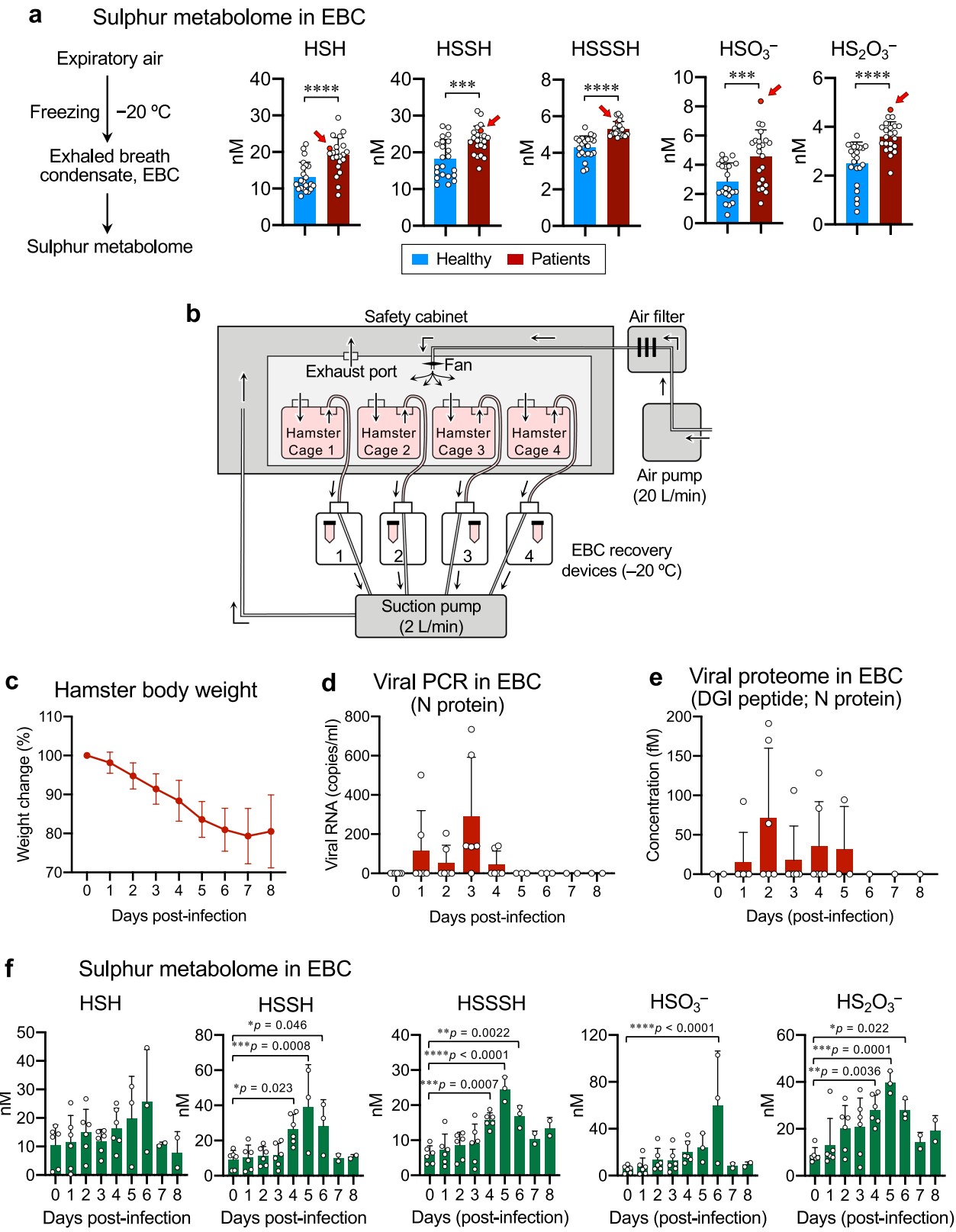

**a** Sulphur metabolome in EBC

**b**

**c** Hamster body weight

**d** Viral PCR in EBC (N protein)

**e** Viral proteome in EBC (DGI peptide; N protein)

**f** Sulphur metabolome in EBC

p21 was significantly higher in lungs of $Cars2^{+/-}$ mice than in lungs of WT mice (Fig. 8f). Immunohistochemical expression of p21 in lungs was also elevated in $Cars2^{+/-}$ mice compared with that in WT mice (Fig. 8g). In addition, development of emphysema was enhanced in lungs in $Cars2^{+/-}$ mice compared with that in WT mouse lungs, as confirmed by measuring the mean linear intercept (Fig. 8h) and LAV/TLV determined by means of a 3D micro-CT morphometric image (Fig. 8i). These data thus may suggest that supersulphides produced by CARS2 could limit the progression of cellular senescence and prevent lung ageing by virtue of their antioxidant function and contribution to mitochondrial energy metabolism, which is supported by our recent studies showing that sulphur respiration is effectively taking place in the mammalian mitochondria[26,37,38].

**Fig. 5 | Breath omics analyses in human and hamster. a** Sulphur metabolome of EBC of healthy subjects and patients with COVID-19. EBC were quickly collected by freezing expired air at −20 °C; levels of sulphur metabolites including HSH ($H_2S$, $S^{2-}$), HSSH ($H_2S_2$, $S_2^{2-}$), HSSSH ($H_2S_3$, $S_3^{2-}$), $HSO_3^-$, and $HS_2O_3^-$ were quantified by using LC–MS/MS analysis after β-(4-hydroxyphenyl)ethyl iodoacetamide (HPE-IAM) labeling. Red arrows show higher levels of sulphur metabolites for a patient with exacerbated COVID-19. Each dot represents data from healthy subjects and COVID patients ($n = 22$ per group). $P < 0.0001$ (HSH), $=0.0006$ (HSSH), $<0.0001$ (HSSSH), $= 0.0009$ ($HSO_3^-$), and $<0.0001$ ($HS_2O_3^-$). **b** Non-invasive hamster EBC collection system. We developed here a hamster model especially for the breath

analysis to experimentally identify biomarkers of COVID-19. **c** Mean body weight changes (95% CI) in SARS-CoV-2-infected WT hamster ($n = 6$). All hamster were intratracheally infected with $6 \times 10^6$ pfu (per 100 μl) of SARS-CoV-2; survival was monitored until day 8 post-infection. **d** Copy numbers of SARS-CoV-2 N protein-encoded RNA analyzed by quantitative RT-PCR with EBC recovered from hamsters for 8 days after infection. **e** Viral proteome analysis with EBC of infected hamster until day 8 post-infection as performed vis LC-ESI-MS/MS. **f** Sulphur metabolome conducted with the EBC obtained from hamsters for 8 days after SARS-CoV-2 infection. $P$ values are described in (**f**). Data are means ± s.d. $^*P < 0.05$, $^{**}P < 0.01$, $^{***}P < 0.001$, $^{****}P < 0.0001$. Source data are provided as a Source Data file.

## Supersulphide therapeutics in COPD

Because endogenous production of supersulphides was markedly reduced in COPD and in animal models of COPD induced by elastase and CSExt, we investigated the airway-protective effects of super-sulphides in COPD pathogenesis by using GSSSG. GSSSG treatment significantly reduced COPD in the elastase-induced COPD model (Fig. 9a–d). For example, GSSSG treatment significantly improved the airspace enlargement (Fig. 9a, b) and the reduced FEV$_{0.1}$/FVC (Fig. 9c) in Cars2$^{+/-}$ mice. Similarly, the same GSSSG treatment reduced CSExt-induced emphysema and its associated cellular senescence in lungs of both WT and Cars2$^{+/-}$ mice (Fig. 9e–g). GSSG had no effect in this COPD model (Fig. 9h). These findings indicated that exogenous administration of supersulphides may have therapeutic potential in COPD by reducing airway inflammation and premature senescence.

## IPF aggravation by reduced supersulphide

We also studied whether supersulphides have any protective effects in the IPF model produced by bleomycin in mice[39]. Cars2$^{+/-}$ mice treated with bleomycin, compared with WT mice treated with bleomycin, showed increased body weight loss and worsened lung fibrosis as evaluated by 3D morphometric analysis with micro-CT (Fig. 10a–c). Lung inflammation and lung fibrosis were also significantly enhanced in Cars2$^{+/-}$ mice versus WT mice, as assessed by the amounts of IL-1β, IL-6, monocyte chemoattractant protein-1 (MCP-1), tumor necrosis factor-α (TNF-α), and keratinocyte chemoattractant (KC) detected in bronchoalveolar lavage fluid (BALF) (Fig. 10d–g). The numbers of terminal deoxynucleotidyl transferase dUTP nick end labeling (TUNEL)-positive cells were also significantly greater in Cars2$^{+/-}$ mice treated with bleomycin than in bleomycin-treated WT mice, which indicates increased numbers of apoptotic cells in Cars2$^{+/-}$ mice (Fig. 10g). This study thus suggests a potent protective function of supersulphides in IPF pathogenesis.

## Discussion

Our current study is the first demonstration of the viral infection- and lung disease-prone nature of two different types of CARS2/CPERS mutant mice. We thus showed that CARS2/CPERS is the major contributor to supersulphide biosynthesis in the lungs, and has host defense and protective functions in respiratory viral infections and chronic lung diseases such as COPD and IPF. As a notable finding, supersulphides, when administered exogenously as synthetic GSSSG, demonstrated not only an appreciable antiviral effect but also significant improvements in the COPD and IPF models, which may warrant their translational application to various viral and inflammatory lung diseases. It is well documented recently that COPD is the highest risk condition to worsen the COVID-19 severity[34]; and IPF is one of the most serious complications reported earlier[40]. Because our current studies suggest that supersulphide such as GSSSG may ameliorate COPD and IPF, as well as viral lung diseases (e.g., influenza and COVID-19), GSSSG may be a good therapeutic approach for COVID-19 treatment.

In view of the redox-active property of supersulphides, Fukuto et al. previously described persulphidation of cysteinyl thiol-dependent enzymes, which occurs most typically with the thiol protease papain

and is inactivated by supersulphide modification of active center thiols[32]. In the same context of protein sulphur modification, activities of SARS-CoV-2 PL$^{pro}$ and 3CL$^{pro}$ were easily abolished via persulphidation induced by GSSSG treatment, thus indicating the strong molecular basis of the antiviral activity of GSSSG as verified by our cell culture study of SARS-CoV-2 replication reported here. In addition, NO may have similar antiviral activity, because of the putative inhibitory effect on viral proteases, possibly through its nitrative and oxidative modifications of protein thiols reported earlier[33]. NO, however, is too inert to cause any biologically relevant thiol chemical modification and prevent the protease activity that depends on thiol-mediated catalysis and is susceptible only to supersulphides, as we confirmed with two SARS-CoV-2 proteases (PL$^{pro}$ and 3CL$^{pro}$). In fact, we found a rather synergistic effect of NO and supersulphides in terms of PL$^{pro}$ inhibition. Because supersulphides reportedly reacted efficiently with protein thiols in a manner that depended on the redox status of the thiol moieties[32], NO-induced chemical modification of thiols may allow supersulphides to change the structure of protease thiols more extensively and eliminate their catalytic activity accordingly. Alternatively, protease thiols may be affected by some reactive derivatives such as nitrosopersulphide, possibly formed via reaction of NO and supersulphides, which are responsible for diverse physiological functions of NO rather than NO per se, as was proposed earlier[41,42]. In fact, previous reports showed some potential interaction of sulphide and NO that effectively produce nitrosopersulphide, including various polysulphides, S-nitrosothiol (HSNO), and S-nitrosopersulphide (HSSNO)[43-45].

There are previous papers proposing the potential protective functions of sulphide against COVID-19[46,47]. Earlier reports only dealt with a possible beneficial effect of hydrogen sulphide on the COVID-19 pathogenesis, where sulphide could be either introduced pharmacologically with sulphur-containing compounds such as diaryltrisulphide and $S_8$ or endogenously generated during the host defense responses. The cytoprotective functions of various sulphide-related compounds are also described lately[47]. However, until our current work, there has been no rigorous study demonstrating the direct contribution of supersulphides to host defense for anti-influenza virus and anti-SARS-CoV-2 effects and in particular their airway-protective effects in vitro and in vivo.

The direct antiviral effect may be caused by supersulphides via the structural alteration of viral proteins such as influenza virus HA that is a major envelope protein and is thereby most responsible for viral attachment and entering into host cells via its membrane fusion activity. As described above, influenza virus was indeed susceptible to be inactivated by supersulphides (Fig. 1e), in which several disulphide bridges of viral HA polypeptides are likely to be conformationally distorted by $Na_2S_{2-4}$ (inorganic hydropolysulphides) in the same manner as that of coronavirus. Thus, the disulphide bond formation in the viral structural proteins (e.g., S protein and HA) might be a key target that determines the viral susceptibility to supersulphides, especially inorganic hydropolysulphides (Supplementary Fig. 13a); and besides, it could be an Achilles heel commonly expressed among all viruses including SARS-CoV-2 and influenza virus. This finding may have important implications for the potential innate antiviral

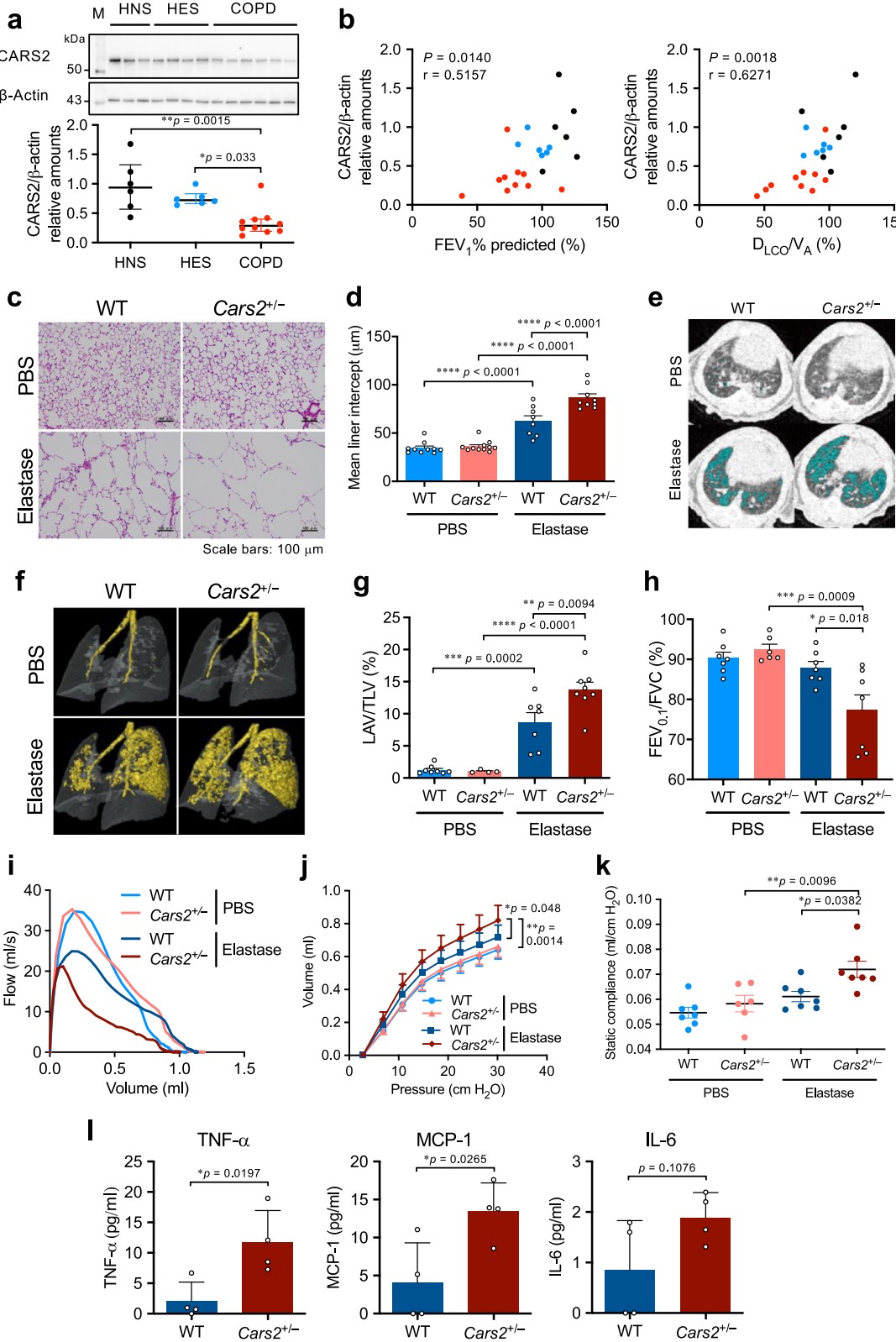

consequences of supersulphides, which we observed here in a similar fashion against influenza virus and SARS-CoV-2, because both viral infections reportedly manifested the same pathogenesis and thereby clinical outcome or in-hospital mortality[48,49].

Although certain modulatory effects of $H_2S$ on immune responses have been reported[50,51], the molecular mechanisms of the immune regulation remain unclear. We previously demonstrated, by using the

supersulphide donor *N*-acetylcysteine (NAC), that polysulphides negatively regulated innate immune responses augmented by lipo-polysaccharide in vitro and in vivo[52]. NAC polysulphides strongly suppressed cytokine-mediated inflammatory responses, such as expression of inducible nitric oxide synthase, through an interferon-β-mediated signaling pathway; these inflammatory responses are highly sensitive to supersulphides and are thus attenuated by NAC

**Fig. 6 | Attenuated CARS2 expression in COPD and enhancement of COPD pathology in elastase-induced COPD model in *Cars2*[+/−] mice. a** Expression of CARS2 in primary bronchial epithelial cells from patients with COPD ($n = 10$), healthy never-smokers (HNS, $n = 6$), and healthy ex-smokers (HES, $n = 6$) as determined by western blotting (upper panel) and quantitative results (lower panel). **b** Correlations between the amounts of CARS2 and FEV$_1$% predicted (left panel) or D$_{LCO}$/V$_A$ predicted (right panel) of study subjects. Black, blue, and red circles indicate HNS, HES, and COPD, respectively. D$_{LCO}$ diffusing capacity for carbon monoxide, VA alveolar volume. **c** Representative HE-stained images showing the extent of airspace enlargement. Scale bars, 100 μm. **d** Quantification of the mean linear intercept in **c**. PBS-treated WT mice ($n = 10$); PBS-treated *Cars2*[+/−] mice ($n = 11$); elastase-treated WT mice ($n = 8$); elastase-treated *Cars2*[+/−] mice ($n = 9$). **e** Representative CT images of a lung of an elastase-induced emphysema model mouse. Blue indicates the low-attenuation area. **f** Representative 3D micro-CT images of lungs of PBS-treated or elastase-induced WT and *Cars2*[+/−] mice. Yellow

indicates the low-attenuation area; the total lung area appears as a transparent shape. **g** The emphysematous area in (**f**) was quantified by calculating low-attenuation volume to total lung volume (LAV/TLV). PBS-treated WT ($n = 8$); PBS-treated *Cars2*[+/−] mice ($n = 4$); elastase-induced WT ($n = 7$); elastase-induced *Cars2*[+/−] mice ($n = 8$). Each dot represents data from an individual mouse ($n = 4$–$8$ per group). **h** Airflow obstruction (FEV$_{0.1}$/FVC) of lungs of elastase-induced WT and *Cars2*[+/−] mice measured by using the flexiVent system on day 21. **i**–**k** Pulmonary function measured on day 21. **i** Representative flow-volume curves. **j** Pressure−volume curves. **k** Static compliance. PBS-treated WT mice ($n = 7$); PBS-treated *Cars2*[+/−] mice ($n = 6$); elastase-treated WT mice ($n = 7$); elastase-treated *Cars2*[+/−] mice ($n = 7$). **l** Concentrations of TNF-α, monocyte chemoattractant protein-1 (MCP-1), and IL−6 in BALF obtained from elastase-induced WT and *Cars2*[+/−] mice on day 5. Data are means ± s.d. *$P < 0.05$, **$P < 0.01$, ***$P < 0.001$, ****$P < 0.0001$. Source data are provided as a Source Data file.

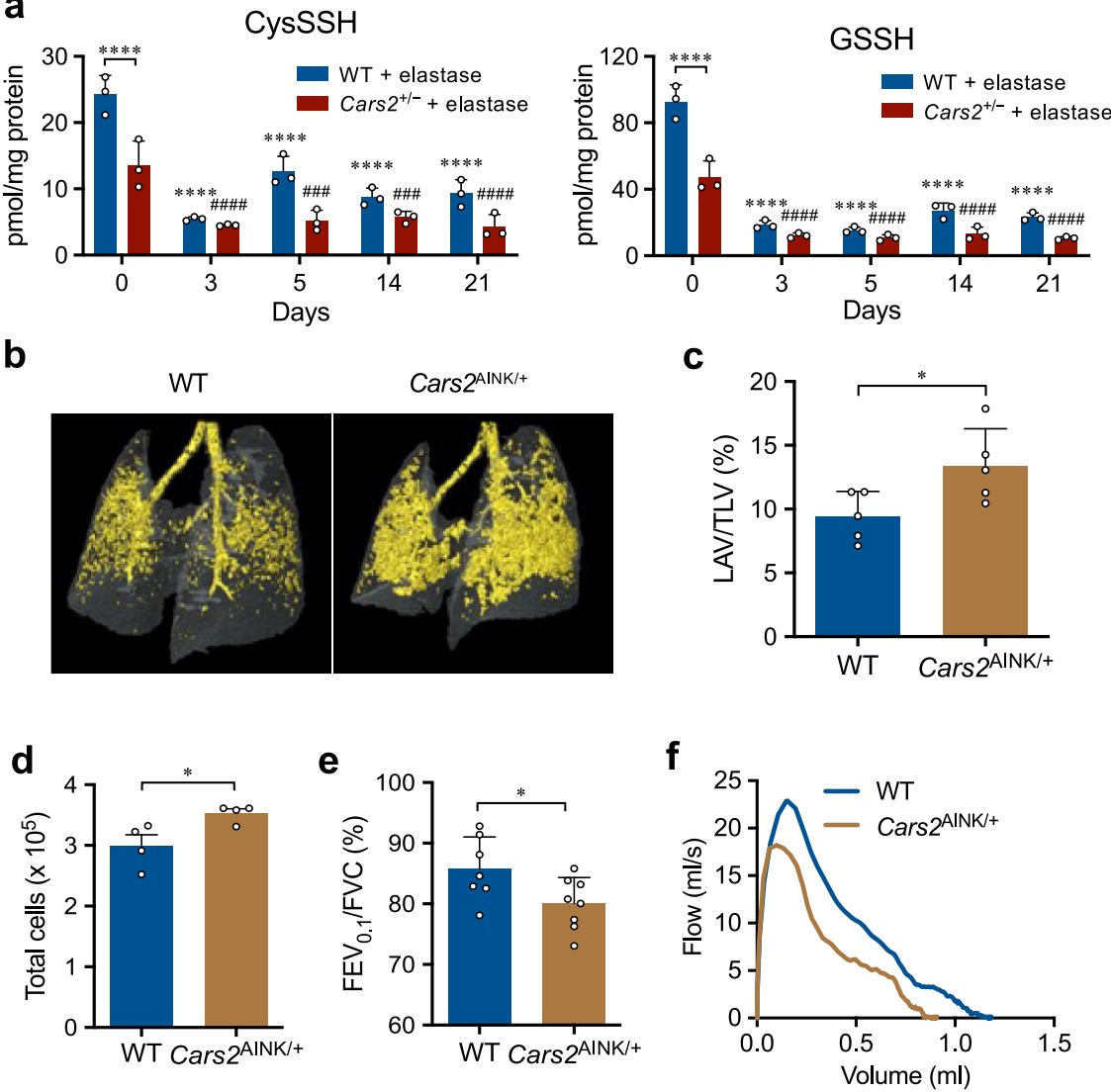

**Fig. 7 | Sulphur metabolome and COPD pathology of elastase-induced COPD model in *Cars2* mutant mice. a** Endogenous levels of GSSH and GSSSG in BALF obtained during 21 days from elastase-induced WT and *Cars2*[+/−] mice, identified via LC−MS/MS analysis with HPE-IAM labeling. All $P < 0.0001$, but ###$P = 0.0002$ (5 days), ###$P = 0.0004$ (14 days) in CysSSH. All $P < 0.0001$ in GSSH. Data are means ± s.d. ****$P < 0.0001$ vs. elastase-treated WT mice/day 0; ###$P < 0.001$, ####$P < 0.0001$ vs. elastase-treated *Cars2*[AINK/+] mice/day 0. **b** Representative 3D micro-CT images of lungs of elastase-treated WT and *Cars2*[AINK/+] mice. Yellow

indicates the low-attenuation area; the total lung area appears as a transparent shape. **c** LAV/TLV for images in (**b**) was quantified by calculating the ratio of LAV to TLV. Elastase-treated WT mice ($n = 5$); elastase-treated *Cars2*[AINK/+] mice ($n = 5$). $P = 0.036$. Data are means ± s.d. *$P < 0.05$. **d** Numbers of total cells in BALF from Flu-infected WT and *Cars2*[AINK/+] mice. $P = 0.034$. Data are means ± s.d. *$P < 0.05$. **e** FEV$_{0.1}$/FVC for WT and *Cars2*[AINK/+] mice. $P = 0.036$. Data are means ± s.d. *$P < 0.05$. **f** Representative flow-volume curves for WT and *Cars2*[AINK/+] mice, as measured by the flexiVent system on day 21. Source data are provided as a Source Data file.

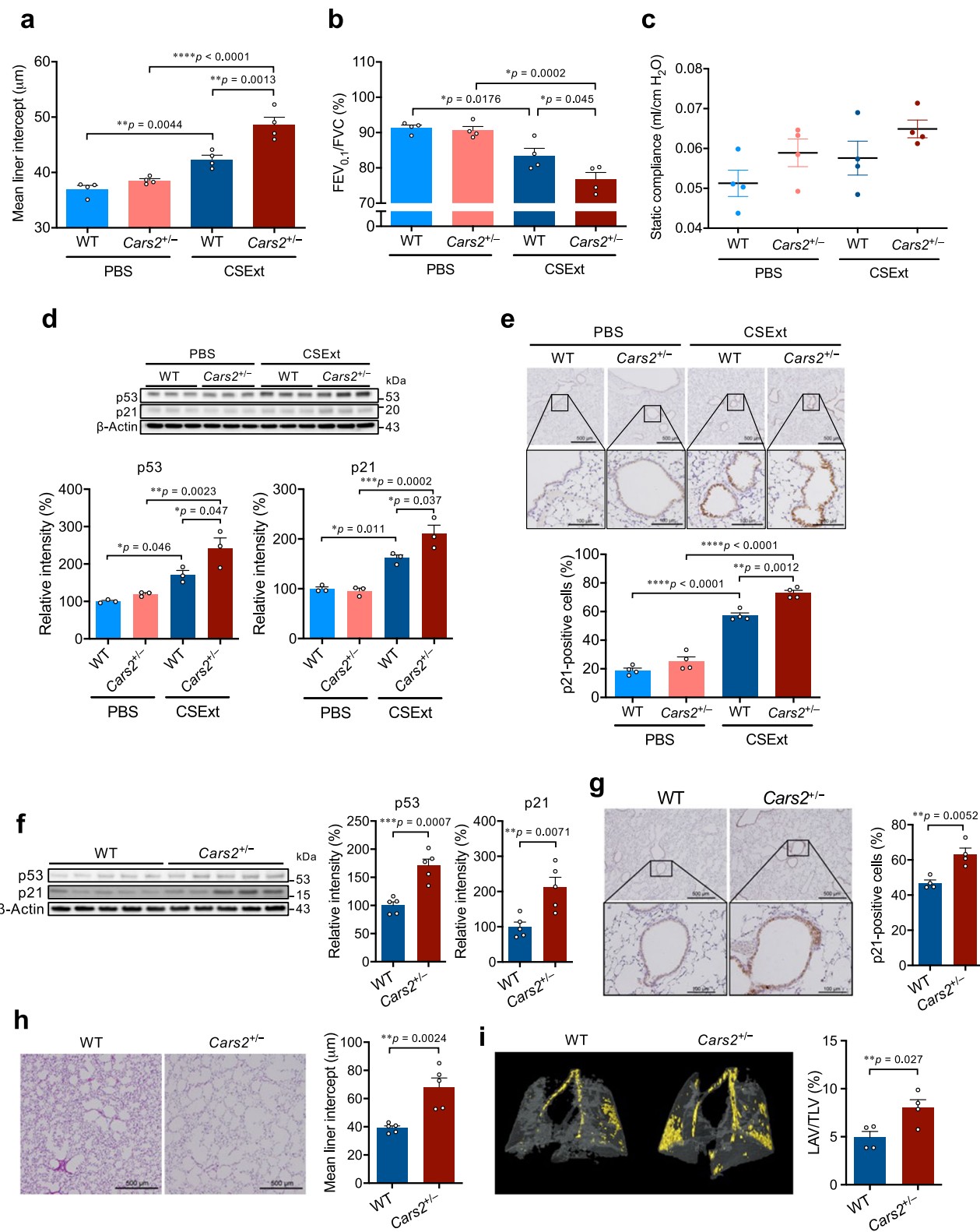

polysulphides[52]. These data are consistent with our GSSSG data in terms of the strong anti-inflammatory effects of GSSSG. Supersulphides may therefore ameliorate not only inflammatory over-reactions occurring during SARS-CoV-2 infections (i.e., the cytokine storm) but also much more specific immune responses and may thereby demonstrate versatile protective functions against viral and inflammatory diseases and may even improve the prognosis of acute viral infections such as COVID-19. COPD, one of the highest risk

conditions reported, is a good therapeutic target for the application of translation of supersulphides.

Our recent study revealed that not only 3-mercaptopyruvate sulfurtransferase (3-MST) KO mice and cystathionine β-synthase (CBS)/cystathionine γ-lyase (CSE)/3-MST triple KO mice, but also single KO mice for either CBS/CSE/3-MST, do not show appreciable changes in several sulphur metabolites[53]. These data indicate that, although three canonical enzymes might be somehow responsible for supersulphide

**Fig. 8 | Enhanced COPD pathology of CSExt-induced COPD model and lung premature ageing in *Cars2*[+/−] mice. a** Quantification of the mean linear intercept for lungs of CSExt-treated COPD model mice (*n* = 4 per group). **b**, **c** Pulmonary function was measured by using the flexiVent system on day 28 after CSExt administration. **b** Airflow obstruction as represented by $FEV_{0.1}$/FVC. **c** Static compliance. PBS-treated WT mice (*n* = 4); PBS-treated *Cars2*[+/−] mice (*n* = 4); CSExt-treated WT mice (*n* = 4); CSExt-treated *Cars2*[+/−] mice (*n* = 4). **d** Representative western blot analysis of p53 and p21 in lungs of WT and *Cars2*[+/−] mice (upper panel) and quantitative results of p53 (bottom left panel) and p21 (bottom right panel) (*n* = 3 for each group). **e** Representative immunohistochemical images showing localization of p21 (upper panels) and quantitative p21-positive cells (lower panels) in lungs of PBS- and CSExt-treated mice. Scale bars, 500 μm (low magnification) and 100 μm (high magnification). **f** p53 and p21 protein levels in whole lungs from aged WT and *Cars2*[+/−] mice (88 weeks old) were measured by using Western blotting (left panel); quantitative results of p53 (middle panel) and p21 (right panel) in whole lungs (*n* = 5). **g** p21 immunostaining in lungs from aged WT and *Cars2*[+/−] mice (88 weeks old) shown as representative images (left panel) and quantitative results (right panel). Scale bars, 500 μm (low magnification) and 100 μm (high magnification). **h** Representative images of airspace enlargement as visualized by HE staining (left panel) and quantification of the mean linear intercept (right panel). Scale bars, 500 μm. **i** Representative 3D micro-CT images of lungs from aged WT and *Cars2*[+/−] mice (88 weeks old) (left panel). Yellow indicates the low-attenuation area; the total lung area appears as a transparent shape (right panel). LAV/TLV was quantified by calculating the ratio of LAV to TLV. Aged WT mice (*n* = 4); aged *Cars2*[+/−] mice (*n* = 4). Data are means ± s.d. *P* < 0.05, **P* < 0.01, ***P* < 0.001. *P* values are described in this figure. Source data are provided as a Source Data file.

biosynthesis[6,7], CARS2/CPERS is the major source of supersulphide production[26,53]. In fact, supersulphides produced by CARS2/CPERS mediate the electron transport chain and maintain the bioenergetics in a critical way[26,37,38]; in addition, mitochondrial dysfunction may be involved in the pathogenesis of COPD[54,55]. Mitochondria play a pivotal role in the regulation of cellular senescence involving oxidative stress[56,57], and oxidative stress and ageing are the common denominators of the pathogenesis of chronic lung diseases including COPD and IPF[16–18]. The translational impact is extended to the breath sulphuromics that we described herein. Our present data therefore support our proposal that supersulphides provide a useful remedy for various viral infections including COVID-19 and even various chronic lung diseases such as COPD and IPF, as well as physiological lung ageing.

# Methods

## Materials
Cysteine, GSH, GSSG, E-64, ebselen, Lipofectamine 2000, Lipofectamine RNAiMAX, and other reagents were obtained from Nacalai Tesque, FUJIFILM Wako Pure Chemical Industries, Invitrogen, and Sigma-Aldrich; all remaining materials were from Sigma-Aldrich unless specified otherwise. Authentic cysteine persulphides and supersulphides and glutathione persulphides and supersulphides were prepared according to methods previously reported[2,26].

## Calculations of structures and charge density for supersulphides
We used the LX 2U Twin2 server 406 Rh-2 (NEC) with Gaussian 16 (Rev. C.01) in the Research Center for Computational Science for calculations of optimized structures of all molecules. We performed all density functional calculations at B3LYP/6-311 G(2d,p) in a vacuum. Charge density was also evaluated by using natural population analysis with Gaussian 16. We confirmed that no imaginary frequency exists in optimized structures with the lowest internal energy by using frequency calculations.

## Preparation of GSSSG
Oxidized GSSSG were synthesized as previously described[58]. GSSSG were diluted in distilled water to make a 1 mM stock solution; 30 mM sodium acetate was added to adjust the stock solution pH to 5.0. The stock solution was diluted to proper concentrations by using DMEM in the in vitro study. In the in vivo study, the stock solution was used without dilution.

## Cell culture
Madin-Darby canine kidney (MDCK) cells, DBT cells, and HFL-1 were obtained from the American Type Culture Collection (Rockville, MD, USA). VeroE6/TMPRSS2 (No. JCRB1819) cell lines were purchased from the Japanese Collection of Research Bioresources (JCRB) Cell Bank and the National Institute of Biomedical Innovation. MDCK cells and DBT cells were cultured in DMEM (high glucose) with culture conditions of 10% fetal bovine serum and 1% penicillin/streptomycin at 37 °C in a humidified 5% $CO_2$ atmosphere, while HFL-1 and VeroE6/TMPRSS2 cells were cultured in DMEM (low glucose) under the same culture conditions.

## Histopathology
Two or 3 sections per lung and 7 to 9 lungs per experimental group were characterized for lung injury scoring analysis. At least 20 random high-power fields (×400 total magnification) were independently scored. Lung injury scores were quantified by two investigators blinded to the treatment groups using previously published criteria[59,60]. Briefly, lung injury was assessed on a scale of 0–2 for each of the following criteria: (i) neutrophils in the alveolar space, (ii) neutrophils in the interstitial space, (iii) number of hyaline membranes, (iv) amount of proteinaceous debris, and (v) extent of alveolar septal thickening. The final injury score was derived from the following calculation: Score = [20*(i) + 14*(ii) + 7*(iii) + 7*(iv) + 2*(v)] / (number of fields *100), which finally gives an overall score of between 0 and 1.

## Real-time RT-PCR analysis for influenza virus PR8
We harvested lungs from infected mice at various time points after influenza virus infection. We extracted total RNA from lungs by using Isogen (Nippon Gene); reverse transcribed 1 ng of total RNA; and used a SYBR green-based real-time PCR method, One-Step TB Green PrimeScript PLUS RT-PCR Kit II (Takara Bio). For quantification of influenza A/H1N1 virus membrane genes, we used forward and reverse primers: 5′-GGACTGCAGCGTTAGACGCTT-3′ and 5′-CATCCTGTT GTATATGAGGCCCAT-3′, respectively. The amplification program was as follows: 42 °C for 10 min and 95 °C for 10 s followed by 40 cycles at 95 °C for 5 s and at 60 °C for 30 s. The specificity of the assay was confirmed by using a melting curve analysis at the end of amplification. In vitro transcribed viral RNA was used as different known amounts of standard samples.

## Real-time RT-PCR for SARS-CoV-2
Real-time RT-PCR for SARS-CoV-2 in hamster EBC samples, lungs and supernatant of VeroE6/TMPRSS2 cell culture was performed using Ampdirect 2019-nCoV Detection Kit (Shimadzu). Experiments were performed according to the manufacturer's instructions. For the RT-PCR of cell culture supernatant of VeroE6/TMPRSS2 cells and lungs of infected hamsters, we extracted total RNA by using Isogen (Nippon Gene) and used One step TB Green PrimeScript PLUS RT-PCR kit (Takara Bio). We used forward and reverse primers: 5′-AAATTTTGGGGACCAGGAAC-3′ and 5′-TGGCAGCTGTGTAGGTCAAC-3′, respectively. The amplification program was as follows: 42 °C for 10 min and 95 °C for 1 min followed by 45 cycles at 95 °C for 5 s and at 60 °C for 30 s.

## Differential cell counts in BALF
After influenza virus infection, BALFs were obtained from mice at various time points until day 14 as previously described[61]. Lungs were lavaged twice with 0.75 ml of PBS, pH 7.0. After centrifugation, the

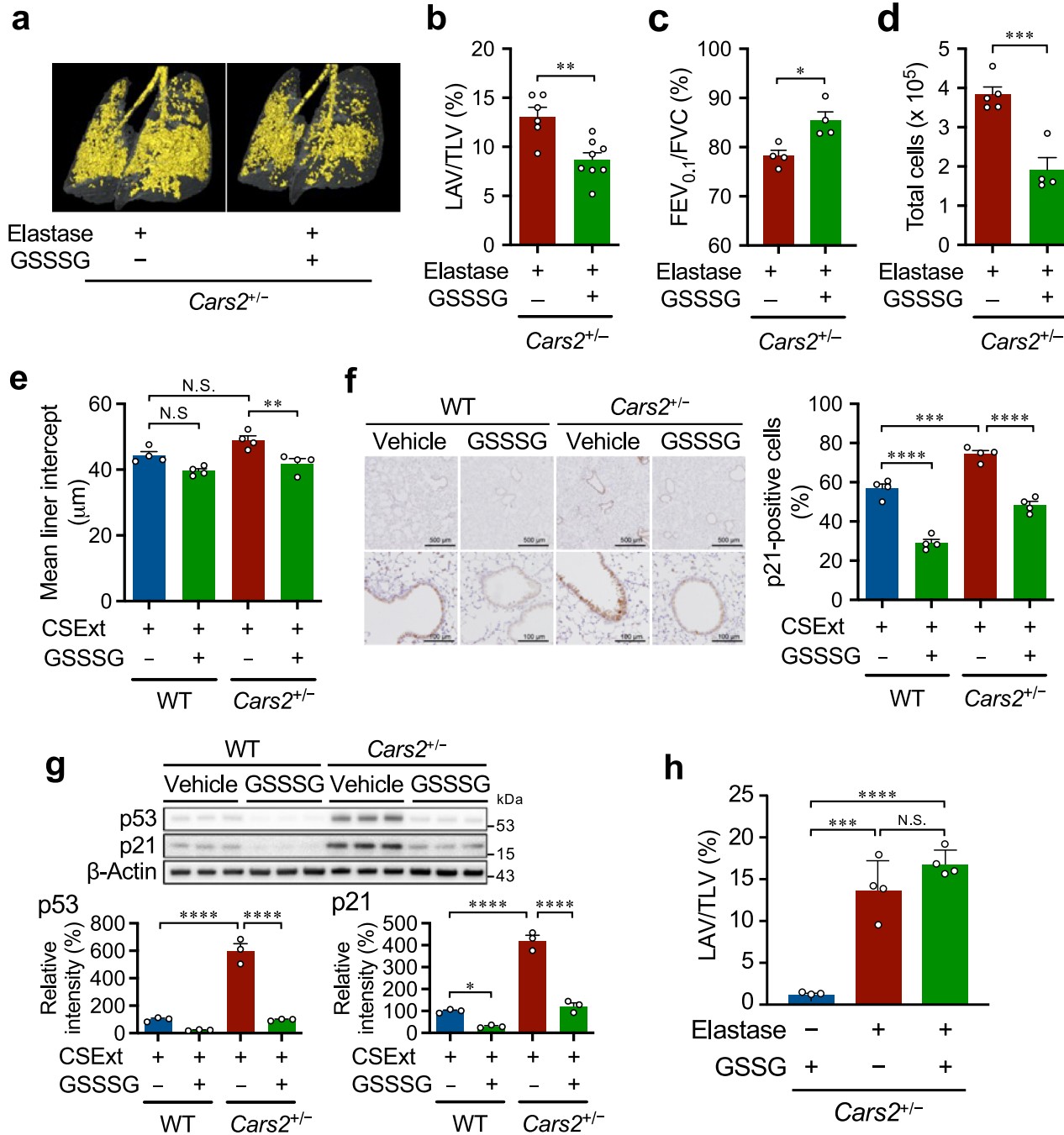

**Fig. 9 | Alleviation of COPD pathology in lungs of elastase- and CSExt-induced COPD models by supersulphides. a** Representative 3D micro-CT images of lungs of elastase-treated $Cars2^{+/-}$ mice administered with 1 mM GSSSG. Yellow indicates the low-attenuation area; the total lung area appears as a transparent shape. Each dot represents data from an individual mouse ($n = 3$ per group). **b** LAV/TLV was quantified by calculating the ratio of LAV to TLV. Elastase-treated $Cars2^{+/-}$ mice treated ($n = 8$) or untreated ($n = 6$) with 1 mM GSSSG. $P = 0.0024$. **c** $FEV_{0.1}$/FVC for elastase-treated $Cars2^{+/-}$ mice with or without 1 mM GSSSG administration ($n = 4$ per group), as measured by using the flexiVent system. $P = 0.014$. **d** Numbers of total cells in BALF from elastase-treated $Cars2^{+/-}$ mice administered with 1 mM GSSSG. $P = 0.0009$. **e** Quantification of the mean linear intercept for CSExt-treated WT and

$Cars2^{+/-}$ mice with or without administration of 1 mM GSSSG ($n = 4$ per group). $P = 0.0055$. **f** p21 immunostaining with lungs of CSExt-induced emphysema in WT and $Cars2^{+/-}$ mice with or without GSSSG treatment, shown as representative images (left panel) and quantitative results (right panel). $P < 0.0001$ (left), $= 0.0002$, and $<0.0001$ (right). Scale bars, 500 μm (low magnification) and 100 μm (high magnification). **g** p53 and p21 protein levels as analyzed by using Western blotting (top panel) and quantitative results of p53 (bottom left panel) and p21 (bottom right panel). All $P < 0.0001$, but *$P = 0.036$. **h** Lack of effect of GSSG on the LAV/TLV for micro-CT images of lungs of elastase-induced $Cars2^{+/-}$ mice. $P = 0.0003$ and $<0.0001$. Data are means ± s.d. *$P < 0.05$, **$P < 0.01$, ***$P < 0.001$, ****$P < 0.0001$, N.S., not significant. Source data are provided as a Source Data file.

total number of BAL cells resuspended in 1 ml of PBS were counted with a hemocytometer. Cytospin cells were stained with Diff-Quik (Sysmex) to differentiate macrophages, lymphocytes, and neutrophils on the basis of cell morphology and staining characteristics. Red blood cells were excluded from BAL cells and lung cells by using ACK Lysing Buffer (Life Technologies).

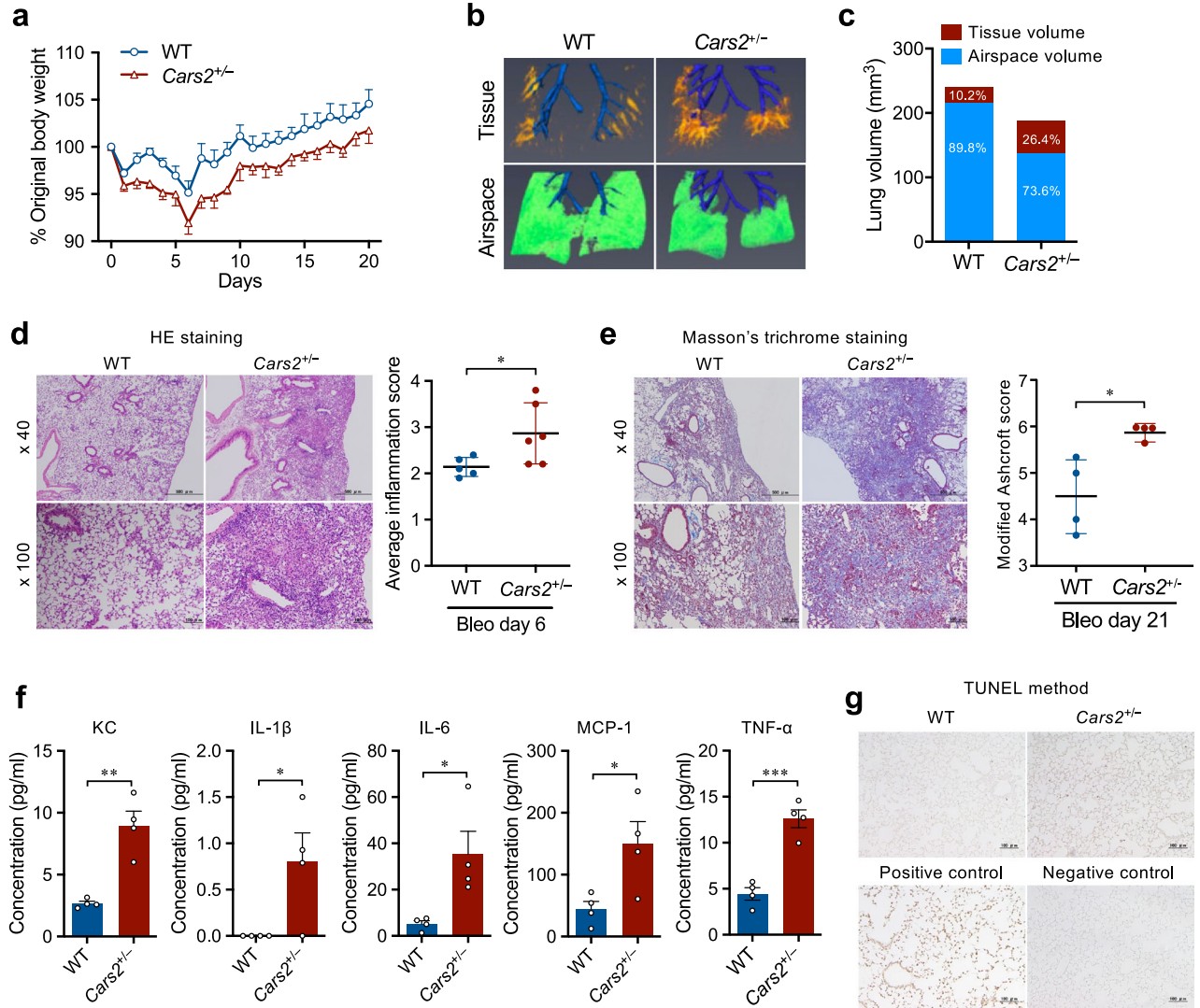

**Fig. 10 | Enhanced pulmonary fibrosis in *Cars2*⁺/⁻ mice (bleomycin-induced IPF model). a** Body weight changes in bleomycin-treated WT (*n* = 5) and *Cars2*⁺/⁻ (*n* = 4) mice. **b** On day 21 after intranasal bleomycin administration, mice underwent micro-CT evaluation of the lungs. Representative 3D reconstruction of lungs from bleomycin-treated WT and *Cars2*⁺/⁻ mice. Tissue volume appears orange-red and airspace volume appears green. Conducting airways that were ignored in the analysis appear in blue. **c** Quantification of tissue and airspace volumes in lungs in (**b**). **d** Representative images (left panel) and semi-quantification (right panel) of HE staining in lungs from bleomycin-treated WT and *Cars2*⁺/⁻ mice on day 6. *P* = 0.044. Scale bars, 500 μm (low magnification, ×40) and 100 μm (high magnification, ×100). **e** Representative images (left panel) and semi-quantification (right) of

Masson's trichrome staining in lungs from bleomycin-treated WT and *Cars2*⁺/⁻ mice on day 21. *P* = 0.034. Scale bars, 500 μm (low magnification, ×40) and 100 μm (high magnification, ×100). **f** Concentration of keratinocyte chemoattractant (KC), IL-1β, IL-6, MCP-1, and TNF-α in BALF of WT and *Cars2*⁺/⁻ mice at 6 days after intranasal administration of bleomycin. Each dot represents data from an individual mouse (*n* = 4 per group). *P* = 0.0017 (KC), 0.041 (IL-1β), 0.024 (IL-6), 0.033 (MCP-1), 0.0004 (TNF-α). **g** Apoptotic cells in lung sections obtained on day 6 after intranasal administration of bleomycin were identified by means of the TUNEL method. Scale bars, 100 μm. Data are means ± s.d. *P < 0.05, **P < 0.01, ***P < 0.001. Source data are provided as a Source Data file.

## Generation of *CARS2* KD cell lines

*CARS2* KD cell lines were produced by using the pSUPER RNAi system (OligoEngine). Briefly, short hairpin RNA sequences of green monkey *CARS2* were designed according to the green monkey *CARS2* sequence (Gene ID: 103214798) by the BLOCK iT™ RNAi design software of Thermo Fisher Scientific. Two pair oligos for *CARS2* shRNA, sense: 5′-GATCCCCGAGGAGACAAGTATGGCAAACTTCAAGAGAGTTTGCCAT ACTTGTCTCCTCTTTTTA-3′ and anti-sense: 5′AGCTTAAAAAGAGGAG AAAGTATGGCAAACTCTCTTGAAGTTTGCCATACTTGTCTCCTCGGG-3′ were synthesized, annealed and ligated into the *Bgl* II and *Hind* III sites of linearized pSUPER retro puro vector. The ligation product was transformed into *Escherichia coli*, clones with the shRNA insert were selected by sequence analysis, and purification of the pSUPER-shRNA plasmid for transfection was performed using NucleoBond Xtra Midi

(MACHEREY-NAGEL GmbH & Co. KG). VeroE6/TMPRSS2 cells were plated in six-well plates (1 × 10⁵ cells per well) at 24 h before transfection. Cultured cells were transfected with 5 μg of pSUPER-shRNA (CARS2) plasmid or empty vector as control by using Lipofectamine 3000 (Thermo Fisher Scientific). The medium was changed 12 h after transfection, and after another 24 h of incubation, the cells were replated on 10-cm dishes and cultured in medium with 7 μg/ml puromycin (Invitrogen). Following the selection of puromycin-resistant clones, the identification of knockdown cell lines of *CARS2* was accomplished via expression analysis using western blotting.

## Coronavirus plaque-forming assay

The plaque-forming assay was according to the method we previously reported[62]. Briefly, the WT and *CARS2* KD VeroE6/TMPRSS2 cells

$(1 \times 10^6$ cells/well) were seeded in a six-well plate and incubated for overnight. SARS-CoV-2 virus (30 pfu formed in the WT VeroE6/TMPRSS2 cells, 100 µl/well) were inoculated onto monolayer cells for 60 min and the cells were overlayed with medium containing 1.5% agarose and infected for 40 h post-infection. The cells were fixed for 3 h with 4% formaldehyde, discard the overlay, and stained with 1% crystal violet/20% ethanol solution. The numbers of infectious virus were determined as pfu/well. All SARS-CoV-2 infection experiments with VeroE6/TMPRSS2 cells were performed at the biosafety level 3 (BSL3). The plaque-forming assay was employed for MHV (A59) and followed the same conditions of the plaque-forming assay for SARS-CoV-2, with the exception that DBT cells were used instead of VeroE6/TMPRSS2 cells[63].

## SARS-CoV-2 plaque-reduction assay

The plaque-reduction assay was performed according to the aforementioned plaque-forming assay with modification. Briefly, the WT VeroE6/TMPRSS2 cells were infected by SARS-CoV-2 virus for 30 min at 37 °C and treated with appropriate concentrations of test compounds (GSSSG, GSSG, and GSH) at 2 h prior and 1 h post-inoculation. After the incubation in the medium containing 1.5% agarose for 40 h, the cells were fixed with formaldehyde and stained with crystal violet. The plaque formation of infectious virus was measured as total plaque area/well $(mm^2)$.

## Viral infectivity assay

The viral infectivity assay was developed in this study. Briefly, Influenza virus $(1 \times 10^7$ pfu per 100 µl) was ultrafiltered using an Amicon Ultra with 100 kDa cutoff (Merck Millipore). The virus was incubated with various concentrations of $Na_2S_{1-4}$, GSSSG, GSSG, and GSH for 30 min at 37 °C and then diluted with DMEM/0.2% BSA. The virus was serially diluted twofold and inoculated at 100 µl/well to the MDCK cells in 96-well plates for 60 min and incubated for 24 h. SARS-CoV-2 virus $(1 \times 10^6$ pfu per 100 µl) was also ultrafiltered using the Amicon Ultra 100 K and then incubated with supersulphides at 37 °C for 30 min. The virus was serially diluted and infected to the WT VeroE6/TMPRSS2 cells for 60 min and incubated for 24 h. The infected cells were washed twice with PBS. The viral infectivity was assessed by measuring the amount of infectious virus yielded in MDCK cells by the quantitative RT-PCR.

## Viral proteome analysis of SARS-CoV-2-infected VeroE6/TMPRSS2 cells

To inactivate the infectivity of the virus in the cell culture supernatant, 5 µl of acetic acid and 5 µl of 2-propanol were added to 10 µl of cell culture supernatant[64,65]. The inactivated samples were then with a known amount of internal standard peptides and concentrated using SpeedVac Concentrator. The concentrates were then resuspended with 4 µl of 0.5% ProteaseMAX™ Surfactant (Promega), 4 µl of 1 M ammonium bicarbonate, 1 µl of 100 mM TCEP, and 10 µl of distilled water, and then subjected to incubation in a shaker (2400 rpm) at 37 °C for 15 min. Alkylation of the proteins was achieved by adding 3.5 µl of 100 mM iodoacetamide (IAM) to the samples followed by agitation using a shaker (1200 rpm) at 37 °C for 15 min. To digest viral proteins, the alkylated samples were incubated with 20 µl of 100 ng/µl Trypsin Gold (Promega) at 40 °C for 3 h. After digestion, 5.5 µl of 5% formic acid was added to terminate the digestion reaction. Following centrifugation, 50 µl of the resultant supernatants was injected into LC-ESI-MS/MS (LCMS-8060NX; Shimadzu) coupled to the Nexera UHPLC system (Shimadzu). Peptides derived from the virus were separated by Nexera UHPLC with Intrada WP-RP (30 × 3 mm, 3 µm, Imtakt) and Shim-pack Velox C18 (100 × 2.1 mm, 2.7 µm, Shimadzu) under the following elution conditions: mobile phase A (0.1% formic acid) with a gradient of mobile phase B (0.3 mM ammonium fluoride and 0.1% formic acid in methanol) from 2 to 100% (2–14% B: 0-8 min,

14−35% B: 8-14 min, 35−47% B: 14−20 min, 47−100% B: 20−20.5 min) at a flow rate of 0.5 ml/min at 50 °C. MS spectra were obtained at each temperature of the ESI probe at 400 °C, desolvation line at 150 °C and heat block at 400 °C; nebulizer, heating, and drying nitrogen gas flows were set at 3, 10, and 10 L/min, respectively. The peptides derived from the virus were identified and quantified by means of multiple reaction monitoring (MRM). Several target peptides were quantified using an external standard curve generated from a synthesized standard peptide. Quantitation of the other peptide was performed by isotope-labeled internal standard. The MRM parameters of target peptides are indicated at Supplementary Table 1.

## Animals

Six- to 7-week-old male C57BL/6J mice were purchased from Japan SLC, Inc. *Cars2*[+/−] mice were established as our previously report[26]. Generation of *Cars2*[AINK/+] and *Cars2*[AINK/+]::ACE2-Tg mice are described below. Mice were housed in a specific pathogen-free facility and maintained under constant temperature (24 °C), humidity (40%), and light cycle, with food and water provided ad libitum. Six-week-old male Syrian hamsters were purchased from Japan SLC, Inc. Hamsters were housed in a specific pathogen-free facility and maintained under the same conditions as that for mice. All experiments with live SARS-CoV-2 were performed in the animal biosafety level 3 containment laboratories at the Tohoku University. All experimental procedures conformed to the Regulations for Animal Experiments and Related Activities at Tohoku University, were reviewed by the Institutional Laboratory Animal Care and Use Committee of Tohoku University, and were finally approved by the President of Tohoku University (#2019MdA-072, #2020MdA-139, #2021MdA-053).

## Virus infections

Influenza virus A/H1N1 (PR8), SARS-CoV-2 (JPN/TY/WK-521 strain, provided by the National Institute of Infectious Diseases, Japan)[29], MHV (A59) were propagated in MDCK cells, VeroE6/TMPRSS2 cells, and DBT cells, respectively. The plaque-forming assay and the plaque-reduction assay for SARS-CoV-2 with VeroE6/TMPRSS2 cells are described above. The mouse model of influenza virus (PR8) infection was produced as previously described[21]. Briefly, 12-weeks-old male WT and *Cars2*[+/−] mice were anesthetized via an intraperitoneal injection of medetomidine (ZENOAQ) (0.6 mg/kg), butorphanol (Meiji Seika Pharma Co., Ltd.) (10 mg/kg), and midazolam (Astellas Pharma Inc.) (8 mg/kg) and inoculated intratracheally (i.t.) with influenza virus (PR8) at dose of $2 \times 10^4$ plaque-forming units (pfu; five times the minimal lethal dose, 20 µl). At 6 h before influenza virus infection with WT mice, they were administered of 50 µl of 1 mM GSSSG (32 µg) or PBS intranasally as a vehicle control. We then monitored survival and body weight loss until day 14 post-infection. Because most of *Cars2*[+/−] mice died or body weight of mice was reduced to the extent and they needed to be euthanized with 5% isoflurane according to the guideline of animal experiment (Tohoku University). Therefore, the change of body weight was monitored until 6 or 7 days after infections. For the mouse model of SARS-CoV-2 infection, 15- to 17-weeks-old male and AINK::ACE2-Tg or ACE2-Tg mice were anesthetized by intraperitoneal injection of medetomidine, butorphanol, and midazolam and then inoculated i.t. with 50 µl containing $1 \times 10^2$ pfu SARS-CoV-2. We then monitored survival and body weight loss of infected mice until day 14 post-infection. On day 4 post-infection, several infected mice were euthanized for sampling of the lungs for further analyses. For hamster model of SARS-CoV-2 infection, 7-weeks-old male hamsters were anesthetized by intraperitoneal injection of medetomidine, butorphanol, and midazolam and then inoculated i.t. with 120 µl containing approximately 1.0 $LD_{50}$ dose of SARS-CoV-2 $(6 \times 10^6$ pfu) simultaneous i.t. administration of 100 or 300 µg GSSSG. The hamsters were subsequently administered i.p. 500 or 1000 µg GSSSG. On day 4 post-infection, the infected hamsters were euthanized to obtain the lung

specimens for pathological examinations. For example, the pathological analysis was performed by measuring the area of consolidation (i.e., pathological lesion due to viral pneumonia) of hamster lungs. Specifically, the macroscopic images of lung lesions that were photographically observed were analyzed semi-quantitatively by ImageJ software (National Institutes of Health)[26]. The morphometric data of consolidation thereby obtained were expressed as percentages of the whole area of the corresponding lung.

### Generation of CPERS activity-selective *Cars2*-deficient mice (*Cars2*^AINK/+)

We used *Cars2*^+/− mice according to our previous report[26]. To generate mice that were selectively deficient in CPERS activity, we introduced mutations in the ^109KIIK^112 motif of mouse *Cars2*, which is essential for CPERS activity, i.e., supersulphide synthesis. Briefly, Cas9 mRNA and *Cars2* gRNA vector were constructed by using pSP64-hCas9 plasmid and pT7-sgRNA plasmid. After digestion of the pSP64-hCas9 plasmid with *Sal* I, humanized Cas9 (hCas9) mRNA was synthesized by means of in vitro RNA transcription kit (mMESSAGE mMACHINE Sp6 Transcription Kit; Ambion) according to the manufacturer's instructions. To construct the *Cars2* gRNA vector, a pair of oligonucleotides targeting *Cars2* was annealed and inserted into the *Bbs* I site of the pT7-sgRNA vector. The oligonucleotide sequences for the gRNA were designed as follows: 5′-GGACAGATCCAGCGAACAGG-3′ and 5′-AATAATCAAGAGAGCTAACG-3′. After digestion of pT7-sgRNA with *Xba* I, gRNA was synthesized by using the MegaShortScript Kit (Ambion). Donor DNA oligonucleotides encoding a lysine to alanine substitution were designed as follows: 5′-TCATGGCGATGAGCATT ACCGACGTGGATGAC**GC**AATAATC**GC**GAGAGCTAACGAGGTAAGCAG CCTCCC-3′ (nucleotides corresponding to the substitution in KIIK are underlined). B6D2F1 (C57BL/6NCr × DBA/2Cr F1) mice were used to obtain fertilized eggs. In brief, we introduced Cas9 mRNA, gRNA, and donor DNA oligonucleotide into B6D2F1 fertilized eggs with Genome Editor (BEX Co. Ltd.), after which we transferred eggs to oviducts of pseudo-pregnant females on the day of the vaginal plug. We obtained *Cars2*^AINK/+ mice by replacing the KIIK motif with AINK, with confirmation by Sanger sequencing. *Cars2*^AINK/+ mice were backcrossed to C57BL/6J mice for more than eight generations, and the progeny were used in this study. It is worth noting that the magnitude of reduced supersulphides is smaller in *Cars2*^AINK/+ mice than in *Cars2*^+/− mice, because only one of the two catalytic sites for persulphide biosynthesis is eliminated in *Cars2*^AINK/+ mice, whereas *Cars2*^+/− mice are deficient in both sites, thereby persulphide production should be much more attenuated in *Cars2*^+/− mice than in *Cars2*^AINK/+ mice. It is reasonable that the phenotypic changes dependent on supersulphides produced by CARS2 are less evident in *Cars2*^AINK/+ mice than in *Cars2*^+/− mice. Hence, the utilization of *Cars2*^+/− mice is still necessary, instead of *Cars2*^AINK/+ mice, concomitant with the administration of supersulphide donors, such as GSSSG, for pharmacological intervention.

### Generation of SARS-CoV-2 sensitive *Cars2*-deficient mice (*Cars2*^AINK/+::ACE2-Tg)

As mice are inherently resistant to SARS-CoV-2 infection, it is necessary to express human ACE2 in mice tissues for being infectious for SARS-CoV-2. The CAG-hACE2 mice are transgenic mice expressing human ACE2 driven by CAG, a promoter for strong and ubiquitous expression[30]. The heterozygous hACE2 mice (ACE2-Tg) were crossed with *Cars2*^AINK/+ mice to obtain SARS-CoV-2-sensitive *Cars2*^AINK/+ mice. The genotype of the transgenic mice was determined by PCR for ear DNA using the primer sets 5′-CTTGGTGA-TATGTGGGGTAGA-3′ and 5′-CGCTTCATCTCCCACCACTT-3′. *Cars2*^AINK/+ mice were crossed with ACE2-Tg to obtain *Cars2*^AINK/+::ACE2-Tg. The ACE2-Tg mice were purchased from the Laboratory Animal Resource Bank of the National Institutes of Biomedical Innovation, Health, and Nutrition.

### Isolation of mitochondria from liver

We isolated mitochondrial fractions from livers from WT and *Cars2*^AINK/+ mice as described previously[26]. Briefly, liver tissues were homogenized in isotonic buffer (10 mM HEPES, 75 mM sucrose, 225 mM mannitol, 2 mM EDTA, pH 7.4) with a Teflon homogenizer for 15 strokes at 700 rpm, followed by centrifugation at $700 \times g$ for 10 min at 4 °C. Supernatants were centrifuged again at $5000 \times g$ for 10 min at 4 °C, and pellets were washed twice and resuspended in isotonic buffer. Expression levels of CARS2 protein in WT and *Cars2*^AINK/+ mice were evaluated by using western blotting of isolated mitochondria from liver tissue and whole liver tissue specimens.

### Western blotting

To determine CARS2 expression levels in control and *CARS2* KD VeroE6/TMPRSS2 cells, primary bronchial epithelial cells and lung fibroblasts under basal conditions, we seeded primary cells in a 60-mm dish and cultured them for 48 h. After medium was replaced with growth factor-free keratinocyte serum-free medium or serum-free DMEM for the next 24 h, cells were harvested by using cell lysis buffer. Cell lysates were homogenized for a few seconds with an ultrasound sonicator. To determine the amounts of cleaved caspase-3 and 3-NT in mouse lung, on day 5 after PPE administration, lung tissues were obtained, immediately snap-frozen, and then homogenized with 100 μl of Tissue Protein Extraction Reagent (Thermo Fisher Scientific) with proteinase and a phosphatase inhibitor cocktail, EDTA free (Thermo Fisher Scientific). Lungs obtained from mice exposed to CSExt were similarly homogenized. To determine the effect of CSExt on protein levels of CARS2, p21, and p53 in HFL-1 cells, cells exposed to CSExt or PBS for 10 days were washed with ice-cold PBS and homogenized in radio-immunoprecipitation assay buffer. The amounts of protein in cell lysates and lung homogenates were quantified by using the bicinchoninic acid (BCA) method according to the manufacturer's instructions (Thermo Fisher Scientific). Equal amounts of protein were loaded and separated by electrophoresis on 12% sodium dodecyl sulfate (SDS) polyacrylamide gels. After electrophoresis, separated proteins were transferred onto polyvinylidene fluoride membranes (Millipore) and membranes were blocked with a blocking reagent (TOYOBO). Membranes were then incubated with primary antibodies overnight at 4 °C. The following antibodies were used to detect target proteins: rabbit polyclonal anti-cysteinyl-tRNA synthetase 1 (CARS1) antibody (1:5000 dilution, HPA002383; Sigma-Aldrich), rabbit polyclonal anti-human CARS2 antibody (1:2000 dilution, HPA043935; Sigma-Aldrich), anti-mouse CARS2 antibody (1:5000 dilution)[26], mouse monoclonal anti-mitochondrial cytochrome *c* oxidase 1 (MTCO1) antibody (1:5000 dilution, ab14705; Abcam), mouse monoclonal anti-succinate dehydrogenase subunit A (SDHA) antibody (1:5000 dilution, ab14715; Abcam), mouse monoclonal anti-p53 antibody (1:200 dilution, 2Q366; Santa Cruz Biotechnology), rabbit monoclonal anti-p21 antibody (1:2000 dilution, 2947; Cell Signaling Technology), mouse monoclonal anti-β-actin antibody (1:10,000 dilution, sc-1615; Santa Cruz Biotechnology), and rabbit polyclonal anti-3-NT antibody (1:1000 dilution, 06-284; Upstate Biotechnology), and rabbit polyclonal anti-glyceraldehyde-3-phosphate dehydrogenase (GAPDH) antibody (1:5000 dilution, sc5778, Santa Cruz Biotechnology). Membranes were then washed 3 times with TBS (20 mM Tris-HCl, 150 mM NaCl, 0.1% Tween 20, pH 7.6) and were incubated with horseradish peroxidase-conjugated secondary antibodies for 1 h at room temperature. ECL plus a western blotting reagent (Amersham Biosciences) was used to detect immunoreactive bands via a luminescent image analyser (LAS-4000; Fujifilm). Band intensity was quantified by means of densitometry with ImageJ 1.52 v software.

## Supersulphide metabolome analysis of VeroE6/TMPRSS2 cells and mouse lung samples

VeroE6/TMPRSS2 cells or mouse lungs (50 mg) were homogenized in 0.15 ml or 0.5 ml of cold methanol solution containing 5 mM β-(4-hydroxyphenyl)ethyl iodoacetamide (HPE-IAM) and 20 mM sodium acetate buffer (pH 6.5), after which samples were incubated for 20 min at 37 °C. Following centrifugation (14,000 × $g$ for 10 min at 4 °C), lysate supernatants were diluted with 0.1% formic acid containing known amounts of isotope-labeled internal standards, and then the LC-ESI-MS/MS (LCMS-8060NX) measurements were performed. Centrifugation pellets were dissolved in PBS containing 0.1% SDS, after which protein concentrations were determined by using the BCA assay. LC-ESI-MS/MS conditions and isotope-labeled internal standards synthesis were employed in the same manner as described earlier[24,26,58].

## Preparation and purification of recombinant proteins of SARS-CoV-2 and MHV

PL$^{pro}$ and 3CL$^{pro}$ gene sequences were obtained from National Center for Biotechnology Information database (NC_045512.2). PL$^{pro}$ and 3CL$^{pro}$ gene sequences were adapted to *E. coli* codon usage and were chemically prepared as-synthesized nucleotides (Integrated DNA Technologies). The PL$^{pro}$ synthesized DNA was introduced into pET53 with Gateway (Thermo Fisher Scientific), and the 3CL$^{pro}$ synthesized DNA was introduced into the pE-SUMO vector with the restriction enzymes *Bsa*I and *Xba*I (BioLabs). The plasmid contains S protein gene of SARS-CoV-2 adapted to human codon usage was kindly gifted from Dr. Krogan (University of California, San Francisco). To obtain MHV 3CL$^{pro}$, MHV was infected into DBT cells, and the supernatants were harvested at 24 h after infection. Viral RNA from the supernatants was extracted by using Isogen-SL (Nippon Gene). TaKaRa One Step RNA PCR Kit (Takara) was used to cDNA synthesis and amplification of MHV 3CL$^{pro}$. We used forward and reverse primers: 5′-CTGGTATAGTGAA-GATG-3′ and 5′-TTACTGTAGCTTGACACCAGCTAG-3′, respectively for the cloning of MHV 3CL$^{pro}$. The amplification program was as follows: 42 °C for 10 min and 95 °C for 10 s followed by 40 cycles at 95 °C for 5 s and at 60 °C for 30 s. The PCR fragment of MHV 3CL$^{pro}$ was subcloned into *E. coli* expression vector pE-SUMO vector. SARS-CoV-2 S protein was purchased (ACROBiosystems, SPN-C52H9) or produced as recombinant protein from HEK293T cell lines. *E. coli* NiCo21(DE3) transformed with pET53-SARS-CoV-2 PL$^{pro}$ was grown at 37 °C to an OD$_{600}$ of 0.6–0.7, after which 0.2 mM isopropyl β- D-thiogalactopyranoside (Sigma-Aldrich) was added to the culture. After incubation for 16 h at 16 °C, cells were harvested via centrifugation; were resuspended in lysis buffer [20 mM Tris-HCl pH 7.5, 150 mM NaCl, 10 mM imidazole, 1 mM DTT] containing 0.5 mg/ml lysozyme; and were lysed by sonication. The cell lysate was centrifuged at 12,000× $g$ for 20 min, and the supernatant was loaded onto a column packed with Ni-NTA agarose (FUJIFILM Wako Pure Chemical Corporation) equilibrated with lysis buffer. After the column was washed with washing buffer (20 mM Tris-HCl pH 7.5, 150 mM NaCl, 20 mM imidazole, 1 mM DTT), His-tagged protein was eluted in a stepwise fashion with 50 mM, 100 mM, 150 mM, 200 mM, and 250 mM imidazole. After desalting by using an PD-10 column (GE Healthcare) with desalting buffer (20 mM Tris-HCl pH 7.5, 150 mM NaCl, and 1 mM DTT), the protein was stored at −80 °C. Protein concentration was determined by using the Protein Assay CBB Solution (Nacalai Tesque), and protein purity was confirmed via SDS-polyacrylamide gel electrophoresis (PAGE) with Coomassie Brilliant Blue (CBB) staining. To prepare a recombinant 3CL$^{pro}$, *E. coli* BL21(DE3) was transformed with pE-SUMO-SARS-CoV-2 3CL$^{pro}$ or pE-SUMO-MHV 3CL$^{pro}$ and cultured with 0.5 mM β-D-thiogalactopyranoside for 3 h at 37 °C to induce 3CL$^{pro}$ expression. Then, 3CL$^{pro}$ fused to double tag (His6 and SUMO) was purified by using Ni-NTA agarose, desalted with PD-10 (GE Healthcare), and digested with the SUMO protease 1 (LifeSensors Inc.) at a ratio of 1 unit of the enzyme to 100 µg of protein for 3 h at 30 °C. 3CL$^{pro}$ tagged His6 were re-adsorbed into Ni-NTA agarose

column to remove SUMO and SUMO protease, and then the fraction eluted with imidazole (Nacalai Tesque) were stored at −80 °C until use. S protein gene sequences adapted to human codon usage was subcloned into the pHEK293 Ultra Expression Vector I (Takara). Subsequently, the plasmid was transfected into 293 T cells, resulting in the expression of the S protein with Strep-tag at its C-terminal region. Protein purification was carried out using Strep-Tactin Sepharose™ with Strep-Tactin™ Buffer Set according to the manufacturer's instruction. Briefly, the cells were lysed with 60 ml of ice-cold wash buffer containing 1% Nonidet P-40, 150 mU/ml avidin, and 1 × protease inhibitor cocktail (Nacalai Tesque). After homogenization, the mixture was incubated on ice for 30 min. Next, the sample was centrifuged at 12,000× $g$ and 4 ˚C for 30 min. Then, the resulting supernatant was loaded onto a Strep-Tactin Sepharose column and the column was washed three times with 3 × column volume of the ice-cold wash buffer. Finally, elution was performed by adding ice-cold elute buffer containing 0.3% Nonidet P-40, and the fraction containing the S protein was confirmed by SDS-PAGE.

## Protease analysis

Recombinant SARS-CoV-2 PL$^{pro}$ protein, SARS-CoV-2 3CL$^{pro}$ protein, and MHV 3CL$^{pro}$ protein were obtained by *E. coli* expression system. Recombinant papaya papain protein (Nacalai Tesque), and human cathepsin B protein (Elabscience) were purchased from vendors. Proteases were applied to the PD SpinTrap G-25 column (GH Healthcare) equilibrated with 50 mM HEPES buffer (pH 7.5) to remove reductants. Protease activity of PL$^{pro}$, 3CL$^{pro}$, papain, and cathepsin B was evaluated using MCA substrates according to our previous report[66] with modification: 1 µM PL$^{pro}$ or 3CL$^{pro}$ or papain or cathepsin B was incubated with various concentrations of GSSG, GSSSG, ebselen, and Na$_2$S$_{1-4}$ in 50 mM HEPES buffer (pH 7.5) for varied time periods at 37 °C. After incubation, the proteases were reacted with or without 1 mM reducing agents (DTT or TCEP) for 15 min at 37 °C. In some experiment, 3 µM PL$^{pro}$ and 3 µM 3CL$^{pro}$ were reacted with NOC7 (NO donor) for 15 min at 37 °C, followed by the addition of 0–100 µM GSSG to the reaction mixture containing 1 µM protease for 60 min at 37 °C. Each reaction mixture for PL$^{pro}$, 3CL$^{pro}$, papain, or cathepsin B was reacted with corresponding MCA substrate (10 µM; PEPTIDE Institute, Inc.), i.e., Z-Leu-Arg-Gly-Gly-MCA, Ac-Abu-Tle-Leu-Gln-MCA, Bz-Arg-MCA, or Z-Arg-Arg-MCA, respectively, in 50 mM HEPES buffer (pH 7.5) at 37 °C. Specific enzyme activities (µM) of PL$^{pro}$, 3CL$^{pro}$, papain, and cathepsin B were determined by measuring fluorescence intensity of fluorescent 7-amino-4-methylcoumarin (AMC) released from the MCA substrates via a fluorescence multiplate reader (SH-9000; Corona Electric), using wavelengths of 380 nm and 460 nm for excitation and emission, respectively. Ebselen has been demonstrated to have a potent 3CL$^{pro}$ inhibition and antiviral activity against SARS-CoV-2[67], possibly through its electrophilic binding to cysteine thiol at a catalytic center of 3CL$^{pro}$.

## Computational modeling of the 3D structures of GSSSG-bound PL$^{pro}$ and 3CL$^{pro}$ of SARS-CoV-2

Molecular docking of GSSSG to PL$^{pro}$ and 3CL$^{pro}$ of SARS-CoV-2 was performed with AutoDock Vina according to a previous report[68]. The crystal structures of SARS-CoV-2 PL$^{pro}$ and 3CL$^{pro}$ were retrieved from the Protein Data Bank (PDB ID: 6W9C and 6Y2E, respectively). The docking results were visualized via PyMOL (https://www.pymol.org).

## Proteome analyses of cysteine residues in PL$^{pro}$, 3CL$^{pro}$, and SARS-CoV-2 S protein

PL$^{pro}$ and 3CL$^{pro}$ (each 50 µg/ml) were incubated with 50 µM GSSSG in 50 mM HEPES (pH 7.5) at 37 °C for 60 min. The samples were alkylated with 2 mM IAM in 0.025% ProteaseMAX surfactant at 37 °C for 30 min, followed by digested with 10 ng/ml Trypsin Gold and Glu-C (Promega) at 37 °C for 90 min and then analyzed using liquid chromatography-electrospray ionization-quadrupole time-of-flight tandem mass

spectrometry (LC-ESI-Q-TOF MS/MS). LC-ESI-Q-TOF analysis was performed by using 6545XT AdvanceBio LC/Q-TOF (Agilent Technologies) connected to the Agilent HPLC-Chip system (Agilent). The modification analysis of the active center cysteine was performed by means of Agilent MassHunter BioConfirm software. Glutathionylation in WADNNCYLATALLTLQQQIELK and GSFLNGSCGSVGFNIDYDCVSFCY MHHME peptides which are including cysteine residues (C111 and C145, respectively) in the active site of PL$^{pro}$ and 3CL$^{pro}$ were detected by monitoring at $m/z$ 909.4437 and 1213.4677, respectively. Perthioglutathionylation of these peptides were detected by monitoring at $m/z$ 920.1017 and $m/z$ 1224.1257, respectively. SARS-CoV-2 S protein [11.1 µg/ml, ACROBiosystems, SARS-CoV-2 S protein, Super stable trimer, SPN-C52H9 (Val 16-Pro 1213)] was incubated with 0, 10, 100, and 300 µM Na$_2$S$_4$ in 150 mM Tris-HCl (pH 7.5) at 37 °C for 30 min. The samples were solubilized with 0.2% ProteaseMAX surfactant and reduced with or without 1 mM DTT. The samples were alkylated with 9.3 mM IAM at 37 °C for 10 min, followed by digested with 9.7 ng/ml Trypsin Gold at 37 °C for 3 h and then analyzed with LC-ESI-Q-TOF MS/MS connected to the Agilent HPLC system. The supersulphidation levels in LNDLCFTNVYADSFVIR and VVVLSFELLHAPATVCGPK peptides, which are including cysteine residues, were detected by monitoring at $m/z$ 1024.0014 and 1019.0637 [for CysS-AM (AM, iodoacetamide adduct)], $m/z$ 1039.9874 and 1035.0497 (for CysS-S-AM), and $m/z$ 1055.9734 and 1051.0357 (for CysS-SS-AM), respectively. The digestion efficiency of S protein by trypsin was normalized with FNGIGVTQNVLYENQK peptide ($m/z$ 912.4679).

## EBC collection for omics analysis

EBC was collected by using a non-invasive equipment combining a freezing condenser device, a mouthpiece, and a 50-ml polystyrene tube. The healthy subjects and patients with COVID-19 ($n = 22$ per group) breathed through a mouthpiece at normal frequency and tidal volume for 5–10 min while sitting comfortably and wearing a nose clip, yielding approximately 0.5–1 ml of EBC. A mouthpiece was connected to a 50-ml polystyrene tube placed in a Peltier device (GL Science). The exhaled breath was recovered as EBC by rapidly freezing at –20 °C with the Peltier device, and stored immediately at –80 °C until analysis. This study was approved by the Ethics Committee of the Institutional Review Board of Tohoku University Graduate School of Medicine (2022-1-254-1). Written informed consent was obtained from the subjects who participated. We also designed a non-invasive EBC collection system, by which we can obtain appreciable amounts of EBC from hamsters with or without SARS-CoV-2 infection. Hamsters were isolated in four rectangular polyvinyl chloride cages of each containing two animals. The exhaled aerosols emitted from the hamsters were recovered via the EBC devices. The cages were aerated by an air pump (GL Science) at a continuous airflow rate (20 l/min) with constant ambient parameters (25 °C and 50% humidity) and the aerosol in the cages containing hamster EBC were aspirated at a continuous airflow rate (2 l/min) by a suction pump (GL Science). The aerosols containing hamster EBC were collected, and stores in the same manner as human EBC just described. The EBC samples thus collected from human and hamsters were subjected to the breath omics analysis that capitalizes on the integrated metabolome and proteome as reported herein.

## Supersulphide metabolome analysis of EBC

We performed supersulphide metabolome for EBC using the LC-ESI-MS/MS system with HPE-IAM to quantitatively measure supersulphides levels as described above. Briefly, samples (human and hamster EBC) were reacted with HPE-IAM at 37 °C for 20 min. Reaction mixtures were diluted with 0.1% formic acid containing known amounts of isotope-labeled internal standards, which were then measured via the LC-ESI-MS/MS system (LCMS-8060NX).

## Viral proteome analysis of hamster EBC

The virus in the EBC samples was inactivated by treating 1 ml of EBC with 500 µl of acetate and 500 µl of 2-propanol. To the samples inactivated were added a known amount of internal standard peptides, followed by concentration in vacuo and subsequent analysis. The EBC concentrates were then resuspended with ProteaseMAX™ Surfactant and TCEP, and then alkylated by IAM. The alkylated proteins were digested by Trypsin Gold for 3 h at 40 °C, and 5% formic acid was added to terminate the digestion reaction. Following centrifugation, the resultant supernatants were injected into the LC-ESI-MS/MS system (LCMS-8060NX). Peptides derived from the virus were identified and quantified by means of MRM.

## Primary human lung-resident cell study

Healthy subjects and patients with COPD from Tohoku University Hospital participated in this study between January 2013 and July 2019. All patients with COPD satisfied the criteria of the Global Initiative for Chronic Obstructive Lung Disease (GOLD) report[69]. Current smokers were excluded, and all ex-smokers had quit smoking for at least 1 year before the study. Lung-resident cells including bronchial epithelial cells and lung fibroblasts were obtained from 15 patients with COPD and 27 non-COPD subjects. Both bronchial and peripheral lung tissues were obtained from patients with or without COPD undergoing lung cancer surgery while avoiding tumor-involved areas. Tissues were used in cultures of primary lung-resident cells. All subjects performed pulmonary function tests after enrollment in the study. Written informed consent was obtained from all subjects who participated. All experiments were approved by the Ethics Committee of the Institutional Review Board of Tohoku University Graduate School of Medicine (2022-1-254-1). Primary human bronchial epithelial cells were isolated from airways that were removed during lobectomy for lung cancer and were then cultured in a growth factor-supplemented medium (BEGM Bullet Kit)[24,70]. Cells were cultured at 37 °C in a humidified 5% CO$_2$ atmosphere and were then subcultured. Cells were routinely grown to 80% confluence, and growth was arrested overnight before the experiments by transfer to growth factor-free media. Bronchial epithelial cells from study subjects were used between passages 2 and 5. Primary human lung fibroblasts were isolated and cultured according to a procedure described in a previous study[71]. Briefly, portions of lung parenchymal tissue that were as distal from any tumor as possible, were free of the pleural surface, and did not contain any cartilaginous airways were dissected under sterile conditions, minced, and placed in culture. HFL-1 cells were obtained from the American Type Culture Collection. Primary lung fibroblasts (between passages 3 and 7) and HFL-1 cells (between passages 14 and 20) were cultured in tissue culture dishes with DMEM supplemented with 10% fetal calf serum, 100 µg/ml penicillin, 250 µg/ml streptomycin, and 2.5 µg/ml amphotericin B.

## Measurement of mitochondrial membrane potential

To determine the membrane potential of mitochondria of primary lung-resident cells obtained from study subjects, tetraethylbenzimidazolyl carbocyanine iodide (JC-1) (Cayman Chemical) staining was used according to the manufacturer's protocol. Briefly, primary human bronchial epithelial cells and primary human lung fibroblasts were seeded in 96-well black glass-bottom plates at a density of $1 \times 10^5$ cells per ml and were cultured for 48 h. Cells were then incubated with 2 µM JC-1 staining solution at 37 °C for 15 min and were rinsed twice with JC-1 assay buffer. Membrane potential was then determined by measuring green and red fluorescence intensities with a fluorescence microplate reader (SpectraMax M2e; Molecular Device Co. Ltd.) at an excitation wavelength of 485 nm and an emission wavelength of 535 nm for green fluorescence (JC-1 monomer), and at an excitation wavelength of 535 nm and an emission wavelength of 595 nm for red fluorescence (JC-1 aggregates)[26]. In cells with a high

membrane potential, JC-1 forms complexes known as J-aggregates with an intense red fluorescence. In cells with a low membrane potential, however, JC-1 remains as monomeric with a green fluorescence. A high ratio of red to green fluorescence indicates a higher mitochondrial membrane potential.

## Murine emphysema models

WT C57BL/6J mice and *Cars2*[+/−] mice were given an intranasal instillation of PPE (3 U per mouse) (FUJIFILM Wako Pure Chemical Industries Ltd.) in 50 μl of PBS or were given 50 μl of PBS alone on day 0 as previously described[72]. GSSSG (1 mM), GSSG (1 mM), or 10 mM PBS was administered intranasally 6 h before elastase administration, repeated every other day 3 times. On days 5 and 21 after elastase administration, mice were killed by exsanguination and lungs were lavaged three times with 500 μl of PBS. Lavage fluid was centrifuged at $800 \times g$ for 5 min. BALF supernatants were collected for biochemical analyses, and cell pellets were resuspended in 500 μl of 10 mM PBS. The total cell number was counted by using a Scepter handheld automated cell counter (Millipore). Differential cell counts were obtained from cytospin-prepared cells stained with Diff-Quick (Sysmex). To generate Cigarette smoke-induced murine emphysema model, 6- to 7-week-old WT C57BL/6J mice and *Cars2*[+/−] mice were anesthetized with isoflurane, after which 50 μl of each CSExt and 32 μg GSSSG or vehicle were instilled intranasally twice weekly for 3 weeks. GSSSG or vehicle was instilled 6 h before CSExt administration. Mice were killed at 4 weeks after the first administration of CSExt for analysis.

## Micro-CT imaging

Micro-CT images were acquired as in the previous papers[73,74]. Briefly, mice were anesthetized via an intraperitoneal injection of medetomidine, butorphanol, and midazolam, after which they were placed in the chamber of a CT scanner for small animals (LaTheta LCT-200; Hitachi). The CT scanner was calibrated by using the standard phantom according to the manufacturer's instructions. The X-ray intensity and X-ray attenuation were adjusted each time to levels measured at the factory. X-ray scanning was performed at 50 kV and 0.5 mA. Scanning time was 6.1 s for a 192-μm-thick slice. The size of the image was $1024 \times 1024$ pixel, the field of view was 48 μm, and the resolution was $48\,\mu m \times 48\,\mu m$. Lung regions were quantitatively analyzed with LaTheta software (version 3.32). The area of each slice was reconstructed, and 3D analyses were undertaken by using Amira 5.4 software (Visage Imaging).

## Measurement of respiratory functions in mice

Lung mechanical properties, including $FEV_{0.1}$, TLV, and static compliance, were determined as described previously[35,36]. Briefly, mice were weighed, were deeply anesthetized by intraperitoneal injection of medetomidine, butorphanol, and midazolam, and then were tracheostomized. The trachea was cannulated, and the cannula was connected to a computer-controlled small animal ventilator system (flexiVent) for measurements of forced oscillations. Mice were again ventilated with an average breathing frequency of 150 breaths/min and then spontaneous breathing was suppressed by intraperitoneal injection of vecuronium bromide (1 mg/kg body weight) (FUJIFILM Wako Pure Chemical Corporation). Maximal pressure–volume (PV) loops (PVs-V 5PV: stepwise-volume regulated; PVr-V 5PV: ramp-volume regulated; PVr-P 5PV: ramp pressure regulated) were generated to obtain static compliance. All manoeuvers and perturbations were performed until three correct measurements were obtained. For flexiVent perturbations, a coefficient of determination of 0.95 was the lower limit for accepting a measurement. For each parameter, an average of three measurements was calculated and shown per mouse. The calibration procedure removed the impedance of the equipment and tracheal tube in this system.

## Morphometric analysis of lung sections

We fixed left lungs of mice at with 4% paraformaldehyde, embedded the specimens in paraffin, and sliced them with a microtome to be 4-μm thick. We processed paraffin sections for HE staining. The extent of emphysematous lesions was assessed by measuring the mean linear intercept by means of the method of Thurlbeck with some modifications[72,75]. Briefly, ten fields at ×200 magnification were randomly sampled for each mouse, followed by point counting. The total distance was divided by the number of alveolar intercepts to determine the mean linear intercept.

## Immunohistochemical analysis of lung sections

We fixed the lung tissues with 10% paraformaldehyde solution and sliced them to a thickness of 4 μm with a Leica CM1950 cryostat (Leica Microsystems). We deparaffinized lung sections through graded alcohols and washed them in PBS. After fixation with 4% paraformaldehyde in PBS for 30 min at room temperature, tissues were permeabilized with 0.1% Triton X-100 (Sigma-Aldrich) in PBS, followed by the use of a heat-induced antigen retrieval method with Tris-EDTA buffer, (10 mM Tris base, 1 mM ethylenediaminetetraacetic acid (EDTA) solution, 0.05% Tween 20, and NaOH to titrate to pH 9.0), at 120 °C for 15 min in an autoclave. Then, endogenous peroxidases were blocked with a 3% $H_2O_2$ solution (Wako Pure Chemical Industries) at room temperature for 30 min. Tissues were blocked with 10% goat serum/0.3% Triton X-100 in PBS for 30 min and were then incubated at 4 °C overnight with rabbit anti-cleaved caspase-3 antibody (1:600 dilution; Cell Signaling Technology), mouse anti-8-OHdG antibody (1:2000 dilution; Santa Cruz Biotechnology), and rabbit monoclonal anti-p21 antibody (1:1000 dilution; Abcam). Negative controls were incubated without primary antibodies. After lung sections were washed, they were incubated with goat anti-rabbit or goat anti-mouse IgG conjugated with peroxidase-labeled dextran (Nichirei) for 1 h at room temperature. The diaminobenzidine reaction was used to visualize immunopositive cells, followed by counterstaining with haematoxylin. Slides were viewed by microscopy (Olympus) and photographed with a digital camera. Haematoxylin staining was used for cell identification. Immunoreactivities to 8-OHdG and cleaved caspase-3 in five areas per lung specimen that were randomly chosen according to a previous study were semi-quantified with ImageJ software[76].

## Measurement of MMPs in BALF

Total MMP-9 amounts in BALF were determined by using Simple Western™ systems (ProteinSimple), which is a capillary-based immunoassay platform, according to the manufacturers' protocols[77]. Briefly, 10 μl of BALF was mixed with a 5 × fluorescent master mix containing SDS, DTT (40 mM), and fluorescent molecular weight standards and was then heated at 95 °C for 5 min. Samples, plus biotinylated molecular weight standards, were loaded along with blocking reagent, primary antibodies against murine MMP-9 (1:50 dilution, 5G3; Thermo Fisher Scientific), horseradish peroxidase-conjugated secondary antibodies, washing buffer, and chemiluminescent substrate into a microplate pre-filled with staking and separation matrices. We utilized completely automated western blotting. We separated proteins by electrophoresis at 375 V for 25 min, immobilized them to the capillary wall by UV cross-linking, and incubated them with primary and secondary antibodies for 30 min each. We used a CCD (charge-coupled device) camera to capture chemiluminescent signals, and the resulting image was analyzed by using Compass software (ProteinSimple) and expressed as peak intensity. To evaluate gelatinolytic activities of MMP-9 in BALF, we concentrated equal amounts of BALF tenfold by precipitation with cold ethanol and resuspended samples in 20 μl of double-distilled water. The samples were solubilized in SDS-PAGE sample buffer without 2-mercaptoethanol. We separated equal amounts of samples in 10% SDS-PAGE containing gelatin (1 mg/ml)

under non-reducing conditions. After electrophoresis, gels were soaked in zymogram renaturing buffer (Invitrogen) for 60 min and incubated in zymogram developing buffer (Invitrogen) for 18 h at 37 °C. Gels were stained with 0.4% CBB (Nacalai Tesque) for 10 min at room temperature and rapidly destained with destaining buffer (30% methanol and 10% acetic acid). Proteolytic zones appeared as clear white bands against a blue background and were scanned with the ChemiDoc 5000MP system (Bio-Rad)[23].

## Preparation of CSExt

CSExt used in the in vitro study was prepared as previously described[72]. Briefly, smoke from two cigarettes (6 mg of tar, 1.0 mg of nicotine) was bubbled through 20 ml of culture medium at a rate of one cigarette per 5 min. The solution was filtered with a 0.22-μm pore filter (Millipore); the product was said to be 100% CSExt. For the in vivo study, CSExt was prepared on the basis of a previous report with some modifications[71]. Briefly, the smoke from 10 cigarettes was bubbled through 40 ml of keratinocyte serum-free medium. The solution was filtered with a 0.22-μm pore filter and used for intranasal instillation without dilution. For all experiments, CSExt was freshly prepared before use.

## siRNA-mediated knockdown

We achieved *CARS2* knockdown via a specific single siRNA with Silencer Select siRNA for *CARS2* (Thermo Fisher Scientific). Silencer Select Negative Control #1 siRNA was the non-targeted siRNA. We used Lipofectamine RNAiMAX (Thermo Fisher Scientific) for siRNA transfection according to the manufacturer's instructions. In brief, HFL-1 cells were seeded in six-well culture plates ($2 × 10^5$ cells per well) and incubated for 24 h. For transfection, we mixed 10 pmol/well siRNA duplex or 2 μl/well Lipofectamine RNAiMAX with 200 μl of Opti-MEM (Invitrogen) in a tube. Before we added siRNA and transfection reagent solutions to the cells, we mixed solutions together and incubated them for 5 min at room temperature. Solutions were added to the cells, which were incubated for 48 h at 37 °C.

## Semi-quantification of supersulphides with SSP4

We accomplished the supersulphide imaging by using SSP4 (Dojindo Laboratories), according to our protocol[2]. Supersulphide-dependent fluorescence responses were determined with different fields of SSP4 imaging, each containing more than 100 cells cultured in multiple wells obtained from representative experiments.

## SA-β-gal assay

We measured SA-β-gal enzyme activity in HFL-1 cells with a Senescence Detection Kit (BioVision) according to the manufacturer's instructions. We detected the development of cytoplasmic blue pigment and photographed it with an inverted microscope equipped with a color CCD camera (Nikon). We determined the percentage of SA-β-gal-positive cells by counting at least 500 cells in total.

## Measurement of proinflammatory cytokines, chemokines, and fibronectin

The amounts of cytokines and chemokines in the BALF of the mouse model were measured by using a cytometric bead array kit (BD Biosciences) according to the manufacturer's instructions. The amounts of IL-6 and IL-8 in the supernatants of HFL-1 cell cultures were determined by using the DuoSet Kit (R&D Systems) according to the manufacturer's instructions. Fibronectin in the floating medium of the gel was quantified in the same way.

## Collagen gel contraction assay

We prepared collagen gels as described previously[78]. Briefly, rat tail tendon collagen, distilled water, and 4 × concentrated DMEM were combined so that the final mixture had 0.75 mg/ml collagen, with a physiological ionic strength of 1 × DMEM and a pH of 7.4. Cells were trypsinized and suspended in DMEM. Cells were then mixed with the neutralized collagen solution so that the final cell density in the collagen solution was $3 × 10^5$ cells per ml. Aliquots (0.5 ml/well) of the cell mixture in collagen were cast into each well of 24-well tissue culture plates and allowed to gel. After the gelling process was completed, normally within 20 min at room temperature, gels were gently released from the culture plates and transferred into 60-mm tissue culture dishes (three gels in each dish), which contained 5 ml of freshly prepared of serum-free DMEM with 50 μM GSSSG. Gels were treated with various concentrations of CSExt and then incubated at 37 °C in a 5% humidified $CO_2$ atmosphere for 5 days. The collagen gel contraction was quantified with a chemiluminescence imaging system (LAS-4000 mini; Fujifilm) daily. Data were expressed as percentages of the initial gel size.

## Bleomycin-induced murine lung inflammation and fibrosis model

Eight- to 10-week-old male C57BL6/J background WT and *Cars2*[+/−] mice were given an intranasal instillation of bleomycin hydrochloride (1 mg/g mouse) (Nippon Kayaku) in 50 μl of PBS or instillation of 50 μl of PBS alone on day 0; body weight was measured daily. On day 6 after bleomycin administration, mice were killed by exsanguination, and lungs were lavaged three times with 500 μl of 10 mM PBS. The lavage fluid was centrifuged at 800× *g* for 5 min. BALF supernatants were collected for biochemical analyses, and cell pellets were resuspended in 500 μl of PBS. The total cell number was counted with a Scepter™ handheld automated cell counter. Differential cell counts were obtained from cytospin-prepared slides stained with Diff-Quick. On day 21 after bleomycin administration, mice underwent micro-CT and were then killed for the study of lung fibrosis pathology. In vivo micro-CT quantification of airspace and tissue lung volumes was performed as previously reported[78].

## Morphometric analysis of lung sections in the bleomycin model

For histological evaluation of murine lung fibrosis, we stained lung sections with Masson's trichrome reagent to visualize collagen. Murine lung fibrosis was semi-quantified by means of a modified Ashcroft score as previously described[79]. For histological evaluation of lung inflammation, we stained lung sections with HE. Lung inflammation was semi-quantified according to a previous report with some modifications[80]. Briefly, we randomly chose at least ten lung areas and graded the severity of perivascular and peribronchial-peribronchiolar inflammation with points as follows: absent 0, minimal (single scattered leukocytes) 1, mild (aggregates less than 10 cells thick) 2, moderate (aggregates about 10 cells thick) 3, severe (numerous coalescing aggregates more than 10 cells thick) 4. The average points were calculated and were termed the average inflammation score.

## TUNEL staining

We used the Apoptosis in situ Detection Kit Wako according to the manufacturer's instructions (Wako Pure Chemical Corporation) for TUNEL staining.

## Statistics and reproducibility

Data are means ± s.d. of at least three independent experiments unless otherwise specified. The n number is defined as biological independent samples or animals. The n numbers in all experiments are described in Source Data file. We analyzed comparisons among multiple groups of primary cells from human subjects healthy never-smokers (HNS), healthy ex-smokers (HES), and COPD or of mice for lung injury score by using Kruskal-Wallis tests followed by Dunn's multiple comparison tests. Log rank test was performed for analysis of survival in infection experiments. We analyzed comparisons among multiple groups of all experiments except lung injury scores with a one-way ANOVA with Tukey's test or multiple *t* test, whereas Student's

$t$ test (two sided) was used for comparisons of continuous variables. Linear regression analysis was performed to assess the association between protein levels of CARS2 and parameters related to lung functions including $FEV_1\%$ predicted and $D_{LCO}/V_A$ (%), and protein levels of senescence markers, including p53 and p21 with the least squares method. We used Spearman's rank test to evaluate the strength of associations. Box centers indicate median, boundaries represent 25th and 75th percentiles, and error bars represent the maximum and minimum values in box plot. $P$ values less than 0.05 were said to be significant. We used GraphPad Prism 9 (GraphPad Software) for statistical analysis.

### Reporting summary
Further information on research design is available in the Nature Portfolio Reporting Summary linked to this article.

### Data availability
All data generated and analyzed in this study are included in this article and its Supplementary Information files. The mass spectrometry proteomics data have been deposited to the ProteomeXchange Consortium via the PRIDE partner repository with the dataset identifier PXD043607.

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

## Acknowledgements

We thank J.B. Gandy for her editing of the manuscript and evaluating the concepts and terminology of the paper with regard to the understanding by non-specialist readers. Thank are also due to T. Miura, T. Watanabe, M. Takahashi, Q. H. Z. Abidin, and N. Hassan for their technical assistance in our study. This study was supported in part by Grants-in-Aid for Scientific Research [(S), (A), (B), (C), Challenging Exploratory Research] from the Ministry of Education, Culture, Sports, Sciences and Technology (MEXT), Japan, to T. Akaike (18H05277, 21H05263 and 22K19397), H. Sugiura (20H03684 and 17H04180), H. Motohashi (21H05258 and 21H05264), T. Sawa (21H05267), H. Oshiumi (19H03480), M. Morita (23K06145), T. Ida (20K07306), T. Matsunaga (22K06893), S. Ogata (23K14333), M. Jung (23K14341), T. Numakura (18K15941), and Y. Kyogoku (20K17208); Japan Science and Technology Agency (JST), CREST Grant Number JPMJCR2024, Japan to T. Akaike; and a grant from the Japan Agency for Medical Research and Development (AMED), the Practical Research Project for Allergic Diseases and Immunology (Research on Allergic Diseases and Immunology) (18ek0410036h0003) and Grant Number JP21zf0127001, Japan, to H. Motohashi and T. Akaike.

## Author contributions

T.M, H. Sano, K.T., and M.M. contributed animal model experiments of infectious diseases and COPD. H. Sano, S. Yamanaka, Y.S., T. Ichikawa, and T. Numakura performed influenza virus infection and supersulphide treatment experiments. T.M., T.S., H.O., M.M, T. Ida, M.J., S.O., and S. Yoon performed SARS-CoV-2 experiments. T. Ida, S.O., and A.T. performed the supersulphide metabolome and proteome. K.T., S. Yamanaka, Y. K., T.T., T. Numakura, and T. Ichikawa contributed to the recruitment of COPD patients, informed consent of patients. H. Sano, T. Ichikawa, T. Numakura, and Y.S. collected EBC samples and informed consent of patients with COVID-19. M.M. generated the knockout mice for this study. M.Y., A.K., and N.F. analyzed chronic lung disorders models. T. Nakabayashi and L.K. performed calculations of supersulphide structures. S.K. supported MHV infection system. K.I. and S.W. analyzed the protein structural analysis with the docking model. P.N. gave technical advice and valuable interpretations of supersulphide chemistry and the experimental results. T.A., H. Sugiura, M.I., F.-Y.W., and H.M. designed the study, guided the experimental design and analysis, and wrote the manuscript with input from all authors.

## Competing interests

The authors declare no competing interests.

## Additional information

[1]Department of Environmental Medicine and Molecular Toxicology, Tohoku University Graduate School of Medicine, Sendai 980-8575, Japan. [2]Department of Respiratory Medicine, Tohoku University Graduate School of Medicine, Sendai 980-8574, Japan. [3]Analytical and Measuring Instruments Division, Shimadzu Corporation, Kyoto 604-8511, Japan. [4]Bio-Structural Chemistry, Graduate School of Pharmaceutical Sciences, Tohoku University, Sendai 980-8578, Japan. [5]Laboratory of Biomedical Science, Department of Veterinary Medical Science, Graduate School of Agricultural and Life Sciences, The University of Tokyo, Tokyo 113-8657, Japan. [6]Institute of Multidisciplinary Research for Advanced Materials, Tohoku University, Sendai 980-8577, Japan. [7]Department of Molecular Immunology and Toxicology, National Institute of Oncology, Budapest 1122, Hungary. [8]Department of Microbiology, Graduate School of Medical Sciences, Kumamoto University, Kumamoto 860-8556, Japan. [9]Department of Immunology, Graduate School of Medical Sciences, Kumamoto University, Kumamoto 860-8556, Japan. [10]Department of Modomics Biology and Medicine, Institute of Development, Aging and Cancer, Tohoku University, Sendai 980-8575, Japan. [11]Department of Gene Expression Regulation, Institute of Development, Aging and Cancer, Tohoku University, Sendai 980-8575, Japan. [12]These authors contributed equally: Tetsuro Matsunaga, Hirohito Sano, Katsuya Takita, Masanobu Morita. [13]These authors jointly supervised this work: Tadahisa Numakura, Tomoaki Ida, Mitsuhiro Yamada. ✉e-mail: sugiura@rm.med.tohoku.ac.jp; hozumim@med.tohoku.ac.jp; takaike@med.tohoku.ac.jp

