## [Peer review File · Nature Communications]

REVIEWER COMMENTS

Reviewer #1 (Remarks to the Author):

The authors studied the protective effect of supersulfides (per- and poly-sulfides) produced by CARS/CPERS and exogenously applied GSSSG on viral airway infection, using heterozygous CARS/CPERS mice. The lung damage caused by oxidative stress and inflammation was augmented in model mice and suppressed by the application of GSSSG. Based on these observations they suggested that supersulfides produced by CARS/CPERS and GSSSG have therapeutic potential in various chronic lung diseases. There are concerns and comments as shown below.

1. In addition to the activity as cysteinyl t-RNA synthetase, CARS2 has another activity as cysteine persulfide synthase (CPERS). Since the present study aims to clarify the effect of supersulfides on lung diseases, mice or cells suppressed only CPERS activity, *Cars2*^{AINK/+}, should be used not only in Fig. 4 but in other figures experiments.
2. Authors previously reported that CBS and CSE produce cysteine persulfide (Proc Natl Acad Sci USA 111, 7606-7611, 2014), and others showed that 3-mercaptopyruvate sulfurtransferase (3MST) and sulfur quinone oxidoreductase also produce those persulfurated molecules (Sci Rep 7: 10459, 2017; J Biol Chem 290, 25072-25080, 2015). For example, the levels of bound sulfane sulfur, which includes S-sulfurated molecules, are half in 3MST knockout mice compared to the wild type. It is to be discussed.
3. The word 'supersulfide' is created to indicate persulfide and polysulfide. It is not the correct chemical nomenclature and very confusing. In line 83 'reactive persulfide' is used.
4. Not only GSSSG but other per- and poly-sulfides such as diacyl trisulfide (DAT) and elemental sulfur S₈ have been proposed for their anti-COVID19 (Br. J. Pharm. 177, 4931-4941, 2020). It should also be discussed.
5. Lines 77 and 78. Nomenclature is confusing. If glutathione persulfide represents GSSH, glutathione trisulfide should be GSSSH but not GSSSG, which may be called 'oxidized glutathione trisulfide'. GSSG is usually called oxidized (form of) glutathione. Line 219 also the same. '...oxidized persulphide, i.e. GSSSG'.
6. Lines 79-83. The description is not correct. H₂S S-sulfurates S-nitrosylated- and S-sulfenylated cysteine residues (Cell Metab 30:1152-1170.e13, 2019; Nat Chem Biol 11, 457-464, 2015).
7. In experiments in lung premature aging, they used siRNA for CARS2 with which both CARS2 and CPERS activities are suppressed. Cells only suppressed CPERS activity should be used.
8. Cytoprotective effect of H₂S, H₂S_n and sulfite have been reported, and these previous studies are to be discussed (FASEB J 18, 1165-1167, 2004; FEBS Lett 587, 3548-3555, 2013; Br J Pharm 176, 571-582, 2019).

9. The description in line 315 is not correct. Several enzymes are involved in the metabolism of H₂S, polysulfides, thiosulfate, sulfite up to sulfate, including sulfur quinone oxidoreductase, sulfur dioxygenase, and rhodanese.

10. Line 337. The interaction of H₂S and NO produces nitrosopersulfides including H₂S_n, HSNO, HNO, and HSSNO (Biochem Biophys Res Commun 343, 303-310, 2006; Nat Commun 5: 4381, 2014; Proc Natl Acad Sci USA 112, E4651-E4660, 2015; Sci Rep 7: 45995, 2017). These previous studies are to be discussed.

11. Lines 351-352. H₂O₂ S-sulphenylates cysteine residues which are S-sulfurated by H₂S (Cell Metab 30:1152-1170.e13, 2019).

12. Lines 352-353. Although reaction between H₂S and H₂O₂ is poor, S-sulphenylated protein reacts two orders of magnitude faster with H₂S than with glutathione (J. Biol. Chem 290, 26866–26880, 2015)

Reviewer #2 (Remarks to the Author):

In this study, the authors aimed to demonstrate the airway protective role of supersulphides and their therapeutic potential. The topic is interesting, and the authors have generated extensive data to support their conclusions. However, I am not convinced about the main conclusion (i.e. the airway protective role of supersulphides and their therapeutic potential) based on the data provided. Particularly, an extremely high concentration of supersulphide (1 mM GSSSG or 300 uM GSSG, GSSSG, ...) was used in their experiments. This concentration (1 mM or 300 uM) is physiologically irrelevant, as the detected concentrations of supersulphides are only in nM or pmol/mg. Supersulphides are very reactive compounds that could non-specifically react with many other proteins and, thus, non-specifically and irreversibly inhibit numerous proteins in the body. The Inhibitory activity data shown in Fig. 2a and b could be due to the non-specific reactivity with many proteins. Supersulphides at such a high concentration would make a lot of unexpected damages in the body due to their non-specific reactivities. In addition, the observed differences in the supersulphide levels between healthy subjects and COVID patients or between WT and Cars2^{+/-} mice do not necessarily support the protective effects of supersulphides. Would a supersulphide at the observed concentrations (in nM or pmol/mg) be protective against any virus?

Reviewer #3 (Remarks to the Author):

General comments:

Sano et al. report that the innate defense functions of supersulphides that protect the lung and airways against viral infections, COPD, pulmonary fibrosis, and ageing. This is interesting paper. They showed the beneficial roles of supersulphides in various diseases; however, they did not address sufficiently the precise mechanisms of these beneficial effects in these diseases.

Major comments:

1. Anti-influenza defense: Authors showed a highly beneficial and protective effect of supersulphides in the influenza model. However, the mechanisms of suppressive effect of supersulphides on the growth of influenza virus in vivo in the lung (Fig. 1d) should be addressed.
2. Anti-COVID activities of supersulphides: Authors should demonstrate the protective effect of supersulphides against COVID-19 in vivo using mice.
3. COPD: In figure 3, authors used elastase-induced emphysema model. However, this mouse model is not appropriate for COPD model. The mouse model by CSE inhalation for at least 3 months is beneficial.
4. Emphysema in CPER-deficient mice: In Figure 4, authors showed that supersulphides produced by CARS2/CPERS contribute to protection of COPD pathogenesis through the strong anti-inflammatory and antioxidant effects of supersulphides. Although authors showed various biomarkers for antioxidative stress, they should also show the results of measurement for oxidative stress.
5. Supersulphides in lung premature aging: In Figure 5, authors described that supersulphides produced by CARS2 limit the progression of cellular senescence and prevent lung ageing by virtue of their antioxidant function and contribution to mitochondrial energy metabolism. However, they only showed that aged mice are more susceptible to CSE instillation. They should show direct evidences that these protective effects were due to antioxidative function and due to contribution to mitochondrial energy metabolism in aged mice.

Reviewer #4 (Remarks to the Author):

In this manuscript, the authors extend their earlier findings—that endogenous supersulphide production is reduced in COPD and related chronic inflammatory airway diseases—and test the hypothesis that supersulphides protect against negative outcomes of diverse inflammatory airway diseases, including that caused by influenza virus, SARS-CoV-2 (the causative agent of COVID-19), chronic obstructive pulmonary disease (COPD), idiopathic pulmonary fibrosis (IPF), and ageing. By using a newly developed knockin mouse model with a mutation in a cysteine persulphide synthase enzyme (*Cars2*AINK/+), they demonstrated that *Cars2* is the major contributor to supersulphide biosynthesis in the lung. In addition, they used *Cars2*AINK/+ mice, a previously developed knockout mouse model (*Cars2*-/+), and/or treatment with exogenous supersulphide (GSSSG), to show that supersulphides protect against lethal

disease outcomes in a mouse model of severe influenza virus disease, impair SARS-CoV-2 protease activity in a cell-free in vitro system, and protect against negative outcomes caused by chronic lung diseases, such as COPD and IPF. They also developed an approach to measure supersulphide metabolites in the breath of COVID-19 patients (“breath-omics”), and generated some data suggesting that supersulphides may be highly expressed and/or heavily metabolized in humans with COVID-19.

Overall, their approach is comprehensive, they developed a new reagent that may be useful to the research community (Cars2AINK/+ mice), they developed a new method to measure metabolites in exhaled breath that might be useful as a diagnostic tool, and they identified a potential avenue for development of a broad-spectrum therapeutic to treat diverse inflammatory lung diseases. The concept of a single type of treatment to mitigate risks associated with acute or chronic lung inflammation of diverse origins is alluring, and the potential long-term benefits to human health could be immense (e.g., by reducing chronic disease burden in ageing populations; or by reducing morbidity and mortality in epidemics or pandemics caused by certain emerging respiratory pathogens). In this regard, the data presented in this manuscript have the potential for a broad impact in biomedical research and translational diagnostic or therapeutic development. On the other hand, the manuscript has some important limitations that need to be considered carefully:

- i. Influenza mouse experiments are solid and generally convincing, with Cars2-/+ mice exhibiting an increase in disease severity and GSSSG-treated mice exhibiting a decrease in disease severity after influenza virus infection (note: higher viral genomic RNA and inflammatory cytokine levels in the lung are associated with more severe disease). In all experiments, infections were performed with a single dosage of a common, mouse-adapted laboratory strain of influenza (PR8). Influenza mouse experiments could be improved by using a clinically relevant influenza virus strain (e.g., the 2009 pandemic H1N1 virus causes severe disease in mice without adaptation), determining the lethal dose 50 (LD50) in Cars2-/+ mice or GSSSG-treated mice relative to appropriate controls, and using virus titration as a readout of virus growth. Also, it would be interesting to assess whether GSSSG treatment can reduce influenza growth in cell culture, to confirm the proposed antiviral effect.
- ii. Currently, SARS-CoV-2 experiments are restricted to examining the effects of GSSSG on viral protease activity (in vitro, cell-free assay) and the production of viral genomic RNA (in cell culture) as a proxy for virus growth. In vitro data strongly suggest a role for GSSSG in the inhibition of viral protease activity (although this remains to be confirmed in cell-based or animal models); however, the effects of GSSSG on viral genomic RNA production in cells are modest, suggesting potentially limited antiviral activity. The SARS-CoV-2 cell culture experiment could be improved by sampling GSSSG-treated cultures in a growth curve (rather than a single time point) and by using virus titration as a readout of virus growth. In addition, given the authors’ emphasis of supersulphides as a potential treatment for COVID-19, it is essential to show the effect of supersulphides on COVID-19 severity in an animal model (e.g., GSSSG treatment in hamsters infected with SARS-CoV-2).
- iii. The authors report that supersulphide metabolites might be useful as biomarkers of COVID-19 (or other lung disease) severity. However, this conclusion cannot be made given the available information and analyses performed: specifically, the number of patients analyzed is low (12 per group, healthy or infected), with only a single patient designated as having exacerbated disease; no information about sampling time points or disease severity upon sampling are reported; it appears that all COVID-19 patients, regardless of disease stage or symptom severity were analyzed as a single group; and statistical

procedures designed to evaluate potential biomarkers were not employed. These limitations substantially weaken the impact of the clinical data. To improve this analysis, the authors need to recruit more participants, stratify participants by disease severity/outcome, and apply a more rigorous statistical analysis designed specifically for biomarker identification.

iv. The writing style is imprecise and at times difficult to follow. In particular, the flow of the results section does not seem logical and the figure legends are frequently inadequate to describe the data shown in the figures. Significant revisions are needed to improve organization and flow, clarity of experimental descriptions and the associated results, and the appropriate inclusion/exclusion of inferences that can and cannot be made from the available data.

Specific Comments

1. Line 1: Suggest changing the title; specifically,

a. "Supersulphide airway protection" should be "Supersulphide-mediated airway protection" or something similar.

b. It would be better to not use a colon.

c. "Viral and inflammatory lung disease" is not a good summary description of the disease models examined in this study. Both influenza disease and COVID-19 are driven by inflammation, like disease in the COPD/IPF/ageing models. The authors need to come up with another phrase that better encompasses the non-infectious disease models (e.g., chronic lung diseases).

2. The Abstract needs substantial revision to increase precision/accuracy and to better reflect the findings of the study. Specifically,

a. Line 58 (abstract): "...viral airway infections, i.e. influenza and COVID-19..."; it would be more accurate to say "i.e. influenza and SARS-CoV-2," since COVID-19 is not a virus, but rather, the disease caused by SARS-CoV-2.

b. Lines 60-61 (abstract): "This strong anti-SARS-CoV-2 defense was evident with the supersulphide donor glutathione trisulphide (GSSSG)." The data presented for SARS-CoV-2 include an in vitro assay to measure the effect of GSSSG on viral protease activity and the effect of GSSSG on production of genomic RNA in cell culture. It is not clear whether the inhibitory effect of GSSSG on viral protease activity, observed in vitro, is similar in the context of viral infection. Moreover, in infected cells, GSSSG treatment caused a modest reduction in viral genomic RNA levels (~80%) at the highest GSSSG concentration tested, and the effect on infectious virus output was not tested. Put simply, the SARS-CoV-2 data is not the strongest data in the manuscript and framing it this way in the abstract imparts undue importance.

c. Lines 61-62 (abstract): "We observed a similar beneficial effect of supersulphides in vivo in influenza pneumonia in mice..." The SARS-CoV-2 and influenza experiments were completely different, so it is incorrect to say that the effect of supersulphides between systems was similar.

d. Lines 64-65 (abstract): "Enhanced endogenous supersulphide production also occurred in human airways in COVID-19, as identified by breath sulphur-omics we developed herein." The reported experiment identified increased levels of sulphur metabolites in the breath of COVID-19 patients. Higher

levels of sulphur metabolites is not direct evidence of increased supersulphide production, and could have other explanations (e.g., increased metabolism of existing supersulphides).

e. Lines 67-68 (abstract): "...were greatly mitigated by supersulphides, either endogenous (CARS2/CPERS) or exogenous (GSSSG)." CARS2/CPERS is not an endogenous supersulphide, as implied by the text, but rather the enzyme that produces supersulphides.

f. Lines 70-71 (abstract): "...which seemingly affect COVID-19 prognosis and even ageing." The placement of this phrase is odd and may warp the intended meaning of the sentence.

3. Introduction:

a. Lines 85-89 (introduction): "More important, oxidative stress associated with airway inflammation has been implicated in the pathogenesis of influenza, COVID-19, and chronic lung disorders including [COPD], emphysema, [IPF] and even lung ageing14-18."

i. It is not clear why this sentence is more important than the preceding information.

ii. The references at the end of this sentence do not include any papers that discuss oxidative stress in influenza or SARS-CoV-2 infections.

b. Line 89 (introduction): "Our preliminary studies" should be "Our previous studies."

c. Line 94 (introduction): "(CARs)" should be "(CARS)"

d. Line 100 (introduction): "We report here the innate but highly evolved defense functions of supersulphides..." Highly evolved relative to what?

e. Lines 107-110 (introduction): Cars2-/+ mice need a reference.

4. Lines 120-121 (results): "...one novel aspect of supersulphides that we identified is their antiviral effects on influenza and COVID-19 viruses (Figs. 1 and 2)."

a. Technically, no antiviral effect has been established in this work. In Fig. 1, the authors show that viral genomic RNA levels are increased (2-3-fold) in Cars2-/+ mice after influenza virus infection relative to WT mice, but they did not report the effect on infectious virus output. In Fig. 2, the authors show that GSSSG treatment caused a modest reduction in viral genomic RNA levels (~80%) at the highest GSSSG concentration tested (in cell culture), but they did not report the effect on infectious virus output.

b. "COVID-19 virus" would be more accurately described as SARS-CoV-2.

c. The authors should cite Extended Data Fig. 2 with Figs. 1 and 2 at the end of this sentence.

5. Lines 123-125 (results): "We found that endogenous supersulphide formation in lungs was much lower in Cars2-deficient (Cars2-/+) mice than in WT mice throughout the course of infection (Extended Data Fig. 3)."

a. This statement is not true for most of the supersulphide metabolome shown in Extended Data Fig. 3.

b. In a few cases (CysSSH, HS-, HSS-), there does appear to be a significant difference in supersulphide levels in WT and Cars2+/- mice at the 0 day time point. However, additional comparisons between WT day 0 and Cars2+/- day 3 or day 7 are not correct—comparisons should be made between WT and

Cars2^{+/-} mice at the same time points, and may need to be normalized to the expression level observed at day 0.

c. Also, it is notable that there are different patterns of expression for different supersulphide metabolites. Does this provide insight into the pathways responsible for supersulphide metabolite production (or use) in the various conditions?

6. Lines 129-131 (results): "Of great importance is that growth of influenza virus in vivo in the lung significantly increased in Cars2^{+/-} mice compared to WT mice." As already noted, the authors have not established an antiviral effect, since they did not examine infectious virus output.

7. Lines 132-134 (results): "Also, supersulphides produced by Cars2 are likely to abolish not only cytokine storm-like excessive inflammatory responses but also nitrative and oxidative stress (Fig. 1; Extended Data Fig. 2a-e), all of which are critically involved in influenza pathogenesis, as we reported earlier²²⁻²⁶." Fig. 1g and 1n suggest that supersulphides can reduce the production of inflammatory cytokines (although inflammation is not abolished). However, there do not appear to be any data to support the idea that supersulphides also mitigate oxidative and nitrosative stress in influenza infections.

8. Lines 136-138 (results): "...influenza virus infection per se reduced endogenous supersulphide production, which may aid viral propagation and inflammatory responses in virus-infected lungs (Extended Data Fig. 3)." As already stated above, this is true only for a subset of the supersulphides presented in Extended Data Fig. 3.

9. Lines 138-141 (results): "In fact, intranasal administration of GSSSG markedly reduced all pathological and inflammatory consequences, including survival, in influenza-infected WT animals. This result suggests that supersulphides supplied exogenously compensate for impaired antiviral and protective functions of supersulphides that are depleted by oxidative stress and excessive inflammatory reactions..." The effect of GSSSG on production of viral genomic RNA is not shown; thus, there is no measure of antiviral activity shown for GSSSG treatment.

10. Figure 1 and Figure 1 legend:

a. This figure could be improved by making the two halves (a-g and h-n) more consistent:

- i. Show viral genomic RNA levels for both.
- ii. Show the same immune cell subset numbers for both.
- iii. Arrange the data in the same order.

b. In the legend, the authors need to indicate mouse age and background strain and how the control (uninfected) animals were treated (i.e., were they inoculated with PBS, media, nothing?)

c. Were the presented experiments performed more than once, or are the data shown from a single experiment?

d. Were the same mice used for Fig. 1a and 1b? If so, why not show the body weights through day 14?

e. Fig. 1c:

- i. The lung injury scores seem very low, and description provided in the Methods section is insufficient to describe how these scores were calculated.
- ii. The statistical comparison between mock/uninfected Cars2+/- mice and influenza-infected WT mice doesn't make sense.
- iii. Typically, pathology scores are expressed as ordinal numbers (e.g., a score of 1 to 5 based on disease severity) for a given parameter. As such, it is not correct to use mean and standard deviation to represent data from a group of animals. Rather, data should be expressed by using medians and confidence intervals, and statistically tested by using non-parametric methods.
- f. Fig. 1d: The use of light blue and pink in this panel (for day 3 tissues) is confusing since it was used in Fig. 1a-1c to represent mock/uninfected animals.
- g. Fig. 1e and 1f: The differences between groups could be better represented if the top values on the Y-axes were reduced.

11. Extended Data Fig. 2 and legend:

- a. Panel a is fuzzy.
- b. Panel d—the statistical test result bar seems to be out of place.
- c. Panel e—Is there a lighter exposure to better highlight the differences?

12. Line 149 (results): The authors should use either 3CLpro or Mpro, but not both abbreviations in the manuscript text.

13. Line 158 (results): "...GSSSG potently inhibited SARS-CoV-2 propagation: virus replication was appreciably reduced..." As already stated, the authors have not measured virus replication/propagation, since they have not quantified the production of infectious virus.

14. Lines 171-174 (results): "Of great importance is a marked increase in levels of sulphur metabolites, particularly sulphite and HS2O3- in the EBC of a patient (as indicated by arrows) just before manifesting a progressive disease caused by severe viral pneumonia."

- a. In Fig. 2g, which metabolite is sulphite?
- b. How many supersulphide metabolites were detected? If more than the three shown in Fig. 2g, the authors should probably provide the additional data in a supplementary figure or table.
- c. While this finding is interesting, its importance is not clear because it is a single data point, and other patients with non-exacerbated COVID-19 have higher levels of some sulphur metabolites.

15. Lines 175-177 (results): "The present... findings... suggest... the application of these sulphur metabolites as excellent biomarkers for risk assessment of severe illness caused by coronavirus."

a. The authors have not stratified their data by disease severity, so there is no way to know whether the sulphur metabolites may be used to differentiate different severity groups.

b. A rigorous statistical analysis for biomarker identification has not been performed; therefore, the authors cannot state one way or another if sulphur metabolites might be useful as biomarkers.

c. Put simply, the finding is overstated based on the given data, and the text must be revised in the absence of additional analyses confirming the findings.

16. At several points in the manuscript the authors use the phrases “sulphur-omics” or “breath-omics” (and once “sulphur breath-omics”). It is not always clear whether these phrases refer to the same or different analyses. Also, are the authors really performing omics analyses, or would their work be better characterized as targeted metabolomics?

17. Figure 2 and Figure 2 legend:

a. Fig. 2a: Somewhere, there should be some explanation of Ebselen.

b. Fig. 2e: It is not clear what is being represented in this panel. More explanation is needed in the legend and/or the text of the results.

c. Fig. 2f: Please include error bars.

d. Fig. 2g and 2h:

i. From the information available, it seems there may be up to 4 severity groups in this analysis: healthy, COVID-19/exacerbated, COVID-19/non-exacerbated, and recovered. However, in panel g, the only comparison is between healthy and COVID-19 patients. Would it be better to reanalyze the data with COVID-19 patients further subsetted based on disease severity? How many of the COVID-19 patients had exacerbated disease, and how many were recovered? What time points were the samples collected (with respect to symptom onset)?

ii. Can the authors infer anything about active pathways based on the supersulphide metabolites that were detected?

18. Lines 182-183 (results): “COPD is one of the highest risk conditions ever documented...” This statement is not very clear. From the reference at the end of the sentence, it appears to be referring to risk for severity of COVID-19. If so, the authors should state this more explicitly.

19. Fig. 3c is too fuzzy.

20. The authors mention the use of Cars2AINK^{-/-} mice on line 199, but the generation of these mice is not described until the next section (starting from line 226).

21. Lines 321-325 (discussion): “Because COPD is the highest risk condition known and IPF is the most serious complication reported, supersulphide-mediated airway protection may greatly benefit GSSSG therapeutics for COVID-19, which should thus be a superior approach to simple antiviral chemotherapy now being developed worldwide.” It is not totally clear what the authors are trying to say here, but perhaps they mean that (i) individuals with COPD are at high risk for severe COVID-19, (ii) IPF is one of the most severe complications of COVID-19, (iii) GSSSG can ameliorate COPD and IPF (and maybe COVID-19), so (iv) GSSSG may be a good therapeutic approach for COVID-19 treatment.

a. The authors need to revise this quote to state it more clearly.

b. It is important to note that the effect of GSSSG treatment has not been tested in an animal model of COVID-19, and the effects of GSSSG on COVID-19 outcome (including the risk of individuals with COPD and the development of IPF) in vivo is completely unknown.

22. Line 346 (discussion): Please reword the phrase “seminal published papers.”

23. Methods:

a. Line 768 (methods): What were BEAS-2B cells used for?

b. Lines 792-800 (methods): The description of histopathology analysis is insufficient. At a minimum, please add

i. The references used to design the analysis

ii. The specific parameters that were scored

iii. The scoring scale and the definition of each number in the scale

iv. The number of sections scored per animal (and if more than one, how far apart from each other?)

v. The calculation of the final score

vi. The number of pathologists that analyzed the sections and whether they were blinded

c. Line 826 (methods): HFL-1 cells were not mentioned in the earlier cell culture section.

d. Lines 859-868 (methods): The authors should explicitly state that the substrate contains a moiety (MCF) that gives fluorescent signal when released from the substrate.

e. Line 917 (methods): “EBC was frozen quickly...” How was this done?

Responses to the comments from the reviewers

Reviewer #1 (Remarks to the Author)

General comments: The authors studied the protective effect of supersulfides (per- and polysulfides) produced by CARS/CPERS and exogenously applied GSSSG on viral airway infection, using heterozygous CARS/CPERS mice. The lung damage caused by oxidative stress and inflammation was augmented in model mice and suppressed by the application of GSSSG. Based on these observations they suggested that supersulfides produced by CARS/CPERS and GSSSG have therapeutic potential in various chronic lung diseases. There are concerns and comments as shown below.

Response: We thank the reviewer for providing valuable comments. We have addressed the reviewer's concerns in the following point-by-point responses, with each comment and response numbered sequentially after C and R, respectively.

The work for this revision was time-consuming, indeed, because it took time to carry out many in vivo experiments mostly by producing animal models of for the SARS-CoV-2 infections and human clinical investigation. However, we believe that our responses and revision are very successful made below, which are supported by several new data obtained from thorough additional experiments and appropriate modifications of the text accordingly.

Comment:

C.1: In addition to the activity as cysteinyl t-RNA synthetase, CARS2 has another activity as cysteine persulfide synthase (CPERS). Since the present study aims to clarify the effect of supersulfides on lung diseases, mice or cells suppressed only CPERS activity, *Cars2^{AINK/+}*, should be used not only in Fig. 4 but in other figures experiments.

Response:

R.1: Following this reviewer's suggestion, we carried out SARS-CoV-2 infection in *Cars2* mutant mice, i.e., AINK mutant mice, crossed with ACE2 transgenic mice (new Fig. 3). We developed these ACE2 and AINK double-mutant (ACE2-Tg::*Cars2^{AINK/+}*) mice to produce SARS-CoV-2 infection in mice because the wild-type mice that do not possess ACE2 originally had to be genetically engineered to express ACE2 so that they can be fully susceptible for SARS-CoV-2 in the airway.

The results showed that the lethal effect of intranasal infection with SARS-CoV-2 was significantly higher in ACE2-AINK-mutant mice than in the ACE2-transgenic mice (new Fig. 3). The enhanced pathogenesis was found to be caused by the accelerated viral growth in the lungs of ACE2-AINK-mutant mice compared with that in ACE2-transgenic mice. These data suggest that persulfides or related supersulfides generated by CARS2/CPERS contribute to the anti-SARS-CoV-2 host defense in vivo. The current interpretation is further supported by the following new findings as well as those originally presented.

1. The persulfide production of ACE2-AINK mutant mice was decreased compared with that of the wild-type mice during SARS-CoV-2 infection (new Supplementary Fig. 8a; related statements in the text, P. 8. L. 204–209).

2. The SARS-CoV-2 replication was significantly higher, when endogenous persulfide formation was attenuated by knocking down the CARS2/CPERS expression in the VeroE6/TMPRESS2 cells in culture, which was nullified by the exogenously added persulfide donor GSSSG (new Fig. 2; Supplementary Fig. 4-6; and related statements in the text, P. 6, L. 150–P. 7, L. 175). Notably, the COVID-19 hamster model unequivocally revealed the antiviral effect of GSSSG (new Fig. 3e-g,

Supplementary Fig. 8b and related text, P. 8, L. 210–218).

3. The suppression of the viral replication is most likely due to inhibition of the viral thiol proteases by supersulfides, as shown in the original version of this manuscript, as well as through the alterations of viral structural proteins such as S-protein by supersulfide-induced modifications of their disulfide bridges, which was newly observed in the current work (new Fig. 4, Supplementary Fig. 9-13, and related description, P. 8, L. 220–P. 10, L. 276) and also reported recently (Ref. 31: Shi et al., PNAS 2022).

Taken together, it is concluded that CARS2-derived supersulfides have potent anti-SARS-CoV-2 activity that mediates innate host defense against COVID-19.

Other physiological and beneficial functions of supersulfides generated by CARS2 have been verified by several experimental models conducted herein for a COPD model induced by the cigarette smoke extract, where various phenotypic changes observed in CARS2-het-KO mice are almost completely reversed by the administration of GSSSG as we had demonstrated in the original version of this manuscript (now shown as new Fig. 9).

Meanwhile, we would like to emphasize, and hope this reviewer would realize, that the magnitude of reduced supersulfides is smaller in AINK mice (be it noted that only heterozygotes are available) than in CARS2-het-KO mice (see a brief comment on P. 8, L. 204-209). This is because only one of the two catalytic sites for persulfide biosynthesis (see Ref. 26) is eliminated in AINK mice, whereas CARS2-het-KO mice are deficient in both sites, thereby persulfide production should be much more attenuated in the latter than in the former. It is, therefore, reasonable to assume that the phenotypic changes dependent on supersulfides produced by CARS2 are less evident in AINK mice than in CARS2-het-KO mice. This is the main reason why we still need to employ CARS2-het-KO mice instead of AINK mice, together with the pharmacological intervention using persulfide donors such as GSSSG, as we are doing here.

Comment:

C.2: Authors previously reported that CBS and CSE produce cysteine persulfide (Proc Natl Acad Sci USA 111, 7606-7611, 2014), and others showed that 3-mercaptopyruvate sulfur transferase (3MST) and sulfur quinone oxidoreductase also produce those persulfurated molecules (Sci Rep 7: 10459, 2017; J Biol Chem 290, 25072-25080, 2015). For example, the levels of bound sulfane sulfur, which includes S-sulfurated molecules, are half in 3MST knockout mice compared to the wild type. It is to be discussed.

Response:

R.2: We have just published a paper entitled as “Synthesis of sulfides and persulfides is not impeded by disruption of three canonical enzymes in sulfur metabolism”, which is cited in this revision (Ref. 53). In this work, we have shown that not only single KO mice for either CBS/CSE/3-MST, but also CBS/CSE/3-MST triple KO mice, do not show appreciable changes in many sulfur metabolites. These data indicate that, although three canonical enzymes must be somehow responsible for supersulfide biosynthesis, CARS2 is the major source of supersulfide production and thereby could compensate for the impaired sulfur metabolism even after elimination of these three enzymes.

These findings and interpretation have been now incorporated in the revised manuscript: P, 16, L. 487–P. 17, L. 492. Meanwhile, although we do understand well that SQR is an important enzyme that catalyzes sulfide to produce persulfide, as we reported earlier (Ref. 37, 38), but will not mention it herein, because this topic is somehow out of scope of our current work.

Comment:

C.3: The word 'supersulfide' is created to indicate persulfide and polysulfide. It is not the correct chemical nomenclature and very confusing. In line 83 'reactive persulfide' is used.

Response:

R.3: The comment from this reviewer is correct. "Supersulfide" is not a chemical term, but rather a conceptual nomenclature. We currently define "SUPERSULFIDES" as hydropersulfide (RSSH) species and polymeric sulfurs with sulfur catenation (RSS_nR , $n > 1$, R = hydrogen or alkyl, or cyclized sulfurs), which are recognized as universal bioactive metabolites formed physiologically in all organisms, as this reviewer well recognizes it. Moreover, although it is still preliminary or yet unpublished issues, there is a huge amount of inorganic sulfur allotropes most typically such as cyclic octa-sulfur (S_8) generated physiologically in mammalian cells and tissues including the human specimens. To our great surprise, not only S_8 but also hetero-cyclic sulfur-containing hydrocarbons are found to be appreciably generated endogenously in many organisms (our unpublished observation); besides, not to mention the protein-bound polysulfides, diverse polymeric sulfur molecules complexed or liganded with metals such as zinc anion (Zn^{2+}) are identified to be present in some specific proteins functioning as enzymes, e.g., GSNOR/ADH5, on which extremely novel work is currently under minor revision in *Sci. Adv.*, and its manuscript is also attached to this review. In such conceptual context, therefore, supersulfides should include all these sulfur compounds and allotropes, which cannot be fully covered by other popular terms such as reactive sulfur species (RSS), sulfane sulfurs, and persulfides/polysulfides. Nevertheless, we will regularly use the terms such as persulfides, etc., for example, cycteine- or glutathione-based persulfides such as CysSSH and GSSH, which are more like chemical nomenclature, as long as the scientific merit of their use is justified and warranted in the chemical context of the usual narratives in scientific documents and literatures.

Regarding this terminology issue, we have added some clearer statements and definitions in the revised text: please see, P. 4, L. 72–76; Supplementary Fig. 1 legend.

Comment:

C.4: Not only GSSSG but other per- and poly-sulfides such as diacyl trisulfide (DAT) and elemental sulfur S_8 have been proposed for their anti-COVID19 (*Br. J. Pharm.* 177, 4931-4941, 2020). It should also be discussed.

Response:

R.4: We thank the reviewer for letting us know the information, which we have been aware of. As this reviewer suggested, we cited the papers, therefore, and also include relevant discussion in the new text by adding other sulfide-related topics on COVID-19, as follows (also see P. 16, L. 460–471 in the new text) .

"There are previous papers proposing the potential protective functions of sulphide against COVID-19^{46,47}. Earlier reports only dealt with a possible beneficial effect of hydrogen sulphide on the COVID-19 pathogenesis, where sulphide could be either introduced pharmacologically with sulphur-containing compounds such as diaryltrisulphide and S_8 or endogenously generated during the host defense responses. The cytoprotective functions of various sulphide-related compounds are also described lately⁴⁷. However, until our current work, there has been no rigorous study demonstrating the direct contribution of supersulphides to host defense for anti-influenza virus and anti-SARS-CoV-2 effects and in particular their airway protective effects in vitro and in vivo. This finding may have important implications to better understand how co-infection of influenza virus

and SARS-CoV-2 exacerbated the viral pathogenesis and thereby worsened the clinical outcome or in-hospital mortality, which has been reported recently^{48,49}.”

In addition, let us emphasize that we found the beneficial effect of supersulfides for both influenza and COVID-19 models. This finding may have important implications to better understand how co-infection of influenza virus and SARS-CoV-2 exacerbated the viral pathogenesis and thereby worsened the clinical outcome or in-hospital mortality, which has been reported recently (Ref. 48, 49). It may also suggest the future perspectives that warrant the further application of supersulfides as a potential protective and even therapeutic endeavor for the control of SARS-CoV-2 and other related infectious diseases.

Comment:

C.5: Lines 77 and 78. Nomenclature is confusing. If glutathione persulfide represents GSSH, glutathionetrisulfide should be GSSSH but not GSSSG, which may be called ‘oxidized glutathione trisulfide’. GSSG is usually called oxidized (form of) glutathione. Line 219 also the same. ‘...oxidized persulphide, i.e.GSSSG’.

Response:

R.5: As this reviewer pointed out, the wording of GSSSG might be confusing, so we described it as suggested. Meanwhile, as Dr. Alton Meister wrote GSSG as glutathione disulfide (Meister A, Ann. Rev. Biochem. 52: 711–760), and similar to DAT that this reviewer commented above, it would be still acceptable to call it as “glutathione trisulfide” and therefore leave its term in a way parallel to the newly added word “oxidized glutathione trisulfide”.

Comment:

C.6: Lines 79-83. The description is not correct. H₂S S-sulfurates S-nitrosylated- and S-sulfenylated cysteine residues (Cell Metab 30:1152-1170.e13, 2019; Nat Chem Biol 11, 457-464, 2015).

Response:

R.6: This has been corrected as suggested (new P. 4, L. 81–89). Meanwhile, we still wanted to emphasize that our present study did show clear evidence of strong antiviral activities for supersulfides like GSSSG and inorganic polysulfides (e.g., Na₂S₂, Na₂S₃, Na₂S₄), but not for the H₂S donor (Na₂S).

Comment:

C.7: In experiments in lung premature aging, they used siRNA for CARS2 with which both CARS2 and CPERS activities are suppressed. Cells only suppressed CPERS activity should be used.

Response:

R.7: We thank the reviewer for the suggestion. However, because it is quite difficult to introduce the mutation that selectively suppresses the CPERS activity of CARS2 into human fetal lung fibroblasts, we did not generate the cells that selectively lack CPERS activity. We performed in vivo experiments using CARS2-het-KO mice and showed that supplementation with GSSSG inhibited premature aging of the lungs, which indicates that the deficiency of CPERS activity, rather than cysteinyl-tRNA synthetase activity, is responsible for premature aging of the lungs in CARS2 deficient mice. There is the other major reason why we had to use the CARS2-het-KO mice, as

mentioned above. The magnitude of reduced supersulfides is smaller in AINK mice (be it noted that only heterozygotes are available) than in CARS2-het-KO mice, because only one of the two catalytic sites for persulfide biosynthesis is eliminated in AINK mice, whereas CARS2-het-KO mice are deficient in both sites, thereby persulfide production should be much more attenuated in the latter than in the former. It is reasonable that the phenotypic changes dependent on supersulfides produced by CARS2 are less evident in AINK mice than in CARS2-het-KO mice. Therefore, we still need to employ CARS2-het-KO mice, instead of AINK mice, together with the pharmacological intervention using persulfide donors such as GSSSG, for example (see response R1 above and a quick comment added on P. 8, L. 204–209).

Comment:

C.8: Cytoprotective effect of H₂S, H₂Sn and sulfite have been reported, and these previous studies are to be discussed (FASEB J 18, 1165-1167, 2004; FEBS let 587, 3548-3555, 2013; Br J Pharm 176, 571-582, 2019).

Response:

R.8: Although Na₂S₂₋₄ have antiviral effect as shown in the new Fig. 1e, 2e,f, 4d,g,h; Supplementary Fig. 3b, 4f, 9–13, H₂S and sulfite did not do so. Nevertheless, we put some comments on it in the new text (P. 5, L. 127–131; P. 6, L. 150–P. 7, L. 181; P. 8, L. 220–P. 10, 276).

Comment:

C.9: The description in line 315 is not correct. Several enzymes are involved in the metabolism of H₂S, polysulfides, thiosulfate, sulfite up to sulfate, including sulfur quinone oxidoreductase, sulfur dioxygenase, and rhodanese.

Response:

R.9: We did not claimed that “other enzymes” are not involved in the sulfide metabolisms, but simply that CARS/CPERS is one of the major persulfide synthases, which is supported by this study and a separate work just published (Ref. 53; Please see the R2 above). However, following this comment, we have modified the text by adding the description related to other suggested enzymes (new P. 16, L. 487–P. 17, 492).

Comment:

C.10: Line 337. The interaction of H₂S and NO produces nitrosopersulfides including H₂Sn, HSNO, HNO, and HSSNO (Biochem Biophys Res Commun 343, 303-310, 2006; Nat Commun 5: 4381, 2014; Proc Natl Acad Sci USA 112, E4651-E4660, 2015; Sci Rep 7: 45995, 2017). These previous studies are to be discussed.

Response:

R.10: We thank the reviewer for this keen insight on the chemistry of NO and sulfide, which is now briefly mentioned in the new text (P. 16, L. 457–459).

Comment:

C. 11: Lines 351-352. H₂O₂ S-sulphenylates cysteine residues which are S-sulfurated by H₂S (Cell Metab30:1152-1170.e13, 2019).

Response:

R.11: This is a really keen insight, which we should have mentioned and is now incorporated in the

new text (Supplementary Fig. 1 legend, L. 27–28).

Comment:

C.12: Lines 352-353. Although reaction between H₂S and H₂O₂ is poor, S-sulfenylated protein reacts two orders of magnitude faster with H₂S than with glutathione (J. Biol. Chem 290, 26866–26880, 2015).

Response:

R.12: We thank the reviewer for this keen insight into the reaction of persulfidated protein thiols with and sulfide, of which idea is now briefly incorporated in the new text (P. 4, L. 82–83; Supplementary Fig. 1 legend, L. 28–30).

Reviewer #2 (Remarks to the Author)

In this study, the authors aimed to demonstrate the airway protective role of supersulphides and their therapeutic potential. The topic is interesting, and the authors have generated extensive data to support their conclusions. However, I am not convinced about the main conclusion (i.e. the airway protective role of supersulphides and their therapeutic potential) based on the data provided. Particularly, an extremely high concentration of supersulphide (1 mM GSSSG or 300 μ M GSSG, GSSSG, ...) was used in their experiments. This concentration (1 mM or 300 μ M) is physiologically irrelevant, as the detected concentrations of supersulphides are only in nM or pmol/mg. Supersulphides are very reactive compounds that could non-specifically react with many other proteins and, thus, non-specifically and irreversibly inhibit numerous proteins in the body. The Inhibitory activity data shown in Fig. 2a and b could be due to the non-specific reactivity with many proteins. Supersulphides at such a high concentration would make a lot of unexpected damages in the body due to their non-specific reactivities. In addition, the observed differences in the supersulphide levels between healthy subjects and COVID patients or between WT and *Cars2*^{+/-} mice do not necessarily support the protective effects of supersulphides. Would a supersulphide at the observed concentrations (in nM or pmol/mg) be protective against any virus?

Response:

We thank the reviewer for this important comment. The work for this revision was time-consuming, indeed, because it took time to carry out many in vivo experiments mostly by producing animal models of for the SARS-CoV-2 infections and human clinical investigation. However, we believe that our responses and revision are very successful made below, which are supported by several new data obtained from thorough additional experiments and appropriate modifications of the text accordingly.

The concentration of all supersulfide metabolites that we can assess and quantify is estimated to be 50-100 μ M or so in various cell lines used in our studies. It was highly plausible that these endogenous supersulfides could have anti-SARS-CoV-2 activity in cells during the viral infection. Indeed, the SARS-CoV-2 replication was significantly higher, when endogenous supersulfide formation was attenuated by knocking down (KD) the *CARS2*/*CPERS* expression in the VeroE6/TMPRESS2 cells in culture. On the contrary, the increased viral yield observed in the *CARS2*/*CPERS* KD cells was reversed by the exogenously added persulfide donor GSSSG (Fig. 2a-b; Supplementary Fig. 4f). When we carefully examined the profile of supersulfide metabolites with or without *CARS2*/*CPERS* KD using our sulfur metabolome, the amount of supersulfides was found to be decreased by approximately 30 μ M, by which the SARS-CoV-2 production, as assessed by the plaque reduction assay, was remarkably elevated by almost 3-folds. This indicates that endogenous and basal formation of supersulfides should contribute significantly to the innate or naturally occurring antiviral effect of the cells. More importantly, such an attenuated antiviral effect was significantly restored by addition of the same range of GSSSG concentrations, supporting the above notion that even baseline supersulfides have physiologically relevant defense-oriented consequences against SARS-CoV-2 replication. It is therefore conceivable that tens to hundred of micromolar concentrations are likely required to achieve apparent antiviral effects for GSSSG administered exogenously to the SARS-CoV-2 infected cells, because it would become evident only when the doses are above normal or baseline supersulfide concentrations. In other words, the application of sub-millimolar GSSSG proposed here is considered to be physiologically and thus pharmacologically intervened in cells and in vivo. In fact, we have not observed any cytotoxic effects with as much as 1 mM GSSSG for various cultured cells employed so far irrespective of viral infections.

These data and corresponding explanation are now included in the revision: Fig. 2a-b;

Supplementary Fig. 4f; related statements in the text, P. 6. L. 150–P. 7, L. 181.

As for the therapeutic advantage of supersulfides, let us emphasize that we found the beneficial effect of supersulfides for both influenza and COVID-19 models. This finding may have important implications to better understand how co-infection of influenza virus and SARS-CoV-2 exacerbated the viral pathogenesis and thereby worsened the clinical outcome or in-hospital mortality, which has been reported recently (new text, P. 16, L. 465-471; Ref. 48, 49). It may also suggest the future perspectives that warrant the further application of supersulfides as a potential protective and even therapeutic endeavor for the control of SARS-CoV-2 and other related infectious diseases.

Reviewer #3 (Remarks to the Author)

General comment:

Sano et al. report that the innate defense functions of supersulphides that protect the lung and airways against viral infections, COPD, pulmonary fibrosis, and ageing. This is interesting paper. They showed the beneficial roles of supersulphides in various diseases; however, they did not address sufficiently the precise mechanisms of these beneficial effects in these diseases.

Response: We thank the reviewer for providing the critical comments for our study, which are all very helpful to improve our work, especially in terms of mechanistic insight into the beneficial effect of supersulfide in various lung disease pathogenesis, including viral infections, COPD, IPF, and ageing.

The work for this revision was time-consuming, indeed, because it took time to carry out many in vivo experiments mostly by producing animal models of for the SARS-CoV-2 infections and human clinical investigation. However, we believe that our responses and revision are very successful made below, which are supported by several new data obtained from thorough additional experiments and appropriate modifications of our work.

We have addressed the reviewer's concerns in point-by-point responses, with each comment and response numbered sequentially as C and R, respectively.

Major comments:

Comment:

C.1: Anti-influenza defense: Authors showed a highly beneficial and protective effect of supersulphides in the influenza model. However, the mechanisms of suppressive effect of supersulfides on the growth of influenza virus in vivo in the lung (Fig. 1d) should be addressed.

Response:

R.1: As pointed out by this reviewer, while supersulfides, either endogenously produced or pharmacologically applied, showed potent beneficial effects for the influenza models we employed herein, we do not have a clear view on the precise or specific mechanism of anti-influenza effect, e.g., whether supersulfides cause a direct or indirect antiviral action, in mice in vivo. Nevertheless, a possible direct antiviral mechanism might be a structural alteration of hemagglutinin (HA), a major envelop protein that is most responsible for viral attachment and integration into host cells via its membrane fusion activity, which could be induced by supersulfides that may directly affect several disulfide bridges of HA polypeptides. The same antiviral effect on the S protein of SARS-CoV-2 has been recently reported and is indeed suggested by our current study, which was performed shown during the revision of the paper (new Fig. 1e, 2e,f, 4; Supplementary Fig. 3b, 4f, 9–13, and related informations, P. 5, L. 127–131; P. 6, L. 150–P. 7, L. 181; P. 8, L. 220–P. 10, 276). H₂S and sulfite did not do so. Nevertheless, we put some comments on it in the new text (P. 16, L. 460–471).

Although a precise antiviral mechanism of supersulfides against influenza remains to be fully elucidated, here we found the beneficial effect of supersulfides for both influenza and COVID-19 models (see below). This finding may have important implications to better understand how co-infection of influenza virus and SARS-CoV-2 exacerbated the viral pathogenesis and thereby worsened the clinical outcome or in-hospital mortality, which has been reported recently (Bai et al., Cell Res 2021). It may also suggest the future perspectives that warrant the further application of supersulfides as a potential protective and even therapeutic endeavor for the control of SARS-CoV-2 and other related infectious diseases.

Comment:

C.2: Anti-COVID activities of supersulphides: Authors should demonstrate the protective effect of supersulphides against COVID-19 in vivo using mice.

Response:

R.2: Following the suggestion by this reviewer, we performed SARS-CoV-2 infection in CARS2 mutant mice, i.e., AINK mutant mice, crossed with ACE2 transgenic mice. We developed these ACE2 and AINK double-mutant (ACE2-Tg::*Cars2*^{AINK/+}) mice to produce SARS-CoV-2 infection in mice, because the wild-type mice that do not possess ACE2 originally had to be genetically engineered to express ACE2 so that they can be fully susceptible to SARS-CoV-2 in the airway (new Fig. 3a-d; Supplementary Fig. 8a, and related text, P. 7, L. 197–P. 8, L. 209).

The results obtained showed that the lethal effect of intranasal infection with SARS-CoV-2 was significantly higher in ACE2-AINK-mutant mice than in the ACE2-transgenic mice. The enhanced pathogenesis was found to be caused by the accelerated viral growth in the lungs of ACE2-AINK-mutant mice compared with that in ACE2-transgenic mice. These data indicate that persulfides or related supersulfides generated by CARS2/CPERS may contribute to the anti-SARS-CoV-2 host defense in vivo. Furthermore, the persulfide production of ACE2-AINK mutants was decreased compared with that of the wild-type mice during SARS-CoV-2 infection (new Supplementary Fig. 8a; related statements in the text, P. 7, L. 197–P. 8, L. 209). Notably, another in vivo COVID-19 model in hamsters confirmed the antiviral effect of GSSSG (new Fig. 3e, g; Supplementary Fig. 8b and related text, P. 8, L. 210–218).

Comment:

C.3: COPD: In figure 3, authors used elastase-induced emphysema model. However, this mouse model is not appropriate for COPD model. The mouse model by CSE inhalation for at least 3 months is beneficial.

Response:

R.3: Although the elastase-induced emphysema model is not a perfect mimic of actual COPD patients, it is widely used as a model of emphysematous changes, an important pathological change in the COPD pathogenesis. Our institution did not have the equipment to perform the cigarette smoke inhalation model and was unable to do so, but instead we conducted experiments in a model in which emphysema is induced by intranasal administration of cigarette smoke extract, as shown in old Figure 5 (new Fig. 8). The results of both models suggest that CARS2 and the supersulfides produced by CARS2 are protective against the pathologies seen in COPD, such as emphysematous changes and pathological aging.

Comment:

C.4: Emphysema in CPER-deficient mice: In Figure 4, authors showed that supersulphides produced by CARS2/CPERS contribute to protection of COPD pathogenesis through the strong anti-inflammatory and antioxidant effects of supersulphides. Although authors showed various biomarkers for antioxidative stress, they should also show the results of measurement for oxidative stress.

Response:

R.4: Based on the data from experiments using CARS2-hetero KO mice, the levels of oxidative stress in elastase-treated AINK mice were supposed to be higher compared to those of WT mice during the acute inflammatory phase. However, the levels of 8OHdG in lung tissues were low and

were not significantly different between the two groups on day 21 of this experiment (data not shown). On the contrary, we would like this reviewer to realize that the magnitude of reduced supersulfides is smaller in AINK mice (be it noted that only heterozygotes are available) than in CARS2-het-KO mice. This is because only one of the two catalytic sites for persulfide biosynthesis is eliminated in AINK mice, whereas CARS2-het-KO mice are deficient in both sites, thereby persulfide production should be much more attenuated in the latter than in the former. It is, therefore, reasonable to assume that the phenotypic changes dependent on supersulfides produced by CARS2 are less evident in AINK mice than in CARS2-het-KO mice.

Comment:

C.5: Supersulphides in lung premature aging: In Figure 5, authors described that supersulphides produced by CARS2 limit the progression of cellular senescence and prevent lung ageing by virtue of their antioxidant function and contribution to mitochondrial energy metabolism. However, they only showed that aged mice are more susceptible to CSE instillation. They should show direct evidences that these protective effects were due to antioxidative function and due to contribution to mitochondrial energy metabolism in aged mice.

Response:

R.5: We would like to explain that panels a to e in Figure 5 (new Fig. 8) show the results of the emphysema model induced by intranasal administration of cigarette smoke extract (CSExt), and panels f to i in Figure 5 (new Fig. 8) show the results of the analysis of 88-week-old mice raised in a normal environment. In the emphysema model, intranasal administration of CSE induced emphysematous changes and pathological aging, which were exacerbated in CARS2^{+/-} mice. Observations in aged mice reared in a normal environment also show that the expression of senescence markers is enhanced in CARS2^{+/-}, and emphysematous changes are also exacerbated. These results indicate that CSE-induced pathological aging as well as the changes observed in aged mice are enhanced in CARS2^{+/-} mice, suggesting that CARS2 and supersulfides produced by CARS2 may have anti-aging properties, but indirectly supported by our recent studies showing that sulfur repletion is effectively taking place in the mammalian mitochondria (Ref. 26, 37, 38). As the reviewer pointed out, the in vivo results in Figure 5 (new Fig. 8) do not directly show that antioxidant function and mitochondrial energy metabolism are impaired in CARS2^{+/-} mice. We have revised the text as pointed out by the reviewer (new P. 14, L. 395–396; P. 17, L. 492–494 of the revised text).

Reviewer #4 (Remarks to the Author)

General comment:

In this manuscript, the authors extend their earlier findings—that endogenous supersulphide production is reduced in COPD and related chronic inflammatory airway diseases—and test the hypothesis that supersulphides protect against negative outcomes of diverse inflammatory airway diseases, including that caused by influenza virus, SARS-CoV-2 (the causative agent of COVID-19), chronic obstructive pulmonary disease (COPD), idiopathic pulmonary fibrosis (IPF), and ageing. By using a newly developed knockin mouse model with a mutation in a cysteine persulphide synthase enzyme (*Cars2*AINK/+), they demonstrated that *Cars2* is the major contributor to supersulphide biosynthesis in the lung. In addition, they used *Cars2*AINK/+ mice, a previously developed knockout mouse model (*Cars2*-/+), and/or treatment with exogenous supersulphide (GSSSG), to show that supersulphides protect against lethal disease outcomes in a mouse model of severe influenza virus disease, impair SARS-CoV-2 protease activity in a cell-free in vitro system, and protect against negative outcomes caused by chronic lung diseases, such as COPD and IPF. They also developed an approach to measure supersulphide metabolites in the breath of COVID-19 patients (“breath-omics”), and generated some data suggesting that supersulphides may be highly expressed and/or heavily metabolized in humans with COVID-19.

Overall, their approach is comprehensive, they developed a new reagent that may be useful to the research community (*Cars2*AINK/+ mice), they developed a new method to measure metabolites in exhaled breath that might be useful as a diagnostic tool, and they identified a potential avenue for development of a broad-spectrum therapeutic to treat diverse inflammatory lung diseases. The concept of a single type of treatment to mitigate risks associated with acute or chronic lung inflammation of diverse origins is alluring, and the potential long-term benefits to human health could be immense (e.g., by reducing chronic disease burden in ageing populations; or by reducing morbidity and mortality in epidemics or pandemics caused by certain emerging respiratory pathogens). In this regard, the data presented in this manuscript have the potential for a broad impact in biomedical research and translational diagnostic or therapeutic development. On the other hand, the manuscript has some important limitations that need to be considered carefully:

Response:

We would like to thank this reviewer for providing valuable and insightful comments. We tried our best to address the reviewer’s concerns by following point-by-point responses, with each comment and response numbered sequentially as C and R, respectively.

The work for this revision was time-consuming, indeed, because it took time to carry out many in vivo experiments mostly by producing animal models of for the SARS-CoV-2 infections and human clinical investigation. However, we believe that our responses and revision are very successful made below, which are supported by several new data obtained from thorough additional experiments and appropriate modifications of this work.

Comment:

C.i: Influenza mouse experiments are solid and generally convincing, with *Cars2*-/+ mice exhibiting an increase in disease severity and GSSSG-treated mice exhibiting a decrease in disease severity after influenza virus infection (note: higher viral genomic RNA and inflammatory cytokine levels in the lung are associated with more severe disease). In all experiments, infections were performed with a single dosage of a common, mouse-adapted laboratory strain of influenza (PR8). Influenza mouse experiments could be improved by using a clinically relevant influenza virus strain (e.g., the 2009 pandemic H1N1 virus causes severe disease in mice without adaptation), determining the lethal dose 50 (LD50) in *Cars2*-/+ mice or GSSSG-treated mice relative to appropriate controls,

and using virus titration as are adout of virus growth. Also, it would be interesting to assess whether GSSSG treatment can reduce influenza growth in cell culture, to confirm the proposed antiviral effect.

Response:

R.i: We are grateful for this suggestion to extend our work on this murine influenza pathogenesis involving innate host defense mediated by supersulfides. It is certainly important to see the beneficial effect of supersulfides on the influenza infection caused by the 2009 pandemic strain (A/H1N1pdm). However, because of biosafety regulations and limited facilities at our university, we can't handle many human pathogenic viruses at the same time and place for basic laboratory environments, and besides, we eagerly wanted to focus more on the COVID-19 rather than influenza here in our work and its revision. Therefore, our additional studies rigorously clarified the protective and host defence antiviral functions of supersulfides against SARS-CoV-2 infections in vivo in mice and hamsters as well (see new Fig. 3, Supplementary Fig. 7, 8; and related description, P. 7, L. 183–P. 8, L. 218).

Comment:

C.ii: Currently, SARS-CoV-2 experiments are restricted to examining the effects of GSSSG on viral protease activity (in vitro, cell-free assay) and the production of viral genomic RNA (in cell culture) as aproxy for virus growth. In vitro data strongly suggest a role for GSSSG in the inhibition of viral protease activity (although this remains to be confirmed in cell-based or animal models); however, the effects of GSSSG on viral genomic RNA production in cells are modest, suggesting potentially limited antiviralactivity. The SARS-CoV-2 cell culture experiment could be improved by sampling GSSSG-treated cultures in a growth curve (rather than a single time point) and by using virus titration as a readout ofvirus growth. In addition, given the authors' emphasis of supersulphides as a potential treatment forCOVID-19, it is essential to show the effect of supersulphides on COVID-19 severity in an animal model (e.g., GSSSG treatment in hamsters infected with SARS-CoV-2).

Response:

R.ii: The antiviral activities of supersulfides generated either endogenously or exogenously are further confirmed by our present additional studies as follows.

We performed the plague reduction assay, by which we can see how exactly the viral replication or production is affected by any tested compounds throughout the course of infection, and we found that endogenous supersulfides can exhibit anti-SARS-CoV-2 activity in the cells during the viral infection. The SARS-CoV-2 replication was significantly higher, when endogenous supersulfides formation was attenuated by knocking down (KD) the CARS2/CPERS expression in the VeroE6/TMPRESS2 cells in culture. The increased viral yield observed in the CARS2/CPERS KD cells was reversed by GSSSG added exogenously (Fig. 2, Supplementary Fig. 4; related statements in the text, P. 6. L. 150–P. 7, L. 181). These cell culture studies for viral replication revealed that supersulfides cause physiologically relevant anti-SARS-CoV-2 effects. The specific antiviral functions of supersulfides were further confirmed by the fact that there was no cytotoxic effect with as much as 1 mM GSSSG for various cultured cells employed irrespective of viral infections.

Following the suggestion from this reviewer, we carried out SARS-CoV-2 infection in two animal models in vivo. Regarding the mouse COVID-19 model, we first developed ACE2-transgenic and AINK double-mutant (ACE2-Tg::*Cars2*^{AINK/+}) mice to effectively produce the SARS-CoV-2 infection in mice, because the wild-type mice that do not possess ACE2 originally needed to be genetically engineered to express ACE2 so that they can be fully susceptible for SARS-CoV-2 in the airway and in vivo. Lethality induced by the intranasal infection of SARS-CoV-2 was significantly higher

in ACE2-AINK-mutant mice that in the ACE2-transgenic mice. The enhanced pathogenesis observed with AINK mutation introduced is likely to be caused by the accelerated viral growth in the lungs of ACE2-AINK-mutant mice compared with that in ACE2-transgenic mice. It is therefore concluded that supersulfides generated by CARS2/CPERS can contribute to the anti-SARS-CoV-2 host defense in vivo. This interpretation is supported by the following new findings as well as those originally shown. The persulfide production of ACE2-AINK mutant mice was decreased compared with that of the wild-type mice during SARS-CoV-2 infection. Moreover, as this reviewer requested above, we also examined the effect of GSSSG pharmacologically administered intraperitoneally (i.p.) on the experimental viral pneumonia in hamsters, which is induced by the intratracheal inoculation of SARS-CoV-2. GSSSG treatment clearly resulted in the beneficial consequences, including reduced viral propagation and attenuated pneumonia pathology as manifested by body weight loss and pulmonary consolidation.

Taken together, it is consistently concluded that CARS2-derived supersulfides have potent anti-SARS-CoV-2 activity and thus mediate the innate host defense against COVID-19. These results are now presented in the revised manuscript: new Fig. 3; Supplementary Fig. 7, 8; and related description, P. 7, L. 183–P. 8, L. 218.

Comment:

C.iii: The authors report that supersulphide metabolites might be useful as biomarkers of COVID-19 (or other lung disease) severity. However, this conclusion cannot be made given the available information and analyses performed: specifically, the number of patients analyzed is low (12 per group, healthy or infected), with only a single patient designated as having exacerbated disease; no information about sampling time points or disease severity upon sampling are reported; it appears that all COVID-19 patients, regardless of disease stage or symptom severity were analyzed as a single group; and statistical procedures designed to evaluate potential biomarkers were not employed. These limitations substantially weaken the impact of the clinical data. To improve this analysis, the authors need to recruit more participants, stratify participants by disease severity/outcome, and apply a more rigorous statistical analysis designed specifically for biomarker identification.

Response:

R.iii: During this revision, we tried our best to recruit some additional COVID-19 patients so that we could conduct stratified sampling for rigorous statistical analysis to determine the significance of supersulfides as biomarkers as detected by breath omics. Unfortunately, however, it was quite difficult for us to collect breath aerosol (exhaled air) via a mouthpiece method from severely or critically ill COVID-19 patients, who were often unable to achieve a stable and spontaneous breathing without the assistance of mechanical ventilation due to viral pneumonia associated with acute respiratory failure. This disease state made it difficult to collect sufficient volumes of EBC (exhaled breath condensates) for the sulfur metabolome that we pursued in this study. Therefore, almost all patients who provided their EBC were of moderate severity, without fatal lung damage or dysfunction, but with mild pneumonia and other disorders or complications, if any.

Nevertheless, more than 20 subjects collected herein have somehow distinct backgrounds or underlying diseases, for example, diabetes mellitus, otherwise different ill conditions with or without hospitalization requirement during infection and thereby breath collection and appropriate therapeutics are necessary upon admission or various laboratory tests for clinical diagnoses. We compared the abundance of various sulfur metabolites and supersulfides among the SARS-CoV-2 infected subjects stratified according to the severity of symptoms, underlying diseases, need for hospitalization, length of hospital stay, and so on. While each EBC value of supersulfides is significantly higher in the COVID-19 individuals than in healthy controls, however, there is no

statistical difference in the striated comparison, as long as we examined the patients with severe illness or long-term hospitalization.

These findings are described in the revised text (P. 10, L. 289–P. 11, L. 300) and shown in the new Fig. 5a and Supplementary Fig. 14.

More importantly, because we encountered some limitations in this clinical study for biomarker exploration, we sought to develop an animal model for the breath sulfur analysis so that we can get some evidence on the potential application of supersulfide for biomarker of COVID-19. Specifically, we constructed a special instrument, as shown in new Fig. 5b-f and Supplementary 14b-f, by which we can collect appreciable amount of EBC from hamsters with or without SARS-CoV-2 infections. The hamster breath analysis thus indicated that various but almost the same sulfur metabolites as detected in human EBC are remarkably elevated in the course of SARS-CoV-2 lung infections, which correlated well with the time profile of body weight loss and extension of pulmonary consolidation, as shown in the new Fig. 5. Also, it should be noted that the profile of the sulfur metabolites detected in the hamster EBC was almost identical to that of human EBC from the COVID-19 patients. It is thus interpreted that supersulfides serve as potential biomarkers for the COVID-19; although they may not be suitable for estimating disease severity or predicting prognosis.

This notion is now included in the revised manuscript: P. 10, L. 278–P. 11, 312.

Comment:

C.iv: The writing style is imprecise and at times difficult to follow. In particular, the flow of the results section does not seem logical and the figure legends are frequently inadequate to describe the data shown in the figures. Significant revisions are needed to improve organization and flow, clarity of experimental descriptions and the associated results, and the appropriate inclusion/exclusion of inferences that can and cannot be made from the available data.

Response:

R.iv: We are grateful for this reviewer's very careful editorial check of the contextual flow of our statement and the narrative of the text. We have tried our best to improve it, and hopefully the reviewer was able to read it much better with this new manuscript than before.

Specific Comments:

Comment:

C.1. Line 1: Suggest changing the title; specifically,

a. "Supersulphide airway protection" should be "Supersulphide-mediated airway protection" or something similar.

b. It would be better to not use a colon.

c. "Viral and inflammatory lung disease" is not a good summary description of the disease models examined in this study. Both influenza disease and COVID-19 are driven by inflammation, like disease in the COPD/IPF/ageing models. The authors need to come up with another phrase that better encompasses the non-infectious disease models (e.g., chronic lung diseases).

Response:

R.1: We modified the title of the paper as suggested, which read now as "Supersulfide-mediated airway protection in viral and chronic lung diseases".

Comment:

C.2.: The Abstract needs substantial revision to increase precision/accuracy and to better reflect the findings of the study. Specifically,

C.2-a. Line 58 (abstract): "...viral airway infections, i.e. influenza and COVID-19..."; it would be more accurate to say "i.e. influenza and SARS-CoV-2," since COVID-19 is not a virus, but rather, the disease caused by SARS-CoV-2.

Response:

R.2-a: We are somehow afraid to say that the word "influenza" does not mean "influenza virus", but is the name of the viral disease "influenza", the same is true for COVID-19 vs. SARS-CoV-2.

Comment:

C.2-b. Lines 60-61 (abstract): "This strong anti-SARS-CoV-2 defense was evident with the supersulphidedonor glutathione trisulphide (GSSSG)." The data presented for SARS-CoV-2 include an in vitro assay to measure the effect of GSSSG on viral protease activity and the effect of GSSSG on production of genomic RNA in cell culture. It is not clear whether the inhibitory effect of GSSSG on viral protease activity, observed in vitro, is similar in the context of viral infection. Moreover, in infected cells, GSSSG treatment caused a modest reduction in viral genomic RNA levels (~80%) at the highest GSSSG concentration tested, and the effect on infectious virus output was not tested. Put simply, the SARS-CoV-2 data is not the strongest data in the manuscript and framing it this way in the abstract imparts undue importance.

Response:

R.2-b: To obtain more robust evidence for anti-SARS-CoV-2 function of supersulfides, we carried out SARS-CoV-2 infection in CARS2 mutant mice, i.e., AINK mutant mice, crossed with ACE2 transgenic mice. We developed these ACE2 and AINK double-mutant (ACE2-Tg::*Cars2*^{AINK/+}) mice to produce SARS-CoV-2 infection in mice, because wild-type mice do not express ACE2, so that they can be fully susceptible to the SARS-CoV-2 infection in vivo. The results showed accelerated viral growth in the lungs of ACE2-AINK-mutant mice after SARS-CoV-2 infection, as compared with that in ACE2-transgenic mice (Fig. 3; Supplementary Fig. 7, 8; and related description, P. 7, L. 183–P. 8, L. 209). This indicates that persulfides or related supersulfides generated by CARS2/CPERS contribute to the anti-SARS-CoV-2 host defense in vivo. The current interpretation is further supported by the new findings as well as the data originally shown as follows.

The effect of supersulfides on the yield of infectious virus was examined rigorously by the plaque reduction assay, which we designed here to most accurately assess endogenous and exogenous supersulfides such as GSSSG by measuring the number and size of the plaques, which are indicative of virus production as affected by supersulfides. Similar to the in vivo experiment, our plaque reduction assay for SARS-CoV-2 replication showed significantly higher viral replication in the cultured VeroE6/TMPRESS2, when endogenous supersulfide formation was attenuated by knocking down the CARS2/CPERS expression, which was reversed by exogenously added persulfide donor GSSSG (Fig. 2, Supplementary Fig. 4; related statements in the text, P. 6. L. 150–L. 181). Notably, the COVID-19 hamster model unequivocally demonstrated the antiviral effect of GSSSG (Fig. 3e,g and Supplementary Fig. 8b; related text, P. 8, L. 210–218). While the suppression of the viral replication is likely due to inhibition of the viral thiol proteases by supersulfides, as shown in the original version of this manuscript, another possible mechanism has been newly proposed in the current revised work (Fig. 4g,h; Supplementary Fig. 10,13; related description, P. 8, L. 221–L. 228) and also recently reported (Ref. 31). That is, it may be mediated via direct impairment of the viral infectivity caused by alteration of a viral structural protein such as S protein, likely induced by supersulfide-mediated cleavage of disulfide bridges.

Therefore, we have fully revised the description pointed out by the reviewer to be more precise and persuasive, and even solid in terms of scientific merit and integrity achieved in this revision.

In the meantime, please note that we had to shorten the Abstract by about 50 words to be no more than 150 words. Therefore, the statements that we modified following your comment mentioned above are moved to a bit later section.

Comment:

C.2-c. Lines 61-62 (abstract): “We observed a similar beneficial effect of supersulphides in vivo in influenza pneumonia in mice...” The SARS-CoV-2 and influenza experiments were completely different, so it is incorrect to say that the effect of supersulphides between systems was similar.

Response:

R.2-c: As stated in the responses above, we now realize that there may be a common denominator in the viral infectivity between SARS-CoV-2 and influenza virus with respect to susceptibility of these viruses to supersulfides, especially inorganic polysulfides. Therefore, we have included some arguments related to this issue in the nw, for example, on P. 8, L. 220 – P. 9, 238; Supplementary Fig. 13 and its legend.

Comment:

C.2-d. Lines 64-65 (abstract): “Enhanced endogenous supersulphide production also occurred in human airways in COVID-19, as identified by breath sulphur-omics we developed herein.” The reported experiment identified increased levels of sulphur metabolites in the breath of COVID-19 patients. Higher levels of sulphur metabolites is not direct evidence of increased supersulphide production, and could have other explanations (e.g., increased metabolism of existing supersulphides).

Response:

R.2-d: This comment is absolutely correct. We thank the reviewer for pointing it out. We have added appropriate explanation (P. 11, L. 301–312) and also quickly mentioned it in the legend (L. 241-244) for Supplementary Fig. 14, accordingly.

Comment:

C.2-e. Lines 67-68 (abstract): “...were greatly mitigated by supersulphides, either endogenous (CARS2/CPERS) or exogenous (GSSSG).” CARS2/CPERS is not an endogenous supersulphide, as implied by the text, but rather the enzyme that produces supersulphides.

Response:

R.2-e: These points have been more accurately written following this comment (L. 68–69).

Comment:

C.2-f. Lines 70-71 (abstract): “...which seemingly affect COVID-19 prognosis and even ageing.” The placement of this phrase is odd and may warp the intended meaning of the sentence.

Response:

R.2-f: We have deleted this phrase.

3. Introduction:

Comment:

3.a-i. Lines 85-89 (introduction): “More important, oxidative stress associated with airway inflammation has been implicated in the pathogenesis of influenza, COVID-19, and chronic lung disorders including [COPD], emphysema, [IPF] and even lung ageing 14-18.”

i. It is not clear why this sentence is more important than the preceding information.

Response:

R.3.i. We have deleted “More important” here.

Comment:

3.a-ii: The references at the end of this sentence do not include any papers that discuss oxidative stress in influenza or SARS-CoV-2 infections.

Response:

R.3.ii: We apologize for this incomplete citation, which is now corrected (see new Ref. 21-23) by citing the old Ref. 24-26 and 45, which were previously included in the list of references, but were not located right here.

Comment:

3b: Line 89 (introduction): “Our preliminary studies” should be “Our previous studies.”

Response:

R.3b: We have corrected as suggested.

Comment:

3c: Line 94 (introduction): “(CARs)” should be “(CAR)”

Response:

R.3c: We have corrected it as suggested, and CPERSs (to CPERS) too, throughout the text.

Comment:

C.3d: Line 100 (introduction): “We report here the innate but highly evolved defense functions of supersulphides...” Highly evolved relative to what?

Response:

R.3d: The statement has been modified on L. 102–104 as follows: “We report here that the innate defense functions of supersulphides, which are highly conserved among organisms, efficiently protect”

Comment:

3e. Lines 107-110 (introduction): Cars2-/+ mice need a reference.

Response:

R.3e: Because we had to shorten the statement in this section due to a strict word limitation for the main text, this corresponding word is now deleted in the new text, instead, the appropriate

information for *Cars2*^{+/+} mice is included in the new Methods section.

Comment:

C4. Lines 120-121 (results): "...one novel aspect of supersulphides that we identified is their antiviraleffects on influenza and COVID-19 viruses (Figs. 1 and 2)."

C4a: Technically, no antiviral effect has been established in this work. In Fig. 1, the authors show that viral genomic RNA levels are increased (2-3-fold) in *Cars2*^{-/+} mice after influenza virus infection relative to WT mice, but they did not report the effect on infectious virus output. In Fig. 2, the authors show that GSSSG treatment caused a modest reduction in viral genomic RNA levels (~80%) at the highest GSSSG concentration tested (in cell culture), but they did not report the effect on infectious virus output.

Response:

R.4a: As we answered the above question in **R.2-b**, the direct antiviral effect of supersulfides has been confirmed mainly for SARS-CoV-2, as our revision shows herein. We might have to conduct much rigorous analysis to clarify how exactly supersulfides affect the disulfide bridges or other residues of S protein and other envelope or structural proteins of SARS-CoV-2. Nevertheless, we have done some rigorous proteome studies for the S protein and spike protein during this revision. For example, our present study shows strong anti-coronavirus activities for supersulphides like GSSSG and inorganic polysulphides (e.g., Na₂S₂, Na₂S₃, Na₂S₄), but not for the H₂S donor (Na₂S). The suppression of the viral replication is most likely due to conformational alterations of the coronavirus structural protein, i.e., spike glycoprotein (S protein), by supersulphide-induced dissociation of their disulphide bridges (Fig. 2e,f; Fig. 4g,h; Supplementary Fig. 10, 13). Intriguingly, a similar antiviral mechanism via the disulphide cleavage at the identical site of S protein was proposed recently (Ref. 31). The same antiviral mechanism via structural alteration of viral proteins may be true for influenza virus HA that is a major envelope protein and is thereby most responsible for viral attachment and integration into host cells via its membrane fusion activity. As described above, influenza virus was indeed susceptible to be inactivated by supersulfides (Fig. 1e), in which several disulfide bridges of viral HA polypeptides are likely to be affected and conformationally distorted by inorganic polysulfides (Na₂S₂₋₄) in the same manner as that of coronavirus.

Although a precise antiviral mechanism of supersulfides against influenza remains to be fully elucidated, here we found the beneficial effect of supersulfides for both influenza and COVID-19 models. This finding may have important implications to better understand how co-infection of influenza virus and SARS-CoV-2 exacerbated the viral pathogenesis and thereby worsened the clinical outcome or in-hospital mortality, which has been reported recently (Ref. 48,49). It may also suggest the future perspectives that warrant the further application of supersulfides as a potential protective and even therapeutic endeavor for the control of SARS-CoV-2 and other related infectious diseases.

Comment:

R.4b. "COVID-19 virus" would be more accurately described as SARS-CoV-2.

Response:

R.4b: We thank the reviewer for letting us know this wrong expression, which has been corrected accordingly.

Comment:

R.4c: The authors should cite Extended Data Fig. 2 with Figs. 1 and 2 at the end of this sentence.

Response:

R.4c: We have cited the figure as suggested.

Comment:

C5. Lines 123-125 (results): “We found that endogenous supersulphide formation in lungs was much lower in Cars2-deficient (Cars2-/-) mice than in WT mice throughout the course of infection (ExtendedData Fig. 3).”

C.5a: This statement is not true for most of the supersulphide metabolome shown in Extended Data Fig. 3.

Response:

R.5a: We have now explained carefully and accurately altered the profile of supersulfide metabolites, as suggested.

Comments:

C.5b: In a few cases (CysSSH, HS-, HSS-), there does appear to be a significant difference in supersulphide levels in WT and Cars2+/- mice at the 0 day time point. However, additional comparisons between WT day 0 and Cars2+/- day 3 or day 7 are not correct—comparisons should be made between WT and Cars2+/- mice at the same time points, and may need to be normalized to the expression level observed at day 0.

Response:

R.5b: We have changed the way of comparison among the different experimental groups, so that each value for the corresponding and related group can be appropriately evaluated.

Comment:

C.5c: Also, it is notable that there are different patterns of expression for different supersulphide metabolites. Does this provide insight into the pathways responsible for supersulphide metabolite production (or use) in the various conditions?

Response:

R.5c: We thank the reviewer for this insightful and thoughtful comment. Although we still do not fully understand it, there should be different modes of metabolic regulation for various sulfide metabolites. For example, in the case of physiological pathway, persulfides will be oxidatively metabolized via SQR and its downstream pathways mediated by ETHE1 and sulfite oxidase, etc. This notion will not be mentioned at this time, however, because it appears out of scope of our present paper, and the related issues will be discussed much profoundly in our future work soon published elsewhere.

Comment:

C.6. Lines 129-131 (results): “Of great importance is that growth of influenza virus in vivo in the lung significantly increased in Cars2+/- mice compared to WT mice.” As already noted, the authors

have not established an antiviral effect, since they did not examine infectious virus output.

Response:

R.6: As we answered earlier, the direct antiviral effect of supersulfides was argued here in our revision. Please see the answers above.

Comment:

C.7. Lines 132-134 (results): “Also, supersulphides produced by Cars2 are likely to abolish not only cytokine storm-like excessive inflammatory responses but also nitrative and oxidative stress (Fig. 1; Extended Data Fig. 2a-e), all of which are critically involved in influenza pathogenesis, as we reported earlier 22-26.” Fig. 1g and 1n suggest that supersulphides can reduce the production of inflammatory cytokines (although inflammation is not abolished). However, there do not appear to be any data to support the idea that supersulphides also mitigate oxidative and nitrosative stress in influenza infections.

Response:

R.7: We thank the reviewer for important comments. We additionally performed immunostaining for 8-hydroxy-2'-deoxyguanosine (8-OHdG) to investigate the status of oxidative stress in the influenza model. We found that influenza infection increased the relative number of 8-OHdG-positive cells in the wild-type lung tissue: the proportion of 8-OHdG-positive cells in the influenza-infected lung was much higher in CARS2^{+/-} mice than in the wild-type mice (Fig. 1i). The increase in the ratio of 8-OHdG-positive cells due to influenza infection observed in CARS2^{+/-} mice looks like somewhat suppressed, although it was not statistically significant, by the treatment with GSSSG (Supplementary Fig. 2j). Another important finding that we obtained herein revealed the exacerbated oxidative stress caused by reduced production of supersulfides, which is evidenced by the increased oxidized GSH (GSSG) ratio vs GSH (GSSG/GSH) in CARS2^{+/-} mouse lungs, as compared with that in wild-type lungs, during influenza infection (Fig. 1j). Also, protein-bound 8-nitrotyrosine formation, indicative of nitration reaction in the tissue, was significantly elevated in CARS2^{+/-} mice after infection (previously presented data; Supplementary Fig. 2e). These results suggest that supersulphides produced by CARS2 may ameliorate the oxidative and nitrative stress induced by influenza infection. As pointed out by the reviewer, the word “abolish” better read “alleviate” or “mitigate”, as you meant. Please see the new statement relevant to the above discussion: P. 5, L. 135–P. 6, L. 138

Comment:

C.8: Lines 136-138 (results): “...influenza virus infection per se reduced endogenous supersulphide production, which may aid viral propagation and inflammatory responses in virus-infected lungs (Extended Data Fig. 3).” As already stated above, this is true only for a subset of the supersulphides presented in Extended Data Fig. 3.

Response:

R.8: We have amended in a way of much appropriate comparison (new Fig. 1j; Supplementary Fig. 3a), so that the time-dependent profile of supersulfides is now more clearly illustrated than before in terms of the time-dependent change during infection of the lung.

Comment:

C.9. Lines 138-141 (results): “In fact, intranasal administration of GSSSG markedly reduced all

pathological and inflammatory consequences, including survival, in influenza-infected WT animals. This result suggests that supersulphides supplied exogenously compensate for impaired antiviral and protective functions of supersulphides that are depleted by oxidative stress and excessive inflammatory reactions..." The effect of GSSSG on production of viral genomic RNA is not shown; thus, there is no measure of antiviral activity shown for GSSSG treatment.

Response:

R.9: As the reviewer pointed out, this experiment only shows that GSSSG attenuates influenza virus-induced lung injury and inflammatory response and improves prognosis, but not its effect on virus production. We have thus deleted the phrase "antiviral and" from the original sentence. Regarding this particular request, also please see the previous response describing the antiviral effect of supersulfides above.

Comment:

C.10. Figure 1 and Figure 1 legend:

- a. This figure could be improved by making the two halves (a-g and h-n) more consistent:
- Show viral genomic RNA levels for both.
 - Show the same immune cell subset numbers for both.
 - Arrange the data in the same order.

Response:

R.10a: We thank the reviewer for the comments. We have modified and rearranged this figure following the reviewers' comments.

- As answered in R9, we did not measure viral genomic RNA in Fig1n-h (the influenza-infected WT mice with or without GSSSG treatment).
- We have revised the top values of Y-axes in panels of each immune cell subset.
- We have rearranged the panels in the same order between Fig.1a, b, c, f, g h and Fig.1k-p.

Comment:

C.10b: In the legend, the authors need to indicate mouse age and background strain and how the control(uninfected) animals were treated (i.e., were they inoculated with PBS, media, nothing?)

Response:

R.10b: We have described mouse age, background strain and treatment for control in the new Methods (P. 18, L. 504–505, L. 524) but not in the figure legend, because of strict word limitation, as you would see. For influenza virus administration experiments, the group that is received only PBS served as a vehicle control.

Comment:

C.10c: Were the presented experiments performed more than once, or are the data shown from a single experiment?

Response:

R.10c: Each experminet was performed once. We added this information in figure legend.

Comment:

C.10d: Were the same mice used for Fig. 1a and 1b? If so, why not show the body weights through day 14?

Response:

R.10d: We used the same mice both for evaluating survival and body mass. However, most of the CARS2+/- mice died, or body weight of the mice was reduced to the extent that the mice had to be sacrificed according to the guideline of animal experiment. Thus, the change of the body weight was monitored up to 6 days as shown in Figure 1b.

Comment:

C.10e. Fig. 1c:

- i. The lung injury scores seem very low, and description provided in the Methods section is insufficient to describe how these scores were calculated.
- ii. The statistical comparison between mock/uninfected Cars2+/- mice and influenza-infected WT mice doesn't make sense.
- iii. Typically, pathology scores are expressed as ordinal numbers (e.g., a score of 1 to 5 based on disease severity) for a given parameter. As such, it is not correct to use mean and standard deviation to represent data from a group of animals. Rather, data should be expressed by using medians and confidence intervals, and statistically tested by using non-parametric methods.

Response:

R.10e: According to the reviewer's suggestion, we revised the method section as follows.

i: We have added other references to show more detailed calculation of lung injury score.

II: We agree with the reviewer's comment. We have deleted the comparison between Cars+/- mice and Flu-infected WT mice.

III: Following the reviewer's suggestion, we have changed expression of the data in the figure and performed nonparametric statistical analysis using Dunn's test for comparison of lung injury in each group (new Fig. 1c; Supplementary Methods, L. 882–884).

Comment:

C.10f. Fig. 1d: The use of light blue and pink in this panel (for day 3 tissues) is confusing since it was used in Fig. 1a-1c to represent mock/uninfected animals.

Response:

R.10f: The type of color used in figure is now modified to avoid people's confusion.

Comment:

C.10g. Fig. 1e and 1f: The differences between groups could be better represented if the top values on the Y-axes were reduced.

Response:

R.10g: The scale of Y-axis is now changed so that each data can be seen more clearly.

Comment:

C.11. Extended Data Fig. 2 and legend:

- a. Panel a is fuzzy.
- b. Panel d—the statistical test result bar seems to be out of place.
- c. Panel e—Is there a lighter exposure to better highlight the differences?

Response:

R.11a-c: We have replaced the images of panel a with a lighter image which shows the difference more clearly. Panels d and e are also replaced with new ones with better illustrated, as was suggested by the reviewer.

Comment:

C.12: Line 149 (results): The authors should use either 3CLpro or Mpro, but not both abbreviations in the manuscript text.

Response:

R.12: We have now decided to use “3CLpro” in a consistent manner.

Comment:

C.13. Line 158 (results): “...GSSSG potently inhibited SARS-CoV-2 propagation: virus replication was appreciably reduced...” As already stated, the authors have not measured virus replication/propagation, since they have not quantified the production of infectious virus.

Response:

R.13: The antiviral activities of either endogenously or exogenously generated supersulfides are extensively studied here in additional experiments. For example, we performed the plaque reduction assay, and found that endogenous supersulfides can exhibit anti-SARS-CoV-2 activity in the cells during the viral infection (Fig. 2a). The increased viral yield observed in the CARS2/CPERS KD cells was abolished by exogenously added GSSSG (Supplementary Fig. 4d,f; related statements in the text, P. 6, L. 162–164). We also carried out the SARS-CoV-2 infection study in two animal models (mice and hamsters) in vivo. Regarding the mouse COVID-19 model, we first developed ACE2-transgenic and AINK double-mutant (ACE2-Tg::*Cars2*^{AINK/+}) mice to effectively produce the SARS-CoV-2 infection in mice, because the wild-type mice, which originally lack ACE2, had to be genetically engineered to express ACE2 to be fully susceptible to SARS-CoV-2 in vivo. Lethality induced by the intranasal infection with SARS-CoV-2 was significantly higher in ACE2-AINK-mutant mice than in the ACE2-transgenic mice. The enhanced pathogenesis observed with the introduction of AINK mutation is likely to be caused by the accelerated viral growth in the lungs of ACE2-AINK-mutant mice compared with that in ACE2-transgenic mice. It is therefore concluded that supersulfides generated by CARS2/CPERS contribute to the anti-SARS-CoV-2 host defense in vivo. This interpretation is supported by the following new findings as well as those originally shown. The persulfide production of ACE2-AINK mutants was decreased compared with that of wild-type mice during SARS-CoV-2 infection. In addition, we also investigated the effect of GSSSG administered intraperitoneally (i.p.) on the SARS-CoV-2 pneumonia induced in hamsters. GSSSG treatment significantly suppressed the viral replication and ameliorated pneumonia pathology as manifested by physical conditions such as body weight loss and pulmonary consolidation.

It is therefore interpreted that supersulfides have a potent anti-SARS-CoV-2 activity and thus mediates the innate host defense against COVID-19. These findings are now stated in the revised manuscript: new Fig. 3 and Supplementary Fig. 8; related statements in the text, P. 7. L. 183–P. 8, L. 209.

Comment:

C.14. Lines 171-174 (results): “Of great importance is a marked increase in levels of sulphur metabolites, particularly sulphite and HS_2O_3^- in the EBC of a patient (as indicated by arrows) just before manifesting a progressive disease caused by severe viral pneumonia.”

a. In Fig. 2g, which metabolite is sulphite?

Response:

R.14-a: “Sulphite” is an idiomatic expression of “ HSO_3^- ”.

Comment:

C.14-b: How many supersulphide metabolites were detected? If more than the three shown in Fig. 2g, the authors should probably provide the additional data in a supplementary figure or table.

Response:

R.14-b: The number of breath sulfur metabolites (of low-molecular-weight that we can measure at this moment) is 10 molecular species. It varies somewhat depending on the sampling conditions, but at least five of them, such as HSH, HSSH, HSSSH, HSO_3^- (sulfite), and HSSO_3^- (thiosulfate), are consistently detected and can be precisely quantified in the EBC samples. Therefore, we now show the data of five sulfur metabolites are included herein. Please see Fig. 5, Supplementary Fig. 14: related description, P. 10, L. 278–P. 11, L. 312.

Comment:

C.14-c: While this finding is interesting, its importance is not clear because it is a single data point, and other patients with non-exacerbated COVID-19 have higher levels of some sulphur metabolites.

Response:

R.14-c: As pointed out and requested by this reviewer, we further recruited patients to increase (almost twice) the number of samples for the breath analysis. Even after addition of new subjects, a single subject still had the highest levels of HSO_3^- (sulfite), and HSSO_3^- (thiosulfate). Moreover, quite high levels of HSO_3^- (sulfite), and HSSO_3^- (thiosulfate) in EBC were observed not only with a human subject but also the hamster model of COVID-19. Therefore, we speculate that these oxidized sulfides could be produced from supersulfides through their regulatory and scavenging functions for reactive oxygen species, which suggests that the oxidative stress response mediated by supersulfides may be a hallmark of the COVID-19 pathogenesis. This notion is now added in the new text: P. 10, L. 278–P. 11, L. 312; the legend (L. 241-244) of Supplementary Fig. 14.

Comment:

C.15. Lines 175-177 (results): “The present... findings... suggest... the application of these sulphurmetabolites as excellent biomarkers for risk assessment of severe illness caused by coronavirus.”

a. The authors have not stratified their data by disease severity, so there is no way to know whether the sulphur metabolites may be used to differentiate different severity groups.

Response:

R.15-a: As we responded earlier, we have been trying our best to recruit additional COVID-19

patients, so that we can conduct stratified sampling for rigorous statistical analysis to determine the significance of supersulfides as biomarkers as detected by breath omics. Unfortunately, however, it was quite difficult for us to collect breath aerosol (exhaled air) from severely or critically ill COVID-19 patients. Almost all samples were collected from moderately affected patients without severe lung damage or dysfunction, and a few samples were collected from cases with mild pneumonia and other complications, if any. Because of this limitation, although each EBC value of supersulfides is significantly higher in the COVID-19 individuals than in healthy controls, there is no statistical difference in terms of stratified comparison, as long as we examined the patients with and without severe illness or long-term hospitalization.

These comments and data have been added in the revised text (P. 10, L. 289–P. 11, L. 300) and in the new Fig. 5a and Supplementary Fig. 14a.

Comment:

C.15-b: A rigorous statistical analysis for biomarker identification has not been performed; therefore, the authors cannot state one way or another if sulphur metabolites might be useful as biomarkers.

Response:

R.15-b: As just mentioned, because we got some limitations in extending our clinical study, we developed a hamster model for the breath analysis to experimentally identify biomarkers of COVID-19. The hamster breath analysis thus revealed that almost the same sulfur metabolites are remarkably elevated in the course of SARS-CoV-2 infections of the lungs, which correlated well with the time course of body weight loss and pulmonary consolidation, as shown in the new Fig. 5b-f and Supplementary Fig. 14b-f. Also, it should be noted that the profile of the sulfur metabolites detected with the Hamster EBC was almost consistent with that of human EBC from the COVID-19 patients. We thus interpreted that supersulfides are potential biomarkers for the COVID-19.

This explanation is now added in the revision: P. 11, L. 301–L. 312 (also see Fig. 5b-f and Supplementary Fig. 14b-f).

Comment:

R.15-c: Put simply, the finding is overstated based on the given data, and the text must be revised in the absence of additional analyses confirming the findings.

Response:

R.15-c: We have then modified appropriately as described above.

Comments:

C.16: At several points in the manuscript the authors use the phrases “sulphur-omics” or “breath-omics”(and once “sulphur breath-omics”). It is not always clear whether these phrases refer to the same or different analyses. Also, are the authors really performing omics analyses, or would their work be better characterized as targeted metabolomics?

Response:

R.16: We apologize if we confused the reviewer on this point. The breath omics means here integrated metabolome and proteome for breath aerosol samples, EBC, collected from human and animals. The current study employed mostly targeted omics approach, but we have another

option for non-target breath omics, of which development and translational application are now going on by our teams and collaborators in Japan. We will briefly explain the “breath omics” in the Methods section in the new text (P. 21, L. 609–P. 22, L. 643).

Comment:

C.17: Figure 2 and Figure 2 legend:

a. Fig. 2a: Somewhere, there should be some explanation of Ebselen.

Response:

R.17-a: The information in Ebselen is now added in the Methods: P. 20, L. 568-570.

Comment:

R.17-b. Fig. 2e: It is not clear what is being represented in this panel. More explanation is needed in the legend and/or the text of the results.

Response:

R.17-b: This data is fully updated and corrected so that it can be clearly understood (new Fig. 4e,f and Supplementary Fig. 9).

Comment:

R.17-c. Fig. 2f: Please include error bars.

Response:

R.17-c. Fig. 2f: The data have been modified and much more extended by several new experiments. Please see the above responses.

Comments:

R.17-d. Fig. 2g and 2h:

i. From the information available, it seems there may be up to 4 severity groups in this analysis: healthy, COVID-19/exacerbated, COVID-19/non-exacerbated, and recovered. However, in panel g, the only comparison is between healthy and COVID-19 patients. Would it be better to reanalyze the data with COVID-19 patients further subsetted based on disease severity? How many of the COVID-19 patients had exacerbated disease, and how many were recovered? What time points were the samples collected (with respect to symptom onset)?

ii. Can the authors infer anything about active pathways based on the supersulphide metabolites that were detected?

Response:

R.17-d. Please see the above responses.

C18: Lines 182-183 (results): “COPD is one of the highest risk conditions ever documented...” This statement is not very clear. From the reference at the end of the sentence, it appears to be referring to risk for severity of COVID-19. If so, the authors should state this more explicitly.

Response:

R.18. We have modified the statement accordingly.

Comments:

C.19: Fig. 3c is too fuzzy.

Response:

R.19: This has been improved and replaced with new high-resolution images (new Fig. 6c).

Comment:

C.20: The authors mention the use of Cars2AINK⁻ mice on line 199, but the generation of these mice is not described until the next section (starting from line 226).

Response:

R.20: This information is now added in the appropriate place and context in the new text (P. 7, L. 185).

Comment:

C.21: Lines 321-325 (discussion): “Because COPD is the highest risk condition known and IPF is the most serious complication reported, supersulphide-mediated airway protection may greatly benefit GSSSG therapeutics for COVID-19, which should thus be a superior approach to simple antiviral chemotherapy now being developed worldwide.” It is not totally clear what the authors are trying to say here, but perhaps they mean that (i) individuals with COPD are at high risk for severe COVID-19, (ii) IPF is one of the most severe complications of COVID-19, (iii) GSSSG can ameliorate COPD and IPF (and maybe COVID-19), so (iv) GSSSG may be a good therapeutic approach for COVID-19 treatment.

a. The authors need to revise this quote to state it more clearly.

Response:

R.21-a: We thank the reviewer for pointing out this awkward writing, which has been now corrected accordingly (P. 15, L. 433–437).

Comment:

C.21-b: It is important to note that the effect of GSSSG treatment has not been tested in an animal model of COVID-19, and the effects of GSSSG on COVID-19 outcome (including the risk of individuals with COPD and the development of IPF) in vivo is completely unknown.

Response:

R.21-b: As responded above, we have performed several additional infection models.

Comment:

C.22: Line 346 (discussion): Please reword the phrase “seminal published papers.”

Response:

R.22: We have deleted the words in this revision.

Comment:

C.23: Methods:

a. Line 768 (methods): What were BEAS-2B cells used for?

Response:

R.23-a: BEAS-2B cells were only used for positive control of CARS2 in the Western Blotting. Therefore, we deleted this description in the revised manuscript.

Comment:

C.23-b. Lines 792-800 (methods): The description of histopathology analysis is insufficient. At a minimum, please add

- i. The references used to design the analysis
- ii. The specific parameters that were scored
- iii. The scoring scale and the definition of each number in the scale
- iv. The number of sections scored per animal (and if more than one, how far apart from each other?)
- v. The calculation of the final score
- vi. The number of pathologists that analyzed the sections and whether they were blinded

Response:

R.23-b: According to the reviewer's suggestion, we have revised the method section as below (see "Histopathology").

i: We have added other references to show more detailed calculation of lung injury score (Am J Respir Cell Mol Biol Vol 44. pp 725–738, 2011 DOI: 10.1165/rcmb.2009-0210ST. European Respiratory Journal 2012 39: 1162-1170; DOI: 10.1183/09031936.00093911).

II-V: Two or three sections per lung and seven to nine lungs per experimental group were characterized for lung injury scoring analysis. At least 20 random high-power fields (400× total magnification) were independently scored. Lung injury was assessed on a scale of 0–2 for each of the following criteria: i) neutrophils in the alveolar space, ii) neutrophils in the interstitial space, iii) number of hyaline membranes, iv) amount of proteinaceous debris, and v) extent of alveolar septal thickening. The final injury score was derived from the following calculation: $\text{Score} = [20*(i) + 14*(ii) + 7*(iii) + 7*(iv) + 2*(v)] / (\text{number of fields} * 100)$, which finally gives an overall score of between 0 and 1.

vi: Lung injury scores were quantified by two investigators blinded to the treatment groups.

Comment:

R.23-c. Line 826 (methods): HFL-1 cells were not mentioned in the earlier cell culture section.

Response:

R.23-c: We have revised it in the Supplementary Methods section (see "Primary human lung-resident cell study").

Comment:

C.23-d. Lines 859-868 (methods): The authors should explicitly state that the substrate contains a moiety (MCF) that gives fluorescent signal when released from the substrate.

Response:

R.23-d: We have clearly explained this (see "Protease analysis").

Comment:

C.23-e. Line 917 (methods): “EBC was frozen quickly...” How was this done?

Response:

R.23-e: Briefly, the collected EBC was frozen quickly by a peltier cooling unit. We added detailed description of the method for the collection of human EBC method. Consequently, the description of the collection of the hamster EBC method has been subjected to slight modifications in line with the aforementioned account. Please see new Methods: P. 21, L.609–628.

REVIEWERS' COMMENTS

Reviewer #1 (Remarks to the Author):

C2 and R2. Ref.53 is authored by the same group. Papers of other groups are also to be discussed.

Reviewer #2 (Remarks to the Author):

The authors aimed to demonstrate the airway protective role of supersulphides and their therapeutic potential. They have appropriately addressed my concerns. With the new data produced, the conclusions are now convincing. So, the revised version is publishable.

Chang-Guo Zhan

Reviewer #3 (Remarks to the Author):

Auhtors responded poin-by-point to my comments sufficiently. I have no more concern to this manuscript.

Reviewer #4 (Remarks to the Author):

Overall, the authors have sufficiently addressed the original review. They did not provide additional experiments in the influenza mouse model (see C.ii and R.ii in the rebuttal document). However, substantial additional work in SARS-CoV-2 mouse and hamster models reinforces findings from the influenza mouse model. Moreover, the extension of breath supersulphur-omics to the hamster model and the alignment of these findings with that observed in humans is interesting. I am satisfied with their revision, though I do still think having an English editor look over the manuscript could improve the clarity substantially. A few additional minor comments about the revisions are describe below:

Figure 1d should not be labeled “Virus Yield” since the measure is viral RNA.

Line 129 and Fig. 1e—The authors may consider mentioning (for non-experts) that inorganic polysulphides (Na₂S₂, etc.) are supersulphides the first time their use is noted in the text.

Line 230—“integration into host cells” is not good phrasing. Influenza viruses enter (or infect) host cells, but do not integrate into them.

Lines 221-238—

- Most of this paragraph probably belongs in the Discussion section.
- “(data not shown)” —Does Nature Communications accept data not shown statements?
- “and besides, it may be an Achilles heel commonly expressed among all viruses including SARS-CoV-2 and influenza virus.” This phrase is either true or not. Are all viral receptor proteins dependent on disulphide formation? Have the authors checked this? If they are not sure, they may consider toning the statement down.

Lines 303-304—“which correlated well with the time course of body weight loss and pulmonary consolidation...” If the authors say the data correlated well, it would be best to show a measure of correlation.

Lines 468-471—“This finding may have important implications to better understand how co-infection of influenza virus and SARS-CoV-2 exacerbated the viral pathogenesis and thereby worsened the clinical outcome or in-hospital mortality, which has been reported recently.” Although supersulphides affect pathogenesis of both viruses, this does not imply anything about co-infection. I suggest leaving this statement out unless the authors want to elaborate their meaning more specifically.

Responses to the comments from the reviewers

Reviewer #1 (Remarks to the Author)

C2 and R2. Ref.53 is authored by the same group. Papers of other groups are also to be discussed.

Response:

We appreciate gratefully the time and effort you dedicated to reviewing our paper. Your commitment is highly commendable and in fact of great help in polishing up our manuscript. We additionally referenced major works by other groups (ref.6,7) in the sentence (L. 491).

Reviewer #2 (Remarks to the Author)

The authors aimed to demonstrate the airway protective role of supersulphides and their therapeutic potential. They have appropriately addressed my concerns. With the new data produced, the conclusions are now convincing. So, the revised version is publishable.

Chang-Guo Zhan

Response:

We would like to thank so much again for your precious time and effort in reviewing our paper. Your thoughtful comments on our work are gratefully appreciated.

Reviewer #3 (Remarks to the Author)

Authors responded point-by-point to my comments sufficiently. I have no more concern to this manuscript.

Response:

Thank you very much again for your positive evaluation of our work. We believe that our work is now greatly improved by your critical inputs.

Reviewer #4 (Remarks to the Author)

Overall, the authors have sufficiently addressed the original review. They did not provide additional experiments in the influenza mouse model (see C.ii and R.ii in the rebuttal document). However, substantial additional work in SARS-CoV-2 mouse and hamster models reinforces findings from the influenza mouse model. Moreover, the extension of breath supersulphur-omics to the hamster model and the alignment of these findings with that observed in humans is interesting. I am satisfied with their revision, though I do still think having an English editor look over the manuscript could improve the clarity substantially. A few additional minor comments about the revisions are describe below:

Comment:

C.1: Figure 1d should not be labeled “Virus Yield” since the measure is viral RNA.

Response:

R.1: Figure 1d is labelled as “Virus PCR”.

Comment:

C.2: Line 129 and Fig. 1e—The authors may consider mentioning (for non-experts) that inorganic polysulphides (Na₂S₂, etc.) are supersulphides the first time their use is noted in the text.

Response:

R.2: We added “supersulphide” for the term, inorganic hydropolysulphides (H₂S_n), which is the first time we used in the text (L. 129 in new text).

Comment:

C.3: Line 230—“integration into host cells” is not good phrasing. Influenza viruses enter (or infect) host cells, but do not integrate into them.

Response:

R.3: We changed the phrase of “integration into host cells” to “entering into host cells” in L. 460 in new text.

Comment:

C.4: Lines 221-238—

- Most of this paragraph probably belongs in the Discussion section.

Response:

R.4: Previous text in L. 228-234 is now transferred to the Discussion section (L. 458-468 in new text).

Comment:

C.5: • “(data not shown)”—Does Nature Communications accept data not shown statements?

Response:

R.5: Thank you very much for your pointing it out. The phrase of “data not shown” is thus deleted in L. 464.

Comment:

C.6: • “and besides, it may be an Achilles heel commonly expressed among all viruses including SARS-CoV-2 and influenza virus.” This phrase is either true or not. Are all viral receptor proteins dependent on disulphide formation? Have the authors checked this? If they are not sure, they may consider toning the statement down.

Response:

R.6: I am afraid that the reviewer may possibly have misunderstood in this part. We discussed right on the disulphide formation in spike and envelope proteins expressed by virus *per se*, as a critical target of supersulphide, rather than viral receptor proteins of the host cells (Please see L. 222-227, L. 458-468).

Comment:

C.7: Lines 303-304—“which correlated well with the time course of body weight loss and pulmonary consolidation...” If the authors say the data correlated well, it would be best to show a measure of correlation.

Response:

R.7: We then deleted that phrase “which correlated...”, and we rewrote its statement (L. XXX) so that the result we observed can be clearly explained in a way that is simple to understand the time profile of sulphur metabolism and pathological changes.

Comment:

C.8: Lines 468-471—“This finding may have important implications to better understand how co-infection of influenza virus and SARS-CoV-2 exacerbated the viral pathogenesis and thereby worsened the clinical outcome or in-hospital mortality, which has been reported recently.” Although supersulphides affect pathogenesis of both viruses, this does not imply anything about co-infection. I suggest leaving this statement out unless the authors want to elaborate their meaning more specifically.

Response:

R.8: This statement is modified according to the reviewer’s comment without mentioning the co-infection (L. 468-471 in new text).